# Neural Active Learning Beyond Bandits

**Yikun Ban**[1]**, Ishika Agarwal**[1]**, Ziwei Wu**[1]**, Yada Zhu**[2]**, Kommy Weldemariam**[2]**,
Hanghang Tong**[1]**, Jingrui He**[1]

[1]University of Illinois Urbana-Champaign, [2]IBM Research
[1]{yikunb2, ishikaa2, ziweiwu2, htong, jingrui}@illinois.edu
[2]{yzhu, kommy}@us.ibm.com

## Abstract

We study both stream-based and pool-based active learning with neural network approximations. A recent line of works proposed bandit-based approaches that transformed active learning into a bandit problem, achieving both theoretical and empirical success. However, the performance and computational costs of these methods may be susceptible to the number of classes, denoted as $K$, due to this transformation. Therefore, this paper seeks to answer the question: "How can we mitigate the adverse impacts of $K$ while retaining the advantages of principled exploration and provable performance guarantees in active learning?" To tackle this challenge, we propose two algorithms based on the newly designed exploitation and exploration neural networks for stream-based and pool-based active learning. Subsequently, we provide theoretical performance guarantees for both algorithms in a non-parametric setting, demonstrating a slower error-growth rate concerning $K$ for the proposed approaches. We use extensive experiments to evaluate the proposed algorithms, which consistently outperform state-of-the-art baselines.

## 1 Introduction

Active learning is one of the primary areas in machine learning to investigate the learning technique on a small subset of labeled data while acquiring good generalization performance compared to passive learning [19]. There are mainly two settings of active learning: stream-based and pool-based settings. For the stream-based setting, the learner is presented with an instance drawn from some distribution in each round and is required to decide on-the-fly whether or not to query the label from the oracle. For the pool-based setting, the learner aims to select one or multiple instances from the unlabeled pool and hand them over to the oracle for labeling. It repeats this process until the label budget is exhausted [51]. The essence of active learning is to exploit the knowledge extracted from labeled instances and explore the unlabeled data to maximize information acquisition for long-term benefits.

Using neural networks (NNs) to perform active learning has been explored extensively in recent works [48; 52; 56; 6]. However, they often lack a provable performance guarantee despite strong empirical performance. To address this issue, a recent line of works [58; 14] proposed the bandit-based approaches to solve the active learning problem, which are equipped with principled exploration and theoretical performance guarantee. In contextual bandits [38; 68], the learner is presented with $K$ arms (context vectors) and required to select one arm in each round. Then, the associated reward is observed. [58; 14] transformed the online $K$-class classification into a bandit problem. Specifically, in one round of stream-based active learning, a data instance $\mathbf{x}_t \in \mathbb{R}^d$ is transformed into $K$ long vectors corresponding to $K$ arms, matching $K$ classes: $\mathbf{x}_{t,1} = [\mathbf{x}_t^\top, \mathbf{0}^\top, \cdots, \mathbf{0}^\top]^\top, \ldots, \mathbf{x}_{t,K} = [\mathbf{0}^\top, \cdots, \mathbf{0}^\top, \mathbf{x}_t^\top]^\top$, where $\mathbf{x}_{t,k} \in \mathbb{R}^{dK}, k \in [K]$. Then, the learner uses an NN model to calculate a score for each arm and selects an arm based on these scores. The index of the selected arm represents the index of the predicted class. This design enables researchers to utilize the exploration strategy and analysis in contextual bandits to solve the active learning problem. Note [58; 14] can only handle the stream-based setting of active learning.

However, bandit-based approaches bear the following two limitations. First, as the instance $\mathbf{x}_t$ is transformed into $K$ arms, it is required to calculate a score for all $K$ arms respectively, producing a cost of $K$ times forward-propagation computation of neural networks. This computation cost is scaled by $K$. Second, the transformed long vector (arm) has $(Kd)$ dimensions, in contrast to the

Table 1: Test accuracy and running time compared to bandit-based methods in stream-based setting.

| | Adult | MT | Letter | Covertype | Shuttle | Fashion |
|---|---|---|---|---|---|---|
| # Number of classes | 2 | 2 | 2 | 7 | 7 | 10 |
| | Accuracy | | | | | |
| NeurAL-NTK [58] | $80.3 \pm 0.12$ | $76.9 \pm 0.15$ | $79.3 \pm 0.21$ | $61.9 \pm 0.08$ | $95.3 \pm 0.20$ | $64.5 \pm 0.16$ |
| I-NeurAL [14] | $84.2 \pm 0.22$ | $79.4 \pm 0.16$ | $82.9 \pm 0.06$ | $65.2 \pm 0.19$ | $99.3 \pm 0.12$ | $73.5 \pm 0.28$ |
| NEURONAL-S | $\mathbf{84.8 \pm 0.51}$ | $\mathbf{83.7 \pm 0.17}$ | $\mathbf{86.5 \pm 0.16}$ | $\mathbf{74.4 \pm 0.19}$ | $\mathbf{99.5 \pm 0.09}$ | $\mathbf{83.2 \pm 0.38}$ |
| | Running Time | | | | | |
| NeurAL-NTK [58] | $163.2 \pm 1.31$ | $259.4 \pm 2.48$ | $134.0 \pm 3.44$ | $461.2 \pm 1.26$ | $384.7 \pm 1.86$ | $1819.4 \pm 10.84$ |
| I-NeurAL [14] | $102.4 \pm 7.53$ | $46.2 \pm 5.58$ | $232.2 \pm 3.80$ | $1051.7 \pm 5.85$ | $503.1 \pm 9.66$ | $1712.7 \pm 12.8$ |
| NEURONAL-S | $\mathbf{54.7 \pm 3.21}$ | $\mathbf{10.5 \pm 0.39}$ | $\mathbf{92.1 \pm 3.4}$ | $\mathbf{166.4 \pm 0.59}$ | $\mathbf{101.2 \pm 2.32}$ | $\mathbf{116.3 \pm 3.39}$ |

$d$ dimensions of the original instance as the input of the NN model. This potentially amplifies the effects of $K$ on an active learning algorithm's performance. We empirically evaluate [58; 14] as shown in Table 1. The results indicate a noticeable degradation in both test accuracy and running time as $K$ increases.

In response, in this paper, we aim to mitigate the adverse effects of $K$ on the bandit-based approach in active learning. Our methods are built upon and beyond [14]. [14] adopted the idea of [13] to employ two neural networks, one for exploitation and another for exploration. As previously mentioned, these two neural networks take the transformed $Kd$-dimension arm as input. Moreover, in each round, [14] decomposed the label vector $\mathbf{y}_t \in \{0, 1\}^K$ into $K$ rewards (scalars), necessitating the training of two neural networks $K$ times for each arm. Next, we summarize our key ideas and contributions to reduce the input dimension back to $d$ and the number of forward propagations to 1 in each round while preserving the essence of exploitation and exploration of neural networks.

*Methodology.* (1) We extend the loss function in active learning from 0-1 loss to Bounded loss, which is more flexible and general. Instead, [58; 14] restricted the loss to be 0-1 loss, because they had to define the reward of each class (arm) due to their bandit-based methodology. (2) We re-designed the input and output exploitation and exploration neural networks to directly take the $d$-dimension instance as input and output the predicted probabilities for $K$ classes synchronously, mitigating the curse of $K$. The connection between exploitation and exploration neural networks is also reconstructed beyond the standard bandit setting. In other words, we avoid the transformation of active learning to the standard bandit setting. This is the first main contribution of this paper. (3) To facilitate efficient and effective exploration, we introduce the end-to-end embedding (Definition 4.1) as the input of the exploration neural network, which removes the dependence of the input dimension while preserving the essential information. (4) In addition to our proposed stream-based algorithm, referred to NEURONAL-S, we also propose a pool-based active learning algorithm, NEURONAL-P. We bring the redesigned exploitation and exploration network into pool-based setting and propose a novel gap-inverse-based selection strategy tailored for pool-based active learning. This is our second main contribution. Note that the stream-based algorithms cannot be directly converted into the pool-based setting, as discussed in Appendix B.

*Theoretical analysis.* We provide the regret upper bounds for the proposed stream-based algorithm under low-noise conditions on the data distribution. Our results indicate the cumulative regret of NEURONAL-S grows slower than that of [58] concerning $K$ by a multiplicative factor at least $\mathcal{O}(\sqrt{T \log(1 + \lambda_0)})$ and up to $\widetilde{\mathcal{O}}(\sqrt{md})$, where $\lambda_0$ is the smallest eigenvalue of Neural Tangent Kernel (NTK) and $m$ is the width of the neural network. This finding helps explain why our algorithms outperform the bandit-based algorithms, particularly when $K$ is large, as shown in Table 1. In the binary classification task, our regret bounds directly remove the dependence of effective dimension $\tilde{d}$, which measures the actual underlying dimension in the RKHS space spanned by NTK, discussed in Sec. 5. We also provide a performance analysis for the proposed pool-based algorithm in the non-parametric setting, tailored for neural network models. In contrast, previous works focus on the regime either in parametric settings that require a finite VC dimension [31] or a linear mapping function assumption [8; 64; 28]. The above theoretical results are our third main contribution. In addition,

*Empirical evaluation.* In the end, we perform extensive experiments to evaluate the proposed algorithms for both stream-based and pool-based algorithms compared to state-of-the-art baselines. Our evaluation encompasses various metrics, including test accuracy and running time, and we have carried out ablation studies to investigate the impact of hyper-parameters and label budgets. This is our fourth main contribution.

## 2 RELATED WORK

Active learning has been studied for decades and adapted to many real-world applications [53]. There are several strategies in active learning to smartly query labels, such as diversity sampling [23; 67], and uncertainty sampling [69; 62; 41]. Neural networks have been widely used in various scenarios[40; 39; 26; 27; 59; 60]. The Contextual bandit is a powerful tool for decision-making [13; 10; 9; 11; 49; 50] and has been used for active learning. Works that use neural networks to perform active learning, according to the model structure, can be classified into online-learning-based [45; 6] and bandit-based [58; 14] methods. For most of the methods, their neural models take the instance as input and calculate the predicted scores for each class synchronously. For example, [45; 6; 17; 35; 54; 56; 66; 5] exploit the neural networks for active learning to improve the empirical performance. As mentioned before, this type of related work often lacks the principled exploration and provable guarantee in a non-parametric setting.

The primary focus of theoretical research in active learning is to determine the convergence rate of the population loss (regret) of the hypothesis produced by the algorithm in relation to the number of queried labels $N$. In the parametric setting, the convergence rates of the regret are of the form $\nu N^{1/2} + e^{-\sqrt{N}}$, where $\nu$ is the population loss of the best function in the class with finite VC-dimension, e.g., [29; 20; 47; 63; 7; 65]. With the realizable assumption (i.e., when the Bayes optimal classifier lies in the function class), minimax active learning rates of the form $N^{-\frac{\alpha+1}{2}}$ are shown in [30; 36] to hold for adaptive algorithms that do not know beforehand the noise exponent $\alpha$ (Defined in Sec. 5). In non-parametric settings, [44; 42; 70] worked under smoothness (Holder continuity/smoothness) assumptions, implying more restrictive assumptions on the marginal data distribution. [42; 70] achieved the minimax active learning rate $N^{-\frac{\beta(\alpha+1)}{2\beta+d}}$ for $\beta$-Holder classes, where exponent $\beta$ plays the role of the complexity of the class of functions to learn. [8; 64; 28] focused on the pool-based setting and achieve the $(\frac{d}{N})^{\alpha+1/\alpha+2}$ minimax rate, but their analysis are built on the linear mapping function. [37] investigated active learning with nearest-neighbor classifiers and provided a data-dependent minimax rate based on the noisy-margin properties of the random sample. [32] introduce the expected variance with Gaussian processes criterion for neural active learning. [58] and [14] represent the closest works related to the analysis of neural networks in active learning. Both [58] and our work utilized the theory of NTK, making their results directly comparable. [14] established a connection between their regret upper bound and the training error of a neural network function class. However, they did not provide a clear trade-off between the training error and the function class radius, making it challenging to compare [14] with our theoretical results.

## 3 PROBLEM DEFINITION

In this paper, we consider the $K$-class classification problem for both stream-based and pool-based active learning.

Let $\mathcal{X}$ denote the input space over $\mathbb{R}^d$ and $\mathcal{Y}$ represent the label space over $\{0,1\}^K$. $\mathcal{D}$ is some unknown distribution over $\mathcal{X} \times \mathcal{Y}$. For any round $t \in [T] = \{1, 2, \ldots, T\}$, an instance $\mathbf{x}_t$ is drawn from the marginal distribution $\mathcal{D}_{\mathcal{X}}$. Accordingly, the associated label $\mathbf{y}_t$ is drawn from the conditional distribution $\mathcal{D}_{\mathcal{Y}|\mathbf{x}_t}$, in which the dimension with value 1 indicates the ground-truth class of $\mathbf{x}_t$. For the clarity of presentation, we use $\mathbf{y}_{t,k}, k \in [K]$ to represent the $K$ possible predictions, i.e., $\mathbf{y}_{t,1} = [1, 0, \ldots, 0]^\top, \ldots, \mathbf{y}_{t,K} = [0, 0, \ldots, 1]^\top$.

Given an instance $\mathbf{x}_t \sim \mathcal{D}_{\mathcal{X}}$, the learner is required to make a prediction $\mathbf{y}_{t,\widehat{k}}, \widehat{k} \in [K]$ based on some function. Then, the label $\mathbf{y}_t$ is observed. A loss function $\ell : \mathcal{Y} \times \mathcal{Y} \to [0, 1]$, represented by $\ell(\mathbf{y}_{t,\widehat{k}}, \mathbf{y}_t)$, reflects the quality of this prediction. We investigate the non-parametric setting of active learning. Specifically, we make the following assumption for the conditional distribution of the loss.

**Assumption 3.1.** The conditional distribution of the loss given $\mathbf{x}_t$ is defined by some unknown function $\mathbf{h} : \mathcal{X} \to [0, 1]^K$, such that

$$\forall k \in [K], \mathbb{E}_{\mathbf{y}_t \sim \mathcal{D}_{\mathcal{Y}|\mathbf{x}_t}} [\ell(\mathbf{y}_{t,k}, \mathbf{y}_t)|\mathbf{x}_t] = \mathbf{h}(\mathbf{x}_t)[k], \tag{3.1}$$

where $\mathbf{h}(\mathbf{x}_t)[k]$ represents the value of the $k^{th}$ dimension of $\mathbf{h}(\mathbf{x}_t)$.

Assumption 3.1 is the standard formulation in [58; 14]. Related works [14; 58] restricted $\ell$ to be 0-1 loss. In contrast, we only assume $\ell$ is bounded for the sake of generality.

**Stream-based**. For stream-based active learning, at each round $t \in [T]$, the learner receives an instance $\mathbf{x}_t \sim \mathcal{D}_\mathcal{X}$. Then, the learner is compelled to make a prediction $\mathbf{y}_{t,\widehat{k}}$, and at the same time, decides on-the-fly whether or not to observe the label $\mathbf{y}_t$.

Then, the goal of active learning is to minimize the *Population Cumulative Regret*. Given the data distribution $\mathcal{D}$ and the number of rounds $T$, the Population Cumulative Regret is defined as:

$$\mathbf{R}_{stream}(T) := \sum_{t=1}^{T} \Big[ \mathbb{E}_{(\mathbf{x}_t, \mathbf{y}_t) \sim \mathcal{D}} [\ell(\mathbf{y}_{t,\widehat{k}}, \mathbf{y}_t)] - \mathbb{E}_{(\mathbf{x}_t, \mathbf{y}_t) \sim D} [\ell(\mathbf{y}_{t,k^*}, \mathbf{y}_t)] \Big], \tag{3.2}$$

where $\mathbf{y}_{t,k^*}$ is the prediction induced by the Bayes-optimal classifier with respect to $\mathbf{x}_t$ in round $t$, i.e., $k^* = \arg\min_{k \in [K]} \mathbf{h}(\mathbf{x}_t)[k]$. The Population Regret reflects the generalization performance of a model in $T$ rounds of stream-based active learning.

At the same time, the goal of the learner is to minimize the expected label query cost: $\mathbf{N}(T) := \sum_{t=1}^{T} \mathbb{E}_{\mathbf{x}_t \sim \mathcal{D}_\mathcal{X}} [\mathbf{I}_t | \mathbf{x}_t]$, where $\mathbf{I}_t$ is an indicator of the query decision in round $t$ such that $\mathbf{I}_t = 1$ if $\mathbf{y}_t$ is observed; $\mathbf{I}_t = 0$, otherwise.

**Pool-based**. The goal pool-based active learning is to maximize the performance of a model with a limited label budget given the pool of instances. In a round, assume there is an instance pool $\mathbf{P}_t = \{\mathbf{x}_1, \mathbf{x}_2, \ldots, \mathbf{x}_B\}$ with $B$ instances, which are drawn from $\mathcal{D}_\mathcal{X}$. Then, for any $\mathbf{x}_t \in \mathbf{P}_t$, the learner is able to request a label from the conditional distribution $\mathbf{y}_t \sim \mathcal{D}_{\mathcal{Y}|\mathbf{x}_t}$ with one unit of budget cost. The total number of queries is set as $Q$. Let $\mathbf{y}_{t,\widehat{k}}$ be the prediction of $\mathbf{x}_t \in \mathbf{P}_t$ by some hypothesis. Then, the Population Cumulative Regret for pool-based active learning is defined as:

$$\mathbf{R}_{pool}(Q) := \sum_{t=1}^{Q} \Big[ \mathbb{E}_{(\mathbf{x}_t, \mathbf{y}_t) \sim \mathcal{D}} [\ell(\mathbf{y}_{t,\widehat{k}}, \mathbf{y}_t)] - \mathbb{E}_{(\mathbf{x}_t, \mathbf{y}_t) \sim D} [\ell(\mathbf{y}_{t,k^*}, \mathbf{y}_t)] \Big], \tag{3.3}$$

where $k^* = \arg\min_{k \in [K]} \mathbf{h}(\mathbf{x}_t)[k]$. $\mathbf{R}_{pool}(Q)$ reflects the generalization performance of a model in $Q$ rounds of pool-based active learning.

## 4 PROPOSED ALGORITHMS

In this section, we present the proposed algorithms for both stream-based and pool-based active learning. The proposed NN models incorporate two neural networks for exploitation and exploration. The exploitation network directly takes the instance as input and outputs the predicted probabilities for $K$ classes synchronously, instead of taking the transformed $K$ long vectors as input and computing the probabilities sequentially. The exploration network has a novel embedding as input to incorporate the information of $K$ classes in the exploitation network simultaneously instead of calculating $K$ embeddings for $K$ arms sequentially as in bandit-based approaches.

**Exploitation Network $\mathbf{f}_1$.** The exploitation network $\mathbf{f}_1$ is a neural network which learns the mapping from input space $\mathcal{X}$ to the loss space $[0,1]^K$. In round $t \in [T]$, we denote the network by $\mathbf{f}_1(\cdot; \boldsymbol{\theta}_t^1)$, where the superscript of $\boldsymbol{\theta}_t^1$ indicates the network and the subscript indicates the index of the round after updates for inference. $\mathbf{f}_1$ is defined as a fully-connected neural network with $L$-depth and $m$-width:

$$\mathbf{f}_1(\mathbf{x}_t; \boldsymbol{\theta}^1) := \sqrt{m} \mathbf{W}_L^1 \sigma(\mathbf{W}_{L-1}^1 \ldots \sigma(\mathbf{W}_1^1 \mathbf{x}_t))) \in \mathbb{R}^K, \tag{4.1}$$

where $\mathbf{W}_1^1 \in \mathbb{R}^{m \times d}, \mathbf{W}_l^1 \in \mathbb{R}^{m \times m}$, for $2 \leq l \leq L-1$, $\mathbf{W}_L^1 \in \mathbb{R}^{K \times m}$, $\boldsymbol{\theta}^1 = [\text{vec}(\mathbf{W}_1^1)^\top, \ldots, \text{vec}(\mathbf{W}_L^1)^\top]^\top \in \mathbb{R}^{p_1}$, and $\sigma$ is the ReLU activation function $\sigma(\mathbf{x}) = \max\{0, \mathbf{x}\}$. We randomly initialize $\boldsymbol{\theta}^1$ denoted by $\boldsymbol{\theta}_1^1$, where each entry is drawn from normal distribution $N(0, 2/m)$ for $\mathbf{W}_l^1, l \in [L-1]$, and each entry is drawn from $N(0, 1/(Km))$ for $\mathbf{W}_L^1$.

Note that we take the basic fully-connected network as an example for the sake of analysis in over-parameterized networks, and $\mathbf{f}_1$ can be easily replaced with other advanced models depending on the tasks. Given an instance $\mathbf{x}_t$, $\mathbf{f}_1(\mathbf{x}_t; \boldsymbol{\theta}_t^1)$ can be considered as the estimation for $\ell(\mathbf{y}_{t,k}, \mathbf{y}_t), k \in [K]$.

In round $t$, after receiving the label $\mathbf{y}_t$, we conduct stochastic gradient descent to update $\boldsymbol{\theta}^1$, based on the loss function $\mathcal{L}_{t,1}(\boldsymbol{\theta}_t^1)$, such as $\mathcal{L}_{t,1}(\boldsymbol{\theta}_t^1) = \sum_{k \in [K]} \big(\mathbf{f}_1(\mathbf{x}_t; \boldsymbol{\theta}_t^1)[k] - \ell(\mathbf{y}_{t,k}, \mathbf{y}_t)\big)^2 / 2$, where $\mathbf{f}_1(\mathbf{x}_t; \boldsymbol{\theta}_t^1)[k]$ represents the value of the $k^{\text{th}}$ dimension of $\mathbf{f}_1(\mathbf{x}_t; \boldsymbol{\theta}_t^1)$.

---

**Algorithm 1** NEURONAL-S

---

**Input:** $T, K, \mathbf{f}_1, \mathbf{f}_2, \eta_1, \eta_2$ (learning rate), $\gamma$ (exploration para.), $\delta$ (confidence para.), $S$ (norm para.)

1: Initialize $\boldsymbol{\theta}_1^1, \boldsymbol{\theta}_1^2; \widehat{\boldsymbol{\theta}}_1^1 = \boldsymbol{\theta}_1^1; \widehat{\boldsymbol{\theta}}_1^2 = \boldsymbol{\theta}_1^2, \Omega_1 = \{(\widehat{\boldsymbol{\theta}}_1^1, \widehat{\boldsymbol{\theta}}_1^2)\}$

2: **for** $t = 1, 2, \ldots, T$ **do**

3:      Receive an instance $\mathbf{x}_t \sim \mathcal{D}_{\mathcal{X}}$

4:      $\mathbf{f}(\mathbf{x}_t; \boldsymbol{\theta}_t) = \mathbf{f}_1(\mathbf{x}_t; \boldsymbol{\theta}_t^1) + \mathbf{f}_2(\phi(\mathbf{x}_t); \boldsymbol{\theta}_t^2)$

5:      $\widehat{k} = \arg\min_{k \in [K]} \mathbf{f}(\mathbf{x}_t; \boldsymbol{\theta}_t)[k]$

6:      $k^{\circ} = \arg\min_{k \in ([K] \setminus \{ \widehat{k} \})} \mathbf{f}(\mathbf{x}_t; \boldsymbol{\theta}_t)[k]$

7:      Predict $\mathbf{y}_{t,\widehat{k}}$

8:      $\mathbf{I}_t = \mathbb{1}\{|\mathbf{f}(\mathbf{x}_t; \boldsymbol{\theta}_t)[\widehat{k}] - \mathbf{f}(\mathbf{x}_t; \boldsymbol{\theta}_t)[k^{\circ}]| < 2\gamma\boldsymbol{\beta}_t\} \in \{0,1\}; \boldsymbol{\beta}_t = \sqrt{\frac{KS^2}{t}} + \sqrt{\frac{2\log(3T/\delta)}{t}}$

9:      Observe $\mathbf{y}_t$ and $\ell$, if $\mathbf{I}_t = 1$; $\mathbf{y}_t = \mathbf{y}_{t,\widehat{k}}$, otherwise

10:     $\widehat{\boldsymbol{\theta}}_{t+1}^1 = \widehat{\boldsymbol{\theta}}_t^1 - \eta_1 \nabla_{\widehat{\boldsymbol{\theta}}^1} \mathcal{L}_{t,1}(\widehat{\boldsymbol{\theta}}_t^1)$

11:     $\widehat{\boldsymbol{\theta}}_{t+1}^2 = \widehat{\boldsymbol{\theta}}_t^2 - \eta_2 \nabla_{\widehat{\boldsymbol{\theta}}^2} \mathcal{L}_{t,2}(\widehat{\boldsymbol{\theta}}_t^2)$

12:     $\Omega_{t+1} = \Omega_t \cup \{(\widehat{\boldsymbol{\theta}}_{t+1}^1, \widehat{\boldsymbol{\theta}}_{t+1}^2)\}$

13:     Draw $(\boldsymbol{\theta}_{t+1}^1, \boldsymbol{\theta}_{t+1}^2)$ uniformly from $\Omega_{t+1}$

14: **end for**

15: **return** $\boldsymbol{\theta}_T$

---

**Exploration Network $\mathbf{f}_2$.** The exploration network $\mathbf{f}_2$ learns the uncertainty of estimating $\mathbf{f}_1(\cdot; \boldsymbol{\theta}^1)$. In round $t \in [T]$, given an instance $\mathbf{x}_t$ and the estimation $\mathbf{f}_1(\mathbf{x}_t; \boldsymbol{\theta}_t^1)$, the input of $\mathbf{f}_2$ is a mapping or embedding $\phi(\mathbf{x}_t)$ that incorporates the information of both $\mathbf{x}_t$ and the discriminative information of $\boldsymbol{\theta}_t^1$. We introduce the following embedding $\phi(\mathbf{x}_t)$:

**Definition 4.1** (End-to-end Embedding). *Given the exploitation network $\mathbf{f}_1(\cdot; \boldsymbol{\theta}_t^1)$ and an input context $\mathbf{x}_t$, its end-to-end embedding is defined as*

$$\phi(\mathbf{x}_t)^{\top} = \left( \sigma(\mathbf{W}_1^1 \mathbf{x}_t)^{\top}, vec(\nabla_{\mathbf{W}_L^1} \mathbf{f}_1(\mathbf{x}_t; \boldsymbol{\theta}_t^1))^{\top} \right) \in \mathbb{R}^{m+Km}, \tag{4.2}$$

*where the first term is the output of the first layer of $\mathbf{f}_1$ and the second term is the partial derivative of $\mathbf{f}_1$ with respect to the parameters of the last layer. $\phi(\mathbf{x}_t)$ is usually normalized.*

The first term $\sigma(\mathbf{W}_1^1 \mathbf{x}_t)^{\top}$ can be thought of as an embedding to transform $\mathbf{x}_t \in \mathbb{R}^d$ into another space in $\mathbb{R}^m$, as the input dimension $d$ can be a very large number. Thus, we leverage the representation power of $\mathbf{f}_1$ and minimize the dependence on $d$ for $\mathbf{f}_2$. The last layer might play the more important role in terms of classification ability [46] and hence we only take the gradient of the last layer which incorporates the discriminative information of $\boldsymbol{\theta}_t^1$, to reduce the computation cost.

Then, specifically, given an embedding $\phi(\mathbf{x}_t)$, $\mathbf{f}_2$ is defined as:

$$\mathbf{f}_2(\phi(\mathbf{x}_t); \boldsymbol{\theta}^2) := \sqrt{m} \mathbf{W}_L^2 \sigma(\mathbf{W}_{L-1}^2 \ldots \sigma(\mathbf{W}_1^2 \phi(\mathbf{x}_t))) \in \mathbb{R}^K, \tag{4.3}$$

where $\mathbf{W}_1^2 \in \mathbb{R}^{m \times (m+Km)}, \mathbf{W}_l^2 \in \mathbb{R}^{m \times m}$, for $2 \leq l \leq L-1$, $\mathbf{W}_L^2 \in \mathbb{R}^{K \times m}$, $\boldsymbol{\theta}^2 = [vec(\mathbf{W}_1^2)^{\top}, \ldots, vec(\mathbf{W}_L^2)^{\top}]^{\top} \in \mathbb{R}^{p_2}$, and $\sigma$ is the ReLU activation function $\sigma(\mathbf{x}) = \max\{0, \mathbf{x}\}$. Similarly, we randomly initialize $\boldsymbol{\theta}^2$ denoted by $\boldsymbol{\theta}_1^2$, where each entry is drawn from Normal distribution $N(0, 2/m)$ for $\mathbf{W}_l^2, l \in [L-1]$, and each entry is drawn from $N(0, 1/Km)$ for $\mathbf{W}_L^2$.

In round $t$, after receiving $\mathbf{y}_t$, the label for training $\mathbf{f}_2$ is the estimated error of $\mathbf{f}_1(\cdot; \boldsymbol{\theta}^1)[k]$, represented by $(\ell(\mathbf{y}_{t,k}, \mathbf{y}_t) - \mathbf{f}_1(\mathbf{x}_t; \boldsymbol{\theta}_t^1)[k])$. Therefore, we conduct stochastic gradient descent to update $\boldsymbol{\theta}^2$, based on the loss $\mathcal{L}_{t,2}(\boldsymbol{\theta}_t^2)$, such as $\mathcal{L}_{t,2}(\boldsymbol{\theta}_t^2) = \sum_{k \in [K]} \Big( \mathbf{f}_2(\phi(\mathbf{x}_t); \boldsymbol{\theta}_t^2)[k] - \big(\ell(\mathbf{y}_{t,k}, \mathbf{y}_t) - \mathbf{f}_1(\mathbf{x}_t; \boldsymbol{\theta}_t^1)[k]\big) \Big)^2 / 2$, where $\mathbf{f}_2(\phi(\mathbf{x}_t); \boldsymbol{\theta}_t^2)[k]$ represents the value of the $k^{\text{th}}$ dimension of $\mathbf{f}_2(\phi(\mathbf{x}_t); \boldsymbol{\theta}_t^2)$.

**Stream-based Algorithm**. Our proposed stream-based active learning algorithm is described in Algorithm 1. In a round, when receiving a data instance, we calculate its exploitation-exploration

---

**Algorithm 2** NEURONAL-P

---

**Input:** $Q, B, K, \mathbf{f}_1, \mathbf{f}_2, \eta_1, \eta_2, \mu, \gamma$

1: Initialize $\boldsymbol{\theta}_1^1, \boldsymbol{\theta}_1^2$
2: **for** $t = 1, 2, \ldots, Q$ **do**
3:     Draw $B$ instances, $\mathbf{P}_t$, from $\mathcal{D}_{\mathcal{X}}$
4:     **for** $\mathbf{x}_i \in \mathbf{P}_t$ **do**
5:         $\mathbf{f}(\mathbf{x}_i; \boldsymbol{\theta}_t) = \mathbf{f}_1(\mathbf{x}_i; \boldsymbol{\theta}_t^1) + \mathbf{f}_2(\phi(\mathbf{x}_i); \boldsymbol{\theta}_t^2)$
6:         $\widehat{k} = \arg\min_{k \in [K]} \mathbf{f}(\mathbf{x}_i; \boldsymbol{\theta}_t)[k]$
7:         $k^\circ = \arg\min_{k \in ([K] \setminus \{\,\widehat{k}\,\})} \mathbf{f}(\mathbf{x}_i; \boldsymbol{\theta}_t)[k]$
8:         $w_i = \mathbf{f}(\mathbf{x}_i; \boldsymbol{\theta}_t)[\widehat{k}] - \mathbf{f}(\mathbf{x}_i; \boldsymbol{\theta}_t)[k^\circ]$
9:     **end for**
10:     $w_{\widehat{i}} = \min_{i \in [B]} w_i$
11:     For each $i \neq \widehat{i}$, $p_i = \frac{w_{\widehat{i}}}{\mu w_{\widehat{i}} + \gamma(w_i - w_{\widehat{i}})}$
12:     $p_{\widehat{i}} = 1 - \sum_{i = \widehat{i}} p_i$
13:     Draw one instance $\mathbf{x}_t$ from $\mathbf{P}_t$ according to probability distribution $P$ formed by $p_i$
14:     Query $\mathbf{x}_t$, Predict $\mathbf{y}_{t,\widehat{k}}$, and observe $\mathbf{y}_t$
15:     Update $\boldsymbol{\theta}^1, \boldsymbol{\theta}^2$ as Lines 11-12 in Algorithm 1
16: **end for**
17: **return** $\boldsymbol{\theta}_Q$

---

score and then make a prediction (Lines 3-7), where $k^\circ$ is used in the decision-maker (Line 8). When $\mathbf{I}_t = 1$, which indicates that the uncertainty of this instance is high, we query the label of this data instance to attain new information; otherwise, we treat our prediction as the pseudo label (Line 9). Finally, we use the SGD to train and update the parameters of NN models (Lines 10-13). Here, we draw the parameters from the pool $\Omega_t$ for the sake of analysis to bound the expected approximation error of one round. One can use the mini-batch SGD to avoid this issue in implementation.

**Pool-based Algorithm**. Our proposed pool-based active learning algorithm is described in Algorithm 2. In each round, we calculate the exploitation and exploration scores, and then calculate the prediction gap $w$ for each data instance (Lines 4-9). Then, we form a distribution of data instances (Lines 10-13), $P$, where Lines 11-12 are inspired by the selection scheme in [2]. The intuition behind this is as follows. The prediction gap $w_i$ reflects the uncertainty of this data instance. Smaller $w_i$ shows the larger uncertainty of $\mathbf{x}_i$. Thus, the smaller $w_i$, the higher the drawing probability $p_i$ (Line 11). Finally, we draw a data instance according to the sampling probability $P$ and conduct SGD to update the neural networks.

## 5 REGRET ANALYSIS

In this section, we provide the regret analysis for both the proposed stream-based and pool-based active learning algorithms.

Existing works such as [58; 42; 70] studied active learning in binary classification, where the label space $\mathcal{Y} \in [0, 1]$ can be parametrized by the Mammen-Tsybakov low noise condition [43]: There exist absolute constants $c > 0$ and $\alpha \geq 0$, such that for all $0 < \epsilon < 1/2, \mathbf{x} \in \mathcal{X}, k \in \{0, 1\}$, $\mathbb{P}(|\mathbf{h}(\mathbf{x})[k] - \frac{1}{2}| \leq \epsilon) \leq c\epsilon^\alpha$. For simplicity, we are interested in the two extremes: $\alpha = 0$ results in no assumption whatsoever on $\mathcal{D}$ while $\alpha = \infty$ gives the hard-margin condition on $\mathcal{D}$. These two conditions can be directly extended to the $K$-class classification task. Next, we will provide the regret analysis and comparison with [58].

Our analysis is associated with the NTK matrix $\mathbf{H}$ defined in C.1. There are two complexity terms in [58] as well as in Neural Bandits [68; 12]. The assumption $\mathbf{H} \succeq \lambda_0 \mathbf{I}$ is generally held in this literature to guarantee that there exists a solution for NTK regression. The first complexity term is $S = \sqrt{\mathbf{h}^\top \mathbf{H}^{-1} \mathbf{h}}$ where $\mathbf{h} = [\mathbf{h}(\mathbf{x}_1)[1], \mathbf{h}(\mathbf{x}_1)[2], \ldots, \mathbf{h}(\mathbf{x}_T)[K]]^\top \in \mathbb{R}^{TK}$. $S$ is to bound the optimal parameters in NTK regression: there exists $\boldsymbol{\theta}^* \in \mathbb{R}^p$ such that $\mathbf{h}(\mathbf{x}_t)[k] = \langle \nabla_{\boldsymbol{\theta}} \mathbf{f}(\mathbf{x}_t; \boldsymbol{\theta}^*)[k], \boldsymbol{\theta}^* - \boldsymbol{\theta}_1 \rangle$ and $\|\boldsymbol{\theta}^* - \boldsymbol{\theta}_1\|_2 \leq S$. The second complexity term is the effective dimension $\tilde{d}$, defined as $\tilde{d} =$

$\frac{\log\det(\mathbf{I}+\mathbf{H})}{\log(1+TK)}$, which describes the actual underlying dimension in the RKHS space spanned by NTK. [58] used the term $L_{\mathbf{H}}$ to represent $\tilde{d}$: $L_{\mathbf{H}} = \log\det(\mathbf{I}+\mathbf{H}) = \tilde{d}\log(1+TK)$. We provide the upper bound and lower bound of $L_{\mathbf{H}}$ in Appendix C: $TK\log(1+\lambda_0) \leq L_{\mathbf{H}} \leq \widetilde{\mathcal{O}}(mdK)$.

First, we present the regret and label complexity analysis for Algorithm 1 in binary classification, which is directly comparable to [58].

**Theorem 5.1.** *[Binary Classification] Given $T$, for any $\delta \in (0,1)$, $\lambda_0 > 0$, suppose $K = 2$, $\|\mathbf{x}_t\|_2 = 1, t \in [T]$, $\mathbf{H} \succeq \lambda_0\mathbf{I}$, $m \geq \widetilde{\Omega}(poly(T,L,S)\cdot\log(1/\delta)), \eta_1 = \eta_2 = \Theta(\frac{S}{m\sqrt{2T}})$. Then, with probability at least $1-\delta$ over the initialization of $\boldsymbol{\theta}_1^1, \boldsymbol{\theta}_1^2$, Algorithm 1 achieves the following regret bound:*

$$\mathbf{R}_{stream}(T) \leq \widetilde{\mathcal{O}}((S^2)^{\frac{\alpha+1}{\alpha+2}}T^{\frac{1}{\alpha+2}}),$$
$$\mathbf{N}(T) \leq \widetilde{\mathcal{O}}((S^2)^{\frac{\alpha}{\alpha+2}}T^{\frac{2}{\alpha+2}}).$$

Compared to [58], Theorem 5.1 removes the term $\widetilde{\mathcal{O}}\left(L_{\mathbf{H}}^{\frac{2(\alpha+1)}{\alpha+2}}T^{\frac{1}{\alpha+2}}\right)$ and further improve the regret upper bound by a multiplicative factor $L_{\mathbf{H}}^{\frac{\alpha+1}{\alpha+2}}$. For the label complexity $\mathbf{N}(T)$, compared to [58], Theorem 5.1 removes the term $\widetilde{\mathcal{O}}\left(L_{\mathbf{H}}^{\frac{2\alpha}{\alpha+2}}T^{\frac{2}{\alpha+2}}\right)$ and further improve the regret upper bound by a multiplicative factor $L_{\mathbf{H}}^{\frac{\alpha}{\alpha+2}}$.

Next, we show the regret bound for the stream-based active learning algorithm (Algorithm 1) in $K$-classification, without any assumption on the distribution $\mathcal{D}$ (Tsybakov noise $\alpha = 0$).

**Theorem 5.2.** *[Stream-based]. Given $T$, for any $\delta \in (0,1)$, $\lambda_0 > 0$, suppose $\|\mathbf{x}_t\|_2 = 1, t \in [T]$, $\mathbf{H} \succeq \lambda_0\mathbf{I}$, $m \geq \widetilde{\Omega}(poly(T,K,L,S)\cdot\log(1/\delta)), \eta_1 = \eta_2 = \Theta(\frac{S}{m\sqrt{TK}})$. Then, with probability at least $1-\delta$ over the initialization of $\boldsymbol{\theta}_1^1, \boldsymbol{\theta}_1^2$, Algorithm 1 achieves the following regret bound:*

$$\mathbf{R}_{stream}(T) \leq \mathcal{O}(\sqrt{T})\cdot\left(\sqrt{K}S + \sqrt{2\log(3T/\delta)}\right)$$

*where $\mathbf{N}(T) \leq \mathcal{O}(T)$.*

Theorem 5.2 shows that NEURONAL-S achieves a regret upper bound of $\widetilde{\mathcal{O}}(\sqrt{TK}S)$. Under the same condition ($\alpha = 0$), [58] obtains the regret upper bound: $\mathbf{R}_{stream}(T) \leq \widetilde{\mathcal{O}}\left(\sqrt{TL_{\mathbf{H}}}S + \sqrt{T}L_{\mathbf{H}}\right)$. This core term is further bounded by $\widetilde{\mathcal{O}}(\sqrt{T}\cdot\sqrt{TK\log(1+\lambda_0)}S) \leq \widetilde{\mathcal{O}}\left(\sqrt{TL_{\mathbf{H}}}S\right) \leq \widetilde{\mathcal{O}}(\sqrt{TmdK}S)$. Concerning $K$, Theorem 5.2 results in regret-growth rate $\sqrt{K}\cdot\widetilde{\mathcal{O}}(\sqrt{T}S)$ in contrast with $\sqrt{K}\cdot\widetilde{\mathcal{O}}(\sqrt{Tmd}S)$ at most in [58]. Therefore, our regret bound has a slower growth rate concerning $K$ compared to [58] by a multiplicative factor at least $\mathcal{O}(\sqrt{T\log(1+\lambda_0)})$ and up to $\widetilde{\mathcal{O}}(\sqrt{md})$. In binary classification, we reduce the regret bound by $\widetilde{\mathcal{O}}(\sqrt{T}L_{\mathbf{H}})$, i.e., remove the dependency of $\tilde{d}$, and further improve the regret bound by a multiplicative factor at least $\mathcal{O}(\sqrt{T\log(1+\lambda_0)})$.

The regret analysis of [14] introduced two other forms of complexity terms: $\widetilde{\mathcal{O}}(\sqrt{T}\cdot(\sqrt{\mu}+\nu))$. $\mu$ is the data-dependent term interpreted as the minimal training error of a function class on the data, while $\nu$ is the function class radius. [3] had implied that when $\nu$ has the order of $\widetilde{\mathcal{O}}(poly(T))$, $\mu$ can decrease to $\widetilde{\mathcal{O}}(1)$. But, their regret bounds also depend on $\mathcal{O}(\nu)$. This indicates that their regret analysis is invalid when $\nu$ has $\mathcal{O}(T)$. Since [14] did not provide the upper bound and trade-off of $\mu$ and $\nu$ or build the connection with NTK, their results are not readily comparable to ours.

For the label complexity, Theorem 5.2 has the trivial $\mathcal{O}(T)$ complexity which is the same as [58; 14]. This turns into the active learning minimax rate of $\mathbf{N}(T)^{-1/2}$, which is indeed the best rate under $\alpha = 0$ [16; 30; 36; 21]. Next, we provide the analysis under the following margin assumption.

**Assumption 5.1** ([14]). Given an instance $\mathbf{x}_t$ and the label $\mathbf{y}_t$, $\mathbf{x}_t$ has a unique optimal class if there exists $\epsilon > 0$ such that

$$\forall t \in [T], \mathbf{h}(\mathbf{x}_t)[k^*] - \mathbf{h}(\mathbf{x}_t)[k^\circ] \geq \epsilon, \tag{5.1}$$

where $k^* = \arg\min_{k\in[K]}\mathbf{h}(\mathbf{x}_t)[k]$ and $k^\circ = \arg\min_{k\in([K]\setminus\{k^*\})}\mathbf{h}(\mathbf{x}_t)[k]$. $\qquad\square$

Assumption 5.1 describes that there exists a unique Bayes-optimal class for each input instance. Then, we have the following theorem.

**Theorem 5.3.** *[Stream-based].* *Given $T$, for any $\delta \in (0,1), \gamma > 1, \lambda_0 > 0$, suppose $\|\mathbf{x}_t\|_2 = 1, t \in [T]$, $\mathbf{H} \succeq \lambda_0 \mathbf{I}$, $m \geq \widetilde{\Omega}(poly(T, K, L, S) \cdot \log(1/\delta)), \eta_1 = \eta_2 = \Theta(\frac{S}{m\sqrt{TK}})$, and Assumption 5.1 holds. Then, with probability at least $1 - \delta$ over the initialization of $\boldsymbol{\theta}_1^1, \boldsymbol{\theta}_1^2$, Algorithm 1 achieves the following regret bound and label complexity:*

$$\mathbf{R}_{stream}(T) \leq \mathcal{O}((KS^2 + \log(3T/\delta))/\epsilon), \ \mathbf{N}(T) \leq \mathcal{O}((KS^2 + \log(3T/\delta))/\epsilon^2). \tag{5.2}$$

Theorem 5.3 provides the regret upper bound and label complexity of $\widetilde{\mathcal{O}}(KS^2)$ for NEURONAL-S under margin assumption. With $\alpha \to \infty$, [58] obtained $\widetilde{\mathcal{O}}(L_{\mathbf{H}}(L_{\mathbf{H}} + S^2)) \leq \widetilde{\mathcal{O}}(mdK(mdK + S^2))$. Therefore, with margin assumption, the regret of NEURONAL-S grows slower than [58] by a multiplicative factor up to $\mathcal{O}(md)$ with respect to $K$. [14] achieved $\widetilde{\mathcal{O}}(\nu^2 + \mu)$, but the results are not directly comparable to ours, as discussed before.

**Theorem 5.4** (Pool-based). *For any $1 > \delta > 0$, $\lambda_0 > 0$, by setting $\mu = Q$ and $\gamma = \sqrt{Q/(KS^2)}$, suppose $\|\mathbf{x}_t\|_2 = 1$ for all instances, $\mathbf{H} \succeq \lambda_0 \mathbf{I}$, $m \geq \widetilde{\Omega}(poly(Q, K, L, R) \cdot \log(1/\delta)), \eta_1 = \eta_2 = \Theta(\frac{1}{mLK})$. Then, with probability at least $1 - \delta$ over the initilization of $\boldsymbol{\theta}_1^1, \boldsymbol{\theta}_1^2$, Algorithm 2 achieves:*

$$\mathbf{R}_{pool}(Q) \leq \mathcal{O}(\sqrt{QK}S) + \mathcal{O}\left(\sqrt{\frac{Q}{KS^2}}\right) \cdot \log(2\delta^{-1}) + \mathcal{O}(\sqrt{Q\log(3\delta^{-1})}).$$

Theorem 5.4 provides a performance guarantee of $\widetilde{\mathcal{O}}(\sqrt{QK}S)$ for NEURONAL-P in the non-parametric setting with neural network approximator. This result shows that the pool-based algorithm can achieve a regret bound of the same complexity as the stream-based algorithm (Theorem 5.2) with respect to the number of labels. Meanwhile, Theorem 5.4 indicates the $\widetilde{O}(\sqrt{1/Q})$ minimax rate in the pool-based setting which matches the rate ($\alpha = 0$) in [8; 64; 28]. However, the results in [8; 64; 28] only work with the linear separator, i.e., they assume $\mathbf{h}$ as the linear function with respect to the data instance $\mathbf{x}$. Note that [58; 14] only work on stream-based active learning.

## 6 EXPERIMENTS

In this section, we evaluate NEURONAL for both stream-based and pool-based settings on the following six public classification datasets: Adult, Covertype (CT), MagicTelescope (MT), Shuttle [24], Fashion [61], and Letter [18]. For all NN models, we use the same width $m = 100$ and depth $L = 2$. Due to the space limit, we move additional results (figures) to Appendix A.

**Stream-based Setups.** We use two metrics to evaluate the performance of each method: (1) test accuracy and (2) cumulative regret. We set $T = 10,000$. After $T$ rounds of active learning, we evaluate all the methods on another $10,000$ unseen data points and report the accuracy. As MT does not have enough data points, we use $5,000$ data points for the evaluation. In each iteration, one instance is randomly chosen and the model predicts a label. If the predicted label is wrong, the regret is 1; otherwise 0. The algorithm can choose to observe the label, which will reduce the query budget by one. We restrict the query budget to avoid the situation of the algorithm querying every data point in the dataset, following [14]. The default label budget is $30\% \times T$. We ran each experiment 10 times and reported the mean and std of results. We use three baselines for comparison in the stream-based setting: (1) NeurAL-NTK [58], (2) I-NeurAL [14], and (3) ALPS [22]. Due to the space limit, we reported the cumulative regret and more details in Appendix A.1.

**Stream-based Results.** To evaluate the effectiveness of NEURONAL-S, first, we report the test accuracy of bandit-based methods in Table 1, which shows the generalization performance of each method. From this table, we can see that the proposed NEURONAL-S consistently trains the best model compared to all the baselines for each dataset, where NEURONAL-S achieves non-trivial improvements under the same label budget with the same width and depth of NN models. Compared to bandit-based approaches, NeurAL-NTK and I-NeurAL, NEURONAL-S has new input and output, which enable us to better exploit the full feedback in active learning instead of bandit feedback and to explore more effectively. The results of ALPS are placed in Table 3. ALPS manages to select the best pre-trained hypotheses for the data. However, the model parameters are fixed before the online active learning process. Hence, ALPS is not able to take the new knowledge obtained by queries into account and its performance is highly limited by the hypothesis class. Table 1 shows the running time

Table 2: Testing accuracy (%) and running time on all methods in **Pool-based** Setting

| | Adult | MT | Letter | Covertype | Shuttle | Fashion |
|---|---|---|---|---|---|---|
| | Accuracy | | | | | |
| CoreSet | $76.7 \pm 1.13$ | $75.1 \pm 0.79$ | $80.6 \pm 0.63$ | $62.6 \pm 3.11$ | $97.7 \pm 0.41$ | $80.4 \pm 0.08$ |
| BADGE | $76.6 \pm 0.49$ | $71.6 \pm 0.81$ | $81.7 \pm 0.57$ | $64.8 \pm 1.02$ | $98.6 \pm 0.39$ | $76.1 \pm 0.21$ |
| DynamicAL | $72.4 \pm 0.14$ | $67.8 \pm 1.01$ | $63.2 \pm 0.31$ | $54.1 \pm 0.12$ | $78.7 \pm 0.05$ | $54.5 \pm 0.19$ |
| ALBL | $78.1 \pm 0.45$ | $73.9 \pm 0.71$ | $81.9 \pm 0.47$ | $65.3 \pm 0.14$ | $98.6 \pm 0.37$ | $77.6 \pm 0.32$ |
| NEURONAL-P | $\mathbf{79.1 \pm 0.04}$ | $\mathbf{81.3 \pm 0.12}$ | $\mathbf{83.7 \pm 0.07}$ | $\mathbf{67.6 \pm 0.21}$ | $\mathbf{99.5 \pm 0.01}$ | $\mathbf{81.1 \pm 0.13}$ |
| | Running Time | | | | | |
| CoreSet | $43.1 \pm 7.65$ | $119.3 \pm 4.42$ | $228.3 \pm 6.51$ | $32.5 \pm 10.94$ | $10.9 \pm 2.22$ | $33.1 \pm 6.32$ |
| BADGE | $159.5 \pm 4.61$ | $212.5 \pm 9.32$ | $484.8 \pm 7.04$ | $545.7 \pm 9.32$ | $222.9 \pm 5.13$ | $437.8 \pm 5.32$ |
| DynamicAL | $24.3 \pm 5.21$ | $821.5 \pm 6.14$ | $382.3 \pm 3.13$ | $621.6 \pm 3.21$ | $483.4 \pm 9.78$ | $413.2 \pm 7.14$ |
| ALBL | $315.8 \pm 4.31$ | $343.5 \pm 6.24$ | $271.3 \pm 6.32$ | $481.3 \pm 5.21$ | $63.2 \pm 2.16$ | $92.1 \pm 3.42$ |
| NEURONAL-P | $\mathbf{17.2 \pm 3.24}$ | $140.1 \pm 3.69$ | $\mathbf{133.7 \pm 12.8}$ | $\mathbf{14.1 \pm 5.81}$ | $15.6 \pm 8.03$ | $\mathbf{25.5 \pm 7.80}$ |

of NEURONAL-S compared to bandit-based approaches. NEURONAL-S achieves the best empirical performance and significantly saves the time cost. The speed-up is boosted when $K$ is large. This is because of NEURONAL-S's new NN model structure to calculate the predicted score synchronously. In contrast, bandit-based approaches calculate the score sequentially, of which the computational cost is scaled by $K$. We also conduct the ablation study for different label budgets in the active learning setting placed in Appendix A.

**Pool-based Setups.** We use the test accuracy to evaluate the performance of each method. Following [57], we use batch active learning. In each active learning round, we select and query 100 points for labels. After 10 rounds of active learning, we evaluate all the methods on another $10,000$ unseen data points and report the accuracy. We run each experiment 10 times and report the mean and standard deviation of results. The four SOTA baselines are: (1) CoreSet [52], (2) BADGE [6], (3) DynamicAL [57], and (4) ALBL [33].

**Pool-based results**. To evaluate the effectiveness of NEURONAL-P, we report the test accuracy of all methods in Table 2. Our method, NEURONAL-P, can consistently outperform other baselines across all datasets. This indicates that these baselines without explicit exploration are easier to be trapped in sub-optimal solutions. Our method has a more effective network structure (including the principled exploration network), allowing us to exploit the full feedback and explore new knowledge simultaneously in active learning. Moreover, we use the inverse gap strategy to draw samples, which further balances exploitation and exploration. CoreSet always chooses the maximal distance based on the embeddings derived by the last layer of the exploitation neural network, which is prone to be affected by outlier samples. BADGE also works on the last layer of the exploitation network using the seed-based clustering method, and it is not adaptive to the state of the exploitation network. DynamicAL relies on the training dynamics of the Neural Tangent Kernel, but the rules it uses might only work on the over-parameterized neural networks based on the analysis. ALBL is a hybrid active learning algorithm that combines Coreset and Conf [55]. ALBL shows a stronger performance but still is outperformed by our algorithm. As mentioned before, these baselines do not provide a theoretical performance guarantee. For the running time, as there are $B$ samples in a pool in each round, NEURONAL-P takes $\mathcal{O}(2B)$ to make a selection. For other baselines, CoreSet takes $\mathcal{O}(2B)$. In addition to $\mathcal{O}(2B)$ cost, BADGE and DynamicAL need to calculate the gradient for each sample, which is scaled by $\mathcal{O}(B)$. Thus, BADGE and DynamicAL are slow. ALBA is slow because it contains another algorithm, and thus, it has two computational costs in addition to neural models. We also conduct the hyper-parameter sensitivity study for different label budgets in the active learning setting placed in Appendix A.2.

## 7 CONCLUSION

In this paper, we propose two algorithms for both stream-based and pool-based active learning. The proposed algorithms mitigate the adverse effects of $K$ in terms of computational cost and performance. The proposed algorithms build on the newly designed exploitation and exploration neural networks, which enjoy a tighter provable performance guarantee in the non-parametric setting. Ultimately, we use extensive experiments to demonstrate the improved empirical performance in stream- and pool-based settings.

ACKNOWLEDGEMENT

This work is supported by National Science Foundation under Award No. IIS-2117902, IIS-2002540, IIS-2134079, Agriculture and Food Research Initiative (AFRI) grant no. 2020-67021-32799/project accession no.1024178 from the USDA National Institute of Food and Agriculture, MIT-IBM Watson AI Lab, and IBM-Illinois Discovery Accelerator Institute - a new model of an academic-industry partnership designed to increase access to technology education and skill development to spur breakthroughs in emerging areas of technology. The views and conclusions are those of the authors and should not be interpreted as representing the official policies of the funding agencies or the government.

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

## A  Experiments Details

In this section, we provide more experiments and details for both stream-based and pool-based settings.

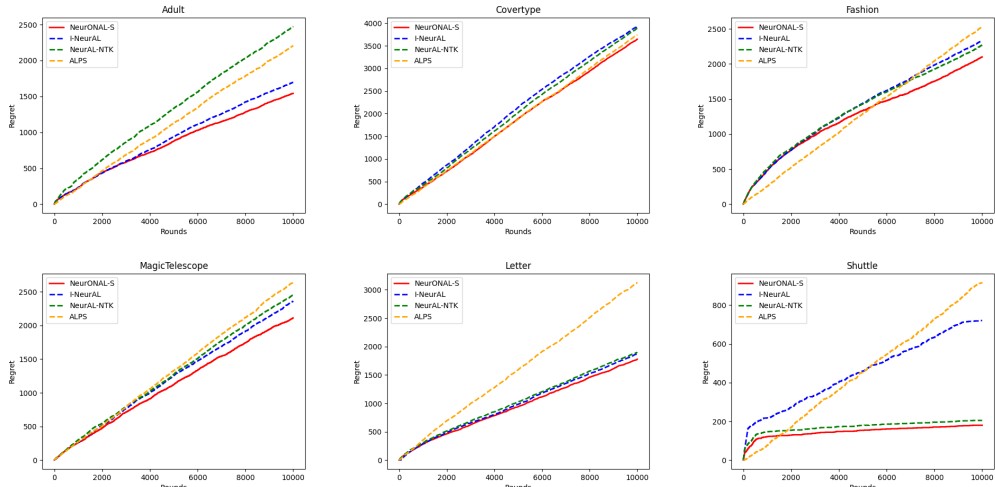

Figure 1: Regret comparison on six datasets in the **stream-based** setting. NEURONAL-S outperforms baselines on most datasets.

### A.1  Stream-Based

Table 3: Test accuracy of all methods in stream-based setting

|  | Adult | MT | Letter | Covertype | Shuttle | Fashion |
|---|---|---|---|---|---|---|
| NeurAL-NTK [58] | $80.1 \pm 0.06$ | $76.9 \pm 0.1$ | $79.6 \pm 0.23$ | $62.1 \pm 0.02$ | $96.2 \pm 0.21$ | $64.3 \pm 0.13$ |
| I-NeurAL | $84.1 \pm 0.29$ | $79.9 \pm 0.14$ | $82.9 \pm 0.04$ | $65.2 \pm 0.16$ | $99.3 \pm 0.07$ | $73.6 \pm 0.29$ |
| ALPS | $75.6 \pm 0.19$ | $35.9 \pm 0.76$ | $73.0 \pm 0.41$ | $36.2 \pm 0.25$ | $78.5 \pm 0.21$ | $74.3 \pm 0.01$ |
| NEURONAL-S | $\mathbf{84.7 \pm 0.32}$ | $\mathbf{83.7 \pm 0.16}$ | $\mathbf{86.8 \pm 0.12}$ | $\mathbf{74.5 \pm 0.15}$ | $\mathbf{99.7 \pm 0.02}$ | $\mathbf{82.2 \pm 0.18}$ |

**Baselines**. Given an instance, NeurAL-NTK is a method that predicts $K$ scores for $K$ classes sequentially, only based on the exploitation NN classifier with an Upper-Confidence-Bound (UCB). On the contrary, I-NeurAL predicts $K$ scores for $K$ classes sequentially, based on both the exploitation and exploration NN classifiers. NeurAL-NTK and I-NeurAL query for a label when the model cannot differentiate the Bayes-optimal class from other classes. Finally, ALPS makes a prediction by choosing a hypothesis (from a set of pre-trained hypotheses) that minimizes the loss of labeled and pseudo-labeled data, and queries based on the difference between hypotheses.

**Implementation Details.** We perform hyperparameter tuning on the training set. Each method has a couple of hyperparameters: the learning rate, number of epochs, batch size, label budget percentage, and threshold (if applicable). During hyperparameter tuning for all methods, we perform a grid search over the values $\{0.0001, 0.0005, 0.001\}$ for the learning rate, $\{10, 20, 30, 40, 50, 60, 70, 80, 90\}$ for the number of epochs, $\{32, 64, 128, 256\}$ for the batch size, $\{0.1, 0.3, 0.5, 0.7, 0.9\}$ for the label budget percentage and $\{1, 2, 3, 4, 5, 6, 7, 8, 9\}$ for the threshold (exploration) parameter. Here are the final hyperparameters in the form {learning rate, number of epochs, batch size, label budget percentage, and the turning hyperparameter}: NeurAL-NTK and ALPS use $\{0.001, 40, 64, 0.3, 3\}$, and I-NeurAL uses $\{0.001, 40, 32, 0.3, 6\}$. For NEURONAL-S, we set $\mu = 1$ for all experiments and conduct the grid search for $\gamma$ over $\{1, 2, 3, 4, 5, 6, 7, 8, 9\}$. In the end, NEURONAL-S uses $\{0.0001, 40, 64, 0.3, 6\}$ for all datasets. We set $S = 1$ as the norm parameter for NEURONAL-S all the time.

**Cumulative Regret**. Figure 1 shows the regret comparison for each of the six datasets in 10,000 rounds of stream-based active learning. NEURONAL-S outperforms baselines on most datasets. This

demonstrates that our designed NN model attains effectiveness in the active learning process, which is consistent with the best performance achieved by NEURONAL-S in testing accuracy.

**Ablation study for label budget**. Tables 4 to 6 show the NEURONAL-S in active learning with different budget percentages: 3%, 10%, 50 %. NEURONAL-S achieves the best performance in most of the experiments. With 3% label budget, almost all NN models are not well trained. Thus, NEURONAL does not perform stably. With 10% and 50% label budget, NEURONAL achieves better performance, because the advantages of NN models can better exploit and explore this label information.

Table 4: Test accuracy with 3% budget in stream-based setting

|  | Adult | Covertype | Fashion | MagicTelescope | Letter | Shuttle |
|---|---|---|---|---|---|---|
| I-NeurAL | 79.4% | 52.8% | 51.9% | 72.3% | 74.6% | 93.0% |
| NeurAL-NTK | 23.9% | 1.56% | 11.9% | 32.9% | 42.8% | 70.6% |
| ALPS | 24.2% | 36.8% | 10.0% | 64.9% | 72.7% | 79.4% |
| NEURONAL-S | **79.9%** | **65.6%** | 69.7% | **77.3%** | 74.2% | **99.8%** |

Table 5: Test accuracy with 10% budget in stream-based setting

|  | Adult | Covertype | Fashion | MagicTelescope | Letter | Shuttle |
|---|---|---|---|---|---|---|
| I-NeurAL | 80.5% | 55.4% | 71.4% | 77.9% | 81.8% | 99.2% |
| NeurAL-NTK | 70.5% | 59.9% | 38.7% | 34.3% | 53.8% | 75.9% |
| ALPS | 24.2% | 36.8% | 10.0% | 35.1% | 79.9% | 79.4% |
| NEURONAL-S | **79.5%** | **71.3%** | 81.3% | **82.1%** | 81.8% | **99.8%** |

Table 6: Test accuracy with 50% budget in stream-based setting

|  | Adult | Covertype | Fashion | MagicTelescope | Letter | Shuttle |
|---|---|---|---|---|---|---|
| I-NeurAL | 83.4% | 65.9% | 82.5% | 77.9% | 85.8% | 99.7% |
| NeurAL-NTK | 76.9% | 73.1% | 56.8% | 81.6% | 79.3% | 97.1% |
| ALPS | 75.8% | 36.8% | 10.0% | 64.9% | 81.5% | 79.4% |
| NEURONAL-S | **84.6%** | **75.9%** | **85.4%** | **86.4%** | **86.9%** | **99.8%** |

## A.2 POOL-BASED

**Baselines**. BADGE uses gradient embeddings to model uncertainty - if the gradient in the last neural network layer is large, the uncertainty is also large. They pick random points to query using the k-meanss++ algorithm and repeat this process. DynamicAL introduces the concept of training dynamics into active learning. Given an instance, they assign it a pseudo-label and monitor how much the model changes. They query for the label of the point with the biggest change in the training dynamics. CoreSet has a simple but powerful approach. The algorithm chooses points to query based on the loss over the labeled points and the loss over the unlabelled points, i.e., the core-set loss. ALBL is a hybrid active learning algorithm that combines Coreset and Conf [55].

**Implementation details**. For all methods, we conduct the same grid search as the stream-based setting for the learning rate and number of epochs. For NEURONAL-P, we perform a grid search over $\mu, \gamma \in \{500, 1000, 2000\}$.

**Testing accuracy**. Figure 2 shows the average test accuracy at each round. NEURONAL-P can outperform baselines in a few query rounds. Our method utilizes a highly effective network structure, including the principled exploration network and inverse-weight-based selection strategy. Unlike CoreSet, which solely relies on the embeddings derived from the last layer of exploitation neural networks to select samples based on maximum distance, our approach avoids susceptibility to outlier samples. Similarly, BADGE also operates on the last layer of the exploitation network using the seed-based clustering method, lacking adaptability to the state of the exploitation network. DynamicAL's approach relies on the training dynamics of the Neural Tangent Kernel that usually requires very wide neural networks. ALBL is a blending approach, but it still suffers from the limitation of CoreSet.

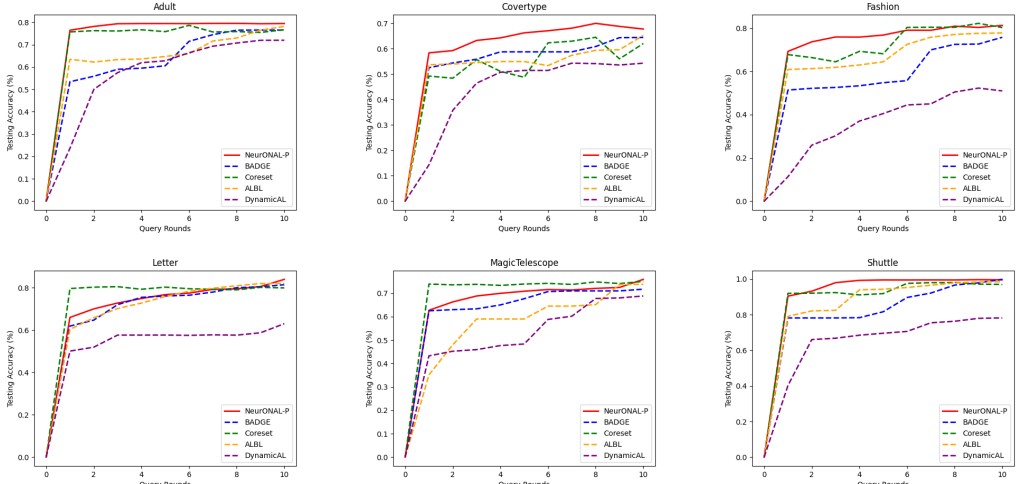

Figure 2: Test accuracy versus the number of query rounds in **pool-based** setting on six datasets. NEURONAL-P outperforms baselines on all datasets.

**Ablation study for $\mu$ and $\gamma$.** Table 7 shows NEURONAL-P with varying $\mu$ and $\gamma$ values (500, 1000, 2000) on four datasets. Intuitively, if $\gamma$ and $\mu$ is too small, NEURONAL-P will place more weights on the tail of the distribution of $P$. Otherwise, NEURONAL-P will focus more on the head of the distribution of $P$. From the results in the table, it seems that different datasets respond to different values of $\mu$ and $\gamma$. This sensitivity study roughly shows good values for $\mu, \gamma$.

Table 7: Testing Accuracy on four datasets (Letter, Adult, Fashion, and MT) with varying $\mu$ and $\gamma$ in pool-based setting

Letter

| | | $\gamma$ | | |
|---|---|---|---|---|
| | | 500 | 1000 | 2000 |
| | 500 | 80.9% | 81.7% | 80.5% |
| $\mu$ | 1000 | 77.9% | **83.9%** | 78.9% |
| | 2000 | 81.7% | 81.8% | 80.1% |

Adult

| | | $\gamma$ | | |
|---|---|---|---|---|
| | | 500 | 1000 | 2000 |
| | 500 | **79.9%** | 79.4% | 78.9% |
| $\mu$ | 1000 | 79.1% | 79.7% | 79.0% |
| | 2000 | 79.4% | 79.4% | 79.7% |

Fashion

| | | $\gamma$ | | |
|---|---|---|---|---|
| | | 500 | 1000 | 2000 |
| | 500 | 80.3% | 80.5% | 79.5% |
| $\mu$ | 1000 | 80.5% | 80.6% | 80.4% |
| | 2000 | 80.8% | **80.9%** | 80.7% |

MT

| | | $\gamma$ | | |
|---|---|---|---|---|
| | | 500 | 1000 | 2000 |
| | 500 | 79.5% | 80.9% | 80.6% |
| $\mu$ | 1000 | 80.2% | 80.9% | 80.1% |
| | 2000 | 80.5% | 80.6% | **81.3%** |

## B STREAM-BASED VS POOL-BASED

To answer the question "Can one directly convert the stream-based algorithm to the pool-based setting?", we implemented the idea that one uniformly samples data from the pool to feed it to the stream-based active learner. Denote the new algorithm by Neu-UniS, described as follows.

**Neu-UniS**: In a round of pool-based active learning, we uniformly draw a sample from the pool and feed it to the stream-based learner. If the stream-based learner decides to query the label, it costs one unit of the label budget; otherwise, we keep uniformly drawing a sample until the stream-based learner decides to query the label. Once the label is queried, we train the neural model based on the sample and label. We keep doing this process until we run out of the label budget. In this algorithm, the stream-based learner is set as NEURONAL-S (Algorithm 1 in the manuscript).

Under the same setting used in our pool-based experiments, the testing accuracy of Neu-UniS compared to our pool-based algorithm NEURONAL-P is reported in Table 8.

Table 8: Test Accuracy of Neu-UniS and NEURONAL-P

|            | Adult | Covertype | Fashion | MT   | Letter | Shuttle |
|------------|-------|-----------|---------|------|--------|---------|
| Neu-UniS   | 78.0  | 59.4      | 78.6    | 71.3 | 79.4   | 97.9    |
| NEURONAL-P | **79.9** | **66.9** | **80.9** | **81.3** | **83.9** | **99.6** |

Why does NEURONAL-P outperform Neu-UniS? Because Neu-UniS does not rank data instances and only randomly chooses the data instances that satisfy the query criterion. All the stream-based algorithms have one criterion to determine whether to query the label for this data point, such as Lines 8-9 in Algorithm 1. Suppose there are 200 data points. If the 100 data points among them satisfy the criterion, then Neu-UniS will randomly choose one from the 100 data points, because we uniformly draw a data point and feed it to stream-based learner in each round.

On the contrary, NEURONAL-P has a novel component (Lines 10-12 in Algorithm 2) to rank all the data points, and then draw a sample from the newly formed distribution, to balance exploitation and exploration. To the best of our knowledge, this is the first inverse-gap weighting strategy in active learning. Thus, its analysis is also novel.

In summary, stream-based algorithms cannot directly convert into pool-based algorithms, because they do not have the ranking component which is necessary in the pool-based setting. Existing works [58; 22; 14] only focus on the stream-based setting and [52] [6] [57] only focus on pool-based setting. We could hardly find existing works that incorporate both stream-based and pool-based settings.

## C   UPPER BOUND AND LOWER BOUND FOR $L_{\mathbf{H}}$

**Definition C.1** (NTK [34; 58]). *Let $\mathcal{N}$ denote the normal distribution. Given the data instances $\{\mathbf{x}_t\}_{t=1}^T$, for all $i, j \in [T]$, define*

$$\mathbf{H}_{i,j}^0 = \Sigma_{i,j}^0 = \langle \mathbf{x}_i, \mathbf{x}_j \rangle, \quad \mathbf{A}_{i,j}^l = \begin{pmatrix} \Sigma_{i,i}^l & \Sigma_{i,j}^l \\ \Sigma_{j,i}^l & \Sigma_{j,j}^l \end{pmatrix}$$

$$\Sigma_{i,j}^l = 2\mathbb{E}_{a,b\sim\mathcal{N}(\mathbf{0},\mathbf{A}_{i,j}^{l-1})}[\sigma(a)\sigma(b)], \quad \mathbf{H}_{i,j}^l = 2\mathbf{H}_{i,j}^{l-1}\mathbb{E}_{a,b\sim\mathcal{N}(\mathbf{0},\mathbf{A}_{i,j}^{l-1})}[\sigma'(a)\sigma'(b)] + \Sigma_{i,j}^l.$$

*Then, the Neural Tangent Kernel matrix is defined as $\mathbf{H} = (\mathbf{H}^L + \Sigma^L)/2$.*

Then, we define the following gram matrix $\mathbf{G}$. Let $g(x; \boldsymbol{\theta}_0) = \nabla_{\boldsymbol{\theta}} f(x; \boldsymbol{\theta}_0) \in \mathbb{R}^p$ and $G = [g(\mathbf{x}_{1,1}; \boldsymbol{\theta}_0)/\sqrt{m}, \ldots, g(\mathbf{x}_{T,K}; \boldsymbol{\theta}_0)/\sqrt{m}] \in \mathbb{R}^{p \times TK}$ where $p = m + mKd + m^2(L-2)$. Therefore, we have $\mathbf{G} = G^\top G$. Based on Theorem 3.1 in [4], when $m \geq \Omega(T^4 K^6 \log(2TK/\delta)/\epsilon^4)$, with probability at least $1 - \delta$, we have

$$\|\mathbf{G} - \mathbf{H}\|_F \leq \epsilon.$$

Then, we have the following bound:

$$\begin{aligned}
\log \det(\mathbf{I} + \mathbf{H}) &= \log \det\left(\mathbf{I} + \mathbf{G} + (\mathbf{H} - \mathbf{G})\right) \\
&\overset{(e_1)}{\leq} \log \det(\mathbf{I} + \mathbf{G}) + \langle (\mathbf{I} + \mathbf{G})^{-1}, (\mathbf{H} - \mathbf{G}) \rangle \\
&\leq \log \det(\mathbf{I} + \mathbf{G}) + \|(\mathbf{I} + \mathbf{G})^{-1}\|_F \|\mathbf{H} - \mathbf{G}\|_F \\
&\overset{(e_2)}{\leq} \log \det(\mathbf{I} + \mathbf{G}) + \sqrt{T}\|\mathbf{H} - \mathbf{G}\|_F \\
&\overset{(e_3)}{\leq} \log \det(\mathbf{I} + \mathbf{G}) + 1
\end{aligned} \tag{C.1}$$

where $(e_1)$ is because of the concavity of $\log \det(\cdot)$, $(e_2)$ is by Lemma B.1 in [68] with the choice of $m$, and $(e_3)$ is by the proper choice of $\epsilon$. Then, $L_{\mathbf{H}}$ can be bounded by:

$$
\begin{aligned}
\log \det(\mathbf{I} + \mathbf{H}) &\leq \log \det(\mathbf{I} + \mathbf{G}) + 1 \\
&\stackrel{(e_1)}{=} \log \det(\mathbf{I} + GG^\top) + 1 \\
&= \log \det \left( \mathbf{I} + \sum_{i=1}^{TK} g(\mathbf{x}_i; \boldsymbol{\theta}_0) g(\mathbf{x}_i; \boldsymbol{\theta}_0)^\top / m \right) + 1 \\
&\stackrel{(e_2)}{\leq} p \cdot \log(1 + \mathcal{O}(TK)/p) + 1
\end{aligned}
$$

where $(e_1)$ is because of $\det(\mathbf{I} + G^\top G) = \det(\mathbf{I} + GG^\top)$ and $(e_2)$ is an application of Lemma 10 in [1] and $\|g(\mathbf{x}_i; \boldsymbol{\theta}_0)\|_2 \leq \mathcal{O}(\sqrt{mL})$ with $L = 2$. Because $p = m + mKd + m^2 \times (L-2)$, we have

$$
L_{\mathbf{H}} = \log \det(\mathbf{I} + \mathbf{H}) \leq \widetilde{\mathcal{O}}(mKd). \tag{C.2}
$$

For the lower bound of $L_{\mathbf{H}}$, we have

$$
L_{\mathbf{H}} = \log \det(\mathbf{I} + \mathbf{H}) \geq \log \left( \lambda_{\min} (\mathbf{I} + \mathbf{H})^{TK} \right) = TK \log(1 + \lambda_0). \tag{C.3}
$$

## D    PROOF OF THEOREM 5.2 AND THEOREM 5.3

First, define the general neural structure:

$$
\mathbf{f}(\mathbf{x}_t; \boldsymbol{\theta}) := \sqrt{m} \mathbf{W}_L \sigma(\mathbf{W}_{L-1} \ldots \sigma(\mathbf{W}_1 \mathbf{x}_t))) \in \mathbb{R}^K, \tag{D.1}
$$

where $\boldsymbol{\theta} = [\text{vec}(\mathbf{W}_1)^\top, \ldots, \text{vec}(\mathbf{W}_L)^\top]^\top \in \mathbb{R}^p$. Following [3; 15], given an instance $\mathbf{x}$, we define the outputs of hidden layers of the neural network:

$$
\mathbf{g}_{t,0} = \mathbf{x}_t, \mathbf{g}_{t,l} = \sigma(\mathbf{W}_l \mathbf{g}_{t,l-1}), l \in [L-1].
$$

Then, we define the binary diagonal matrix functioning as ReLU:

$$
\mathbf{D}_{t,l} = \text{diag}(\mathbb{1}\{(\mathbf{W}_l \mathbf{g}_{t,l-1})_1\}, \ldots, \mathbb{1}\{(\mathbf{W}_l \mathbf{g}_{t,l-1})_m\}), l \in [L-1].
$$

Accordingly, the neural network is represented by

$$
\mathbf{f}(\mathbf{x}_t; \boldsymbol{\theta}) = \sqrt{m} \mathbf{W}_L (\prod_{l=1}^{L-1} \mathbf{D}_{t,l} \mathbf{W}_l) \mathbf{x}_t,
$$

and we have the following gradient form:

$$
\nabla_{\mathbf{W}_l} \mathbf{f}(\mathbf{x}_t; \boldsymbol{\theta}) = \begin{cases} \sqrt{m} \cdot [\mathbf{g}_{t,l-1} \mathbf{W}_L (\prod_{\tau=l+1}^{L-1} \mathbf{D}_{t,\tau} \mathbf{W}_\tau)]^\top, l \in [L-1] \\ \sqrt{m} \cdot \mathbf{g}_{t,L-1}^\top, l = L. \end{cases}
$$

Let $\boldsymbol{\theta}_1$ be the random initialization of parameters of neural networks. Then, we define the following neural network function class: Given a constant $R > 0$, the function class is defined as

$$
\mathcal{B}(\boldsymbol{\theta}_1, R) = \{\boldsymbol{\theta} \in \mathbb{R}^p : \|\boldsymbol{\theta} - \boldsymbol{\theta}_1\|_2 \leq R/m^{1/2}\}. \tag{D.2}
$$

**Lemma D.1** ([3]). *With probability at least* $1 - \mathcal{O}(TL) \cdot \exp[-\Omega(m\omega^{2/3L})]$, *given* $\omega \leq \mathcal{O}(L^{-9/2}[\log(m)]^{-3/2})$, *for all* $\boldsymbol{\theta}, \boldsymbol{\theta}' \in \mathcal{B}(\boldsymbol{\theta}_1, R), i \in [T], l \in [L-1]$

$$
\|\mathbf{g}_{t,l}\| \leq \mathcal{O}(1)
$$
$$
\|\mathbf{D}_{i,l} - \mathbf{D}'_{i,l}\|_2 \leq \mathcal{O}(L\omega^{2/3}m).
$$

**Lemma D.2.** *With probability at least* $1 - \mathcal{O}(TL^2) \cdot \exp[-\Omega(m\omega^{2/3}L)]$, *uniformly over any diagonal matrices* $\mathbf{D}''_{i,1}, \ldots, \mathbf{D}''_{i,L-1} \in [-1,1]^{m \times m}$ *with at most* $\mathcal{O}(m\omega^{2/3}L)$ *non-zero entries, for any* $\boldsymbol{\theta}, \boldsymbol{\theta}' \in \mathcal{B}(\boldsymbol{\theta}_1; \omega)$ *with* $\omega \leq \mathcal{O}(L^{-6}[\log(m)]^{-3/2})$, *we have the following results:*

$$(1)\|\prod_{\tau \in l}(\mathbf{D}_{i,\tau} + \mathbf{D}''_{i,\tau})\mathbf{W}_\tau\|_2 \leq \mathcal{O}(\sqrt{L}) \tag{D.3}$$

$$(2)\|\mathbf{W}_L \prod_{\tau=l_1}^{L-1}(\mathbf{D}_{i,\tau} + \mathbf{D}''_{i,\tau})\mathbf{W}_\tau\|_F \leq \mathcal{O}\left(\frac{1}{\sqrt{K}}\right) \tag{D.4}$$

$$(3)\left\|\mathbf{W}'_L \prod_{\tau=l_1}^{L-1}(\mathbf{D}'_{i,\tau} + \mathbf{D}''_{i,\tau})\mathbf{W}'_\tau - \mathbf{W}_L \prod_{\tau=l_1}^{L-1}\mathbf{D}_{i,\tau}\mathbf{W}_\tau\right\|_F \leq \mathcal{O}\left(\frac{\omega^{1/3}L^2\sqrt{\log(m)}}{\sqrt{K}}\right). \tag{D.5}$$

*Proof.* Based on Lemma D.1, with high probability at least $1 - \mathcal{O}(nL) \cdot \exp(-\Omega(L\omega^{2/3}m))$, $\|\mathbf{D}'_{i,l} + \mathbf{D}''_{i,l} - \mathbf{D}^{(1)}_{i,l}\|_0 \leq \mathcal{O}(L\omega^{2/3}m)\|_0$. Applying Lemma 8.6 in [3] proves D.3. Then, by lemma 8.7 in [3] with $s = \mathcal{O}(m\omega^{2/3}L)$ to $\mathbf{W}$ and $\mathbf{W}'$, the results hold:

$$\sqrt{m}\left\|\mathbf{W}^{(1)}_L \prod_{\tau=l_1}^{L-1}(\mathbf{D}'_{i,\tau} + \mathbf{D}''_{i,\tau})\mathbf{W}'_\tau - \mathbf{W}^{(1)}_L \prod_{r=l_1}^{L-1}\mathbf{D}^{(1)}_{i,\tau}\mathbf{W}^{(1)}_\tau\right\|_2 \leq \mathcal{O}\left(\frac{\omega^{1/3}L^2\sqrt{m\log(m)}}{\sqrt{K}}\right)$$

$$\sqrt{m}\left\|\mathbf{W}^{(1)}_L \prod_{\tau=l_1}^{L-1}\mathbf{D}_{i,\tau}\mathbf{W}_\tau - \mathbf{W}^{(1)}_L \prod_{\tau=l_1}^{L-1}\mathbf{D}^{(1)}_{i,\tau}\mathbf{W}^{(1)}_\tau\right\| \leq \mathcal{O}\left(\frac{\omega^{1/3}L^2\sqrt{m\log(m)}}{\sqrt{K}}\right). \tag{D.6}$$

Moreover, using Lemma D.1 gain, we have

$$\left\|(\mathbf{W}'_L - \mathbf{W}^{(1)}_L) \prod_{\tau=l_1}^{L-1}(\mathbf{D}'_{i,\tau} + \mathbf{D}''_{i,\tau})\mathbf{W}'_\tau\right\|_2 \leq \mathcal{O}(\sqrt{L}\omega) \leq \mathcal{O}\left(\frac{\omega^{1/3}L^2\sqrt{m\log(m)}}{\sqrt{K}}\right)$$

$$\left\|(\mathbf{W}_L - \mathbf{W}^{(1)}_L) \prod_{\tau=l_1}^{L-1}\mathbf{D}_{i,\tau}\mathbf{W}_\tau\right\|_2 \leq \mathcal{O}(\sqrt{L}\omega) \leq \mathcal{O}\left(\frac{\omega^{1/3}L^2\sqrt{m\log(m)}}{\sqrt{K}}\right) \tag{D.7}$$

Then, combining (D.6) and (D.7) leads the result. For, it has

$$\left\|\mathbf{W}_L \prod_{\tau=l_1}^{L-1}(\mathbf{D}_{i,\tau} + \mathbf{D}''_{i,\tau})\mathbf{W}_\tau\right\|_2 \leq \left\|\mathbf{W}_L \prod_{\tau=l_1}^{L-1}(\mathbf{D}_{i,\tau} + \mathbf{D}''_{i,\tau})\mathbf{W}_\tau - \mathbf{W}^{(1)}_L \prod_{\tau=l_1}^{L-1}\mathcal{D}^{(1)}_{i,\tau}\mathbf{W}^{(1)}_\tau\right\|_2$$

$$+ \|\mathbf{W}^{(1)}_L \prod_{\tau=l_1}^{L-1}\mathbf{D}^{(1)}_{i,\tau}\mathbf{W}^{(1)}_\tau\|_2$$

$$\overset{(a)}{\leq} \mathcal{O}\left(\frac{\omega^{1/3}L^2\sqrt{m\log(m)}}{\sqrt{K}}\right) + \mathcal{O}(\frac{1}{\sqrt{K}}) = \mathcal{O}(\frac{1}{\sqrt{K}})$$

where $(a)$ is applying the and Lemma 7.4 in [3]. The proof is completed. $\square$

**Lemma D.3.** *Suppose the derivative of loss function $\mathcal{L}' \leq \mathcal{O}(1)$. With probability at least $1 - \mathcal{O}(TKL^2) \cdot \exp[-\Omega(m\omega^{2/3}L)]$, for all $t \in [T]$, $\|\boldsymbol{\theta} - \boldsymbol{\theta}_1\| \leq \omega$ and $\omega \leq \mathcal{O}(L^{-6}[\log(m)]^{-3})$, suppose $\|\mathcal{L}_t(\boldsymbol{\theta})'\|_2 \leq \sqrt{K}$, it holds uniformly that*

$$\|\nabla_{\boldsymbol{\theta}}\mathbf{f}(\mathbf{x}_t; \boldsymbol{\theta})\|_2 \leq \mathcal{O}(\sqrt{Lm}) \tag{D.8}$$

$$\|\nabla_{\boldsymbol{\theta}}\mathbf{f}(\mathbf{x}_t; \boldsymbol{\theta})[k]\| \leq \mathcal{O}(\sqrt{Lm}) \tag{D.9}$$

$$\|\nabla_{\boldsymbol{\theta}}\mathcal{L}_t(\boldsymbol{\theta})\|_2 \leq \mathcal{O}\left(\sqrt{(K+L-1)m}\right) \tag{D.10}$$

*Proof.* By Lemma D.1, the result holds:

$$\|\nabla_{\mathbf{W}_L}f(\mathbf{x}_t; \boldsymbol{\theta})\|_F = \|\sqrt{m}\mathbf{g}_{t,L-1}\|_2 \leq \mathcal{O}(\sqrt{m}).$$

For $l \in [L-1]$, the results hold:

$$\|\nabla_{\mathbf{W}_l} f(\mathbf{x}_t; \boldsymbol{\theta})\|_F = \sqrt{m} \|\mathbf{g}_{t,l-1} \mathbf{W}_L (\prod_{\tau=l+1}^{L-1} \mathbf{D}_{t,\tau} \mathbf{W}_\tau)\|$$

$$= \sqrt{m} \cdot \|\mathbf{g}_{t,l-1}\|_2 \cdot \|\mathbf{W}_L (\prod_{\tau=l+1}^{L-1} \mathbf{D}_{t,\tau} \mathbf{W}_\tau)\|_F$$

$$\leq \mathcal{O}\left(\frac{\sqrt{m}}{\sqrt{K}}\right)$$

Thus, applying the union bound, for $l \in [L], t \in [T], k \in [K]$ it holds uniformly that

$$\|\nabla_{\boldsymbol{\theta}} \mathbf{f}(\mathbf{x}_t; \boldsymbol{\theta})\|_2 \leq \sqrt{\sum_{l-1}^{L} \|\nabla_{\mathbf{W}_l} \mathbf{f}(\mathbf{x}_i; \boldsymbol{\theta})\|_F^2} \leq \mathcal{O}\left(\sqrt{\frac{K+L}{K}} m\right) = \mathcal{O}(\sqrt{Lm}).$$

Let $\mathbf{e}_k$ be the $k$-th basis vector. Then, we have

$$\|\nabla_{\boldsymbol{\theta}} \mathbf{f}(\mathbf{x}_t; \boldsymbol{\theta})[k]\|_2 \leq \sqrt{\sum_{l-1}^{L} \|\mathbf{e}_k\|_2^2 \|\nabla_{\mathbf{W}_l} \mathbf{f}(\mathbf{x}_i; \boldsymbol{\theta})\|_F^2} \leq \mathcal{O}(\sqrt{Lm}).$$

These prove (D.8) and (D.9). For $\mathbf{W}_L$, the result holds:

$$\|\nabla_{\mathbf{W}_L} \mathcal{L}_t(\boldsymbol{\theta})\|_F = \|\mathcal{L}_t(\boldsymbol{\theta})'\|_2 \cdot \|\nabla_{\mathbf{W}_L} f(\mathbf{x}_t; \boldsymbol{\theta})\|_F \leq \mathcal{O}(\sqrt{Km}).$$

For $l \in [L-1]$, the results hold:

$$\|\nabla_{\mathbf{W}_l} \mathcal{L}_t(\boldsymbol{\theta})\|_F = \|\mathcal{L}_t(\boldsymbol{\theta})'\|_2 \cdot \|\nabla_{\mathbf{W}_l} f(\mathbf{x}_t; \boldsymbol{\theta})\|_F \leq \mathcal{O}(\sqrt{m}).$$

Therefore, $\|\nabla_{\boldsymbol{\theta}} \mathcal{L}_t(\boldsymbol{\theta})\|_2 = \sqrt{\sum_{l=1}^{L} \|\nabla_{\mathbf{W}_l} \mathcal{L}_t(\boldsymbol{\theta})\|_F^2} \leq \sqrt{(K+L-1)m}.$ $\qquad\square$

**Lemma D.4.** *With probability at least $1 - \mathcal{O}(TL^2 K) \exp[-\Omega(m\omega^{2/3}L)]$ over random initialization, for all $t \in [T]$, and $\boldsymbol{\theta}, \boldsymbol{\theta}'$ satisfying $\|\boldsymbol{\theta} - \boldsymbol{\theta}_1\|_2 \leq w$ and $\|\boldsymbol{\theta}' - \boldsymbol{\theta}_1\|_2 \leq w$ with $\omega \leq \mathcal{O}(L^{-6}[\log m]^{-3/2})$, , it holds uniformly that*

$$|\mathbf{f}(\mathbf{x}_t; \boldsymbol{\theta}')[k] - \mathbf{f}(\mathbf{x}_t; \boldsymbol{\theta})[k] - \langle \nabla_{\boldsymbol{\theta}} \mathbf{f}(\mathbf{x}_t; \boldsymbol{\theta})[k], \boldsymbol{\theta}' - \boldsymbol{\theta}\rangle| \leq \mathcal{O}\left(\frac{\omega^{1/3} L^3 \sqrt{m \log(m)}}{\sqrt{K}}\right) \|\boldsymbol{\theta} - \boldsymbol{\theta}'\|_2.$$

*Proof.* Let $F(\mathbf{x}_t; \boldsymbol{\theta}')[k] = \mathbf{f}(\mathbf{x}_t; \boldsymbol{\theta})[k] - \langle \nabla_{\boldsymbol{\theta}} \mathbf{f}(\mathbf{x}_t; \boldsymbol{\theta})[k], \boldsymbol{\theta}' - \boldsymbol{\theta}\rangle$ and $\mathbf{e}_k \in \mathbb{R}^K$ be the $k$-th basis vector. Then, we have

$$\mathbf{f}(\mathbf{x}_t; \boldsymbol{\theta}')[k] - F(\mathbf{x}_t; \boldsymbol{\theta}')[k] = -\sqrt{m} \sum_{l-1}^{L-1} \mathbf{e}_k^\top \mathbf{W}_L (\sum_{\tau=l+1}^{L-1} \mathbf{D}_{t,\tau} \mathbf{W}_\tau) \mathbf{D}_{t,\tau} (\mathbf{W}_l' - \mathbf{W}_l) \mathbf{g}_{i,l=1}$$

$$+ \sqrt{m} \mathbf{e}_k^\top \mathbf{W}_L' (\mathbf{g}_{t,L-1}' - \mathbf{g}_{t,L-1}).$$

Using Claim 8.2 in [3], there exist diagonal matrices $\mathbf{D}_{t,l}'' \in \mathbb{R}^{m \times m}$ with entries in $[-1, 1]$ such that $\|\mathbf{D}_{t,l}''\|_0 \leq \mathcal{O}(m\omega^{2/3})$ and

$$\mathbf{g}_{t,L-1} - \mathbf{g}_{t,L-1}' = \sum_{l=1}^{L-1} \left[\prod_{\tau=l+1}^{L-1} (\mathbf{D}_{t,\tau}' + \mathbf{D}_{t,\tau}'') \mathbf{W}_\tau'\right] (\mathbf{D}_{t,l}' + \mathbf{D}_{t,l}'')(\mathbf{W}_l - \mathbf{W}_l') \mathbf{g}_{t,l-1}.$$

Thus, we have

$$
\begin{aligned}
|\mathbf{f}(\mathbf{x}_t; \boldsymbol{\theta}')[k] - F(\mathbf{x}_t; \boldsymbol{\theta}')[k]| &= \sqrt{m} \sum_{l=1}^{L-1} \mathbf{e}_k^\top \mathbf{W}_L' \left[ (\mathbf{D}_{t,\tau}' + \mathbf{D}_{t,\tau}'') \mathbf{W}_\tau' (\mathbf{D}_{t,l}' + \mathbf{D}_{t,l}'') (\mathbf{W}_l - \mathbf{W}_l') \mathbf{g}_{t,l-1} \right] \\
&\quad - \sqrt{m} \sum_{l-1}^{L-1} \mathbf{e}_k^\top \mathbf{W}_L ( \sum_{\tau=l+1}^{L-1} \mathbf{D}_{t,\tau} \mathbf{W}_\tau ) \mathbf{D}_{t,l} (\mathbf{W}_l' - \mathbf{W}_l) \mathbf{g}_{i,l-1} \\
&\overset{(a)}{\leq} \left( \frac{\omega^{1/3} L^2 \sqrt{m \log(m)}}{\sqrt{K}} \right) \cdot \sum_{l=1}^{L-1} \|\mathbf{g}_{t,l-1} \cdot \mathbf{W}_l' - \mathbf{W}_l\|_2 \\
&\overset{(b)}{\leq} \left( \frac{\omega^{1/3} L^3 \sqrt{m \log(m)}}{\sqrt{K}} \right) \|\boldsymbol{\theta} - \boldsymbol{\theta}'\|_2
\end{aligned}
$$

where $(a)$ is applying (D.5) and $(b)$ is based on Lemma D.1. The proof is completed.

$\square$

Define $\mathcal{L}_t(\boldsymbol{\theta}) = \|\mathbf{f}(\mathbf{x}_t; \boldsymbol{\theta}) - \mathbf{l}_t\|_2$ and $\mathcal{L}_{t,k} = |\mathbf{f}(\mathbf{x}_t; \boldsymbol{\theta})[k] - \mathbf{l}_t[k]|$, where $\mathbf{l}_t \in \mathbb{R}^K$ represents the corresponding label to train.

**Lemma D.5** (Almost Convexity). *With probability at least $1 - \mathcal{O}(TL^2 K) \exp[-\Omega(m\omega^{2/3}L)]$ over random initialization, for any $\epsilon > 0$ and all $t \in [T]$, and $\boldsymbol{\theta}, \boldsymbol{\theta}'$ satisfying $\|\boldsymbol{\theta} - \boldsymbol{\theta}_1\|_2 \leq \omega$ and $\|\boldsymbol{\theta}' - \boldsymbol{\theta}_1\|_2 \leq \omega$ with $\omega \leq \mathcal{O}\left(\epsilon^{3/4} L^{-9/4} (Km[\log m])^{-3/8}\right) \wedge \mathcal{O}(L^{-6}[\log m]^{-3/2})$, it holds uniformly that*

$$
\mathcal{L}_t(\boldsymbol{\theta}') \geq \mathcal{L}_t(\boldsymbol{\theta}) + \sum_{k=1}^K \langle \nabla_{\boldsymbol{\theta}} \mathcal{L}_{t,k}(\boldsymbol{\theta}), \boldsymbol{\theta}' - \boldsymbol{\theta} \rangle - \epsilon.
$$

*Proof.* Let $\mathcal{L}_{t,k}(\boldsymbol{\theta})$ be the loss function with respect to $\mathbf{f}(\mathbf{x}_t; \boldsymbol{\theta}')[k]$. By convexity of $\mathcal{L}$, it holds uniformly that

$$
\begin{aligned}
&\mathcal{L}_t(\boldsymbol{\theta}') - \mathcal{L}_t(\boldsymbol{\theta}) \\
&\overset{(a)}{\geq} \sum_{k=1}^K \mathcal{L}_{t,k}'(\boldsymbol{\theta}) \left( \mathbf{f}(\mathbf{x}_t; \boldsymbol{\theta}')[k] - \mathbf{f}(\mathbf{x}_t; \boldsymbol{\theta})[k] \right) \\
&\overset{(b)}{\geq} \sum_{k=1}^K \mathcal{L}_{t,k}'(\boldsymbol{\theta}) \langle \nabla \mathbf{f}(\mathbf{x}_t; \boldsymbol{\theta})[k], \boldsymbol{\theta}' - \boldsymbol{\theta} \rangle \\
&\quad - \sum_{k=1}^K \left| \mathcal{L}_{t,k}'(\boldsymbol{\theta}) \cdot [\mathbf{f}(\mathbf{x}_t; \boldsymbol{\theta}')[k] - \mathbf{f}(\mathbf{x}_t; \boldsymbol{\theta})[k] - \langle \nabla \mathbf{f}(\mathbf{x}_t; \boldsymbol{\theta})[k], \boldsymbol{\theta}' - \boldsymbol{\theta} \rangle] \right| \\
&\overset{(c)}{\geq} \sum_{k=1}^K \langle \nabla_{\boldsymbol{\theta}} \mathcal{L}_{t,k}(\boldsymbol{\theta}), \boldsymbol{\theta}' - \boldsymbol{\theta} \rangle - \sum_{k=1}^K |\mathbf{f}(\mathbf{x}_t; \boldsymbol{\theta}')[k] - \mathbf{f}(\mathbf{x}_t; \boldsymbol{\theta})[k] - \langle \nabla \mathbf{f}(\mathbf{x}_t; \boldsymbol{\theta})[k], \boldsymbol{\theta}' - \boldsymbol{\theta} \rangle| \\
&\overset{(d)}{\geq} \sum_{k=1}^K \langle \nabla_{\boldsymbol{\theta}} \mathcal{L}_{t,k}(\boldsymbol{\theta}), \boldsymbol{\theta}' - \boldsymbol{\theta} \rangle - K \cdot \mathcal{O}\left( \frac{\omega^{4/3} L^3 \sqrt{m \log m}}{\sqrt{K}} \right) \\
&\overset{(e)}{\geq} \sum_{k=1}^K \langle \nabla_{\boldsymbol{\theta}} \mathcal{L}_{t,k}(\boldsymbol{\theta}), \boldsymbol{\theta}' - \boldsymbol{\theta} \rangle - \epsilon
\end{aligned}
$$

where $(a)$ is due to the convexity of $\mathcal{L}_t$, $(b)$ is an application of triangle inequality, $(c)$ is because of the Cauchy–Schwarz inequality, $(d)$ is the application of Lemma D.4, and $(e)$ is by the choice of $\omega$. The proof is completed. $\square$

**Lemma D.6** (Loss Bound). *With probability at least $1 - \mathcal{O}(TL^2 K) \exp[-\Omega(m\omega^{2/3}L)]$ over random initialization and suppose $R, \eta, m$ satisfy the condition in Theorem 5.2, the result holds that*

$$
\sum_{t=1}^T \mathcal{L}_t(\boldsymbol{\theta}_t) \leq \sum_{t=1}^T \mathcal{L}_t(\boldsymbol{\theta}^*) + \mathcal{O}(\sqrt{TK}R). \tag{D.11}
$$

*where $\boldsymbol{\theta}^* = \arg\inf_{\boldsymbol{\theta} \in \mathcal{B}(\boldsymbol{\theta}_1, R)} \sum_{t=1}^T \mathcal{L}_t(\boldsymbol{\theta})$.*

*Proof.* Let $w = \mathcal{O}\left(\epsilon^{3/4} L^{-6}(Km)^{-3/8}[\log m]^{-3/2}\right)$ such that the conditions of Lemma D.5 are satisfied. Next, we aim to show $\|\boldsymbol{\theta}_t - \boldsymbol{\theta}_1\|_2 \leq \omega$, for any $t \in [T]$. The proof follows a simple induction. Obviously, $\boldsymbol{\theta}_1$ is in $B(\boldsymbol{\theta}_1, R)$. Suppose that $\boldsymbol{\theta}_1, \boldsymbol{\theta}_2, \ldots, \boldsymbol{\theta}_T \in \mathcal{B}(\boldsymbol{\theta}_1, R)$. We have, for any $t \in [T]$,

$$\|\boldsymbol{\theta}_T - \boldsymbol{\theta}_1\|_2 \leq \sum_{t=1}^T \|\boldsymbol{\theta}_{t+1} - \boldsymbol{\theta}_t\|_2 \leq \sum_{t=1}^T \eta \|\nabla \mathcal{L}_t(\boldsymbol{\theta}_t)\|_2 \leq \sum_{t=1}^T \eta \mathcal{O}(\sqrt{L\kappa m})$$
$$\leq T \cdot \mathcal{O}(\sqrt{L\kappa m}) \cdot \frac{R}{m\sqrt{TK}} \leq \omega$$

when $m > \widetilde{\Omega}(T^4 L^{52} K R^8 \epsilon^{-6})$. This also leads to $\frac{R}{\sqrt{m}} \leq \omega$.

In round $t$, therefore, based on Lemma D.5, for any $\|\boldsymbol{\theta}_t - \boldsymbol{\theta}'\|_2 \leq \omega$, it holds uniformly

$$\mathcal{L}_t(\boldsymbol{\theta}_t) - \mathcal{L}_t(\boldsymbol{\theta}') \leq \sum_{k-1}^K \langle \nabla_{\boldsymbol{\theta}} \mathcal{L}_{t,k}(\boldsymbol{\theta}_t), \boldsymbol{\theta}_t - \boldsymbol{\theta}' \rangle + \epsilon,$$

Then, it holds uniformly

$$\mathcal{L}_t(\boldsymbol{\theta}_t) - \mathcal{L}_t(\boldsymbol{\theta}'_t) \overset{(a)}{\leq} \frac{\langle \boldsymbol{\theta}_t - \boldsymbol{\theta}_{t+1}, \boldsymbol{\theta}_t - \boldsymbol{\theta}' \rangle}{\eta} + \epsilon$$
$$\overset{(b)}{=} \frac{\|\boldsymbol{\theta}_t - \boldsymbol{\theta}'\|_2^2 + \|\boldsymbol{\theta}_t - \boldsymbol{\theta}_{t+1}\|_2^2 - \|\boldsymbol{\theta}_{t+1} - \boldsymbol{\theta}'\|_2^2}{2\eta} + \epsilon$$
$$\leq \frac{\|\boldsymbol{\theta}_t - \boldsymbol{\theta}'\|_2^2 - \|\boldsymbol{\theta}_{t+1} - \boldsymbol{\theta}'\|_2^2}{2\eta} + 2\eta \|\nabla_{\boldsymbol{\theta}} \mathcal{L}_t(\boldsymbol{\theta}_t)\|_2^2 + \epsilon$$
$$\overset{(c)}{\leq} \frac{\|\boldsymbol{\theta}_t - \boldsymbol{\theta}'\|_2^2 - \|\boldsymbol{\theta}_{t+1} - \boldsymbol{\theta}'\|_2^2}{2\eta} + \eta(K + L - 1)m + \epsilon$$

where $(a)$ is because of the definition of gradient descent, $(b)$ is due to the fact $2\langle A, B \rangle = \|A\|_F^2 + \|B\|_F^2 - \|A - B\|_F^2$, $(c)$ is by $\|\boldsymbol{\theta}_t - \boldsymbol{\theta}_{t+1}\|_2^2 = \|\eta \nabla_{\boldsymbol{\theta}} \mathcal{L}_t(\boldsymbol{\theta}_t)\|_2^2 \leq \mathcal{O}(\eta^2(K + L - 1)m)$.

Then, for $T$ rounds, we have

$$\sum_{t=1}^T \mathcal{L}_t(\boldsymbol{\theta}_t) - \sum_{t=1}^T \mathcal{L}_t(\boldsymbol{\theta}'_t)$$
$$\overset{(a)}{\leq} \frac{\|\boldsymbol{\theta}_1 - \boldsymbol{\theta}'\|_2^2}{2\eta} + \sum_{t=2}^T \|\boldsymbol{\theta}_t - \boldsymbol{\theta}'\|_2^2 (\frac{1}{2\eta} - \frac{1}{2\eta}) + \sum_{t=1}^T (L + K - 1)\eta m + T\epsilon$$
$$\leq \frac{\|\boldsymbol{\theta}_1 - \boldsymbol{\theta}'\|_2^2}{2\eta} + \sum_{t=1}^T (L + K - 1)\eta m + T\epsilon$$
$$\leq \frac{R^2}{2m\eta} + T(K + L - 1)\eta m + T\epsilon$$
$$\overset{(b)}{\leq} \mathcal{O}\left(\sqrt{TK}R\right)$$

where $(a)$ is by simply discarding the last term and $(b)$ is setting by $\eta = \frac{R}{m\sqrt{TK}}$, $L \leq K$, and $\epsilon = \frac{\sqrt{K}R}{\sqrt{T}}$. The proof is completed. $\qquad \square$

**Lemma D.7.** *Let* $\mathbf{G} = [\nabla_{\boldsymbol{\theta}_0}\mathbf{f}(\mathbf{x}_1;\boldsymbol{\theta}_0)[1], \nabla_{\boldsymbol{\theta}_0}\mathbf{f}(\mathbf{x}_1;\boldsymbol{\theta}_0)[2], \ldots, \nabla_{\boldsymbol{\theta}_0}\mathbf{f}(\mathbf{x}_T;\boldsymbol{\theta}_0)[K]]/\sqrt{m} \in \mathbb{R}^{p \times TK}$.
*Let* $\mathbf{H}$ *be the NTK defined in. Then, for any* $0 < \delta \leq 1$, *suppose* $m = \Omega(\frac{T^4K^4L^6\log(TKL/\delta)}{\lambda_0^4})$, *then
with probability at least* $1 - \delta$, *we have*

$$\|\mathbf{G}^\top\mathbf{G} - \mathbf{H}\|_F \leq \lambda_0/2;$$
$$\mathbf{G}^\top\mathbf{G} \succeq \mathbf{H}/2.$$

*Proof.* Using Theorem 3.1 in [4], for any $\epsilon > 0$ and $\delta \in (0,1)$, suppose $m = \Omega(\frac{L^6\log(L/\delta)}{\epsilon^4})$, for any
$i, j \in [T], k, k' \in [K]$, with probability at least $1 - \delta$, the result holds;

$$|\langle\nabla_{\boldsymbol{\theta}_0}\mathbf{f}(\mathbf{x}_i;\boldsymbol{\theta}_0)[k], \nabla_{\boldsymbol{\theta}_0}\mathbf{f}(\mathbf{x}_j;\boldsymbol{\theta}_0)[k']\rangle/m - \mathbf{H}_{ik,jk'}| \leq \epsilon.$$

Then, take the union bound over $[T]$ and $[K]$ and set $\epsilon = \frac{\lambda_0}{2TK}$, with probability at least $1 - \delta$, the
result hold

$$\|\mathbf{G}^\top\mathbf{G} - \mathbf{H}\|_F = \sqrt{\sum_{i=1}^{T}\sum_{k=1}^{K}\sum_{j=1}^{T}\sum_{k'=1}^{K}|\langle\nabla_{\boldsymbol{\theta}_0}\mathbf{f}(\mathbf{x}_i;\boldsymbol{\theta}_0)[k], \nabla_{\boldsymbol{\theta}_0}\mathbf{f}(\mathbf{x}_i;\boldsymbol{\theta}_0)[k']\rangle/m - \mathbf{H}_{ik,jk'}|^2}$$
$$\leq TK\epsilon = \frac{\lambda_0}{2}$$

where $m = \Omega(\frac{L^6T^4K^4\log(T^2K^2L/\delta)}{\lambda_0})$.

$\square$

**Lemma D.8.** *Define* $\mathbf{u} = [\ell(\mathbf{y}_{1,1}, \mathbf{y}_1), \cdots, \ell(\mathbf{y}_{T,K}, \mathbf{y}_T)] \in \mathbb{R}^{TK}$ *and* $S' = \sqrt{\mathbf{u}^\top\mathbf{H}^{-1}\mathbf{u}}$. *With
probability at least* $1 - \delta$, *the result holds:*

$$\inf_{\boldsymbol{\theta} \in \mathcal{B}(\boldsymbol{\theta}_0; S')} \sum_{t=1}^{T}\mathcal{L}_t(\boldsymbol{\theta}) \leq \sqrt{TK}S'$$

*Proof.* Suppose the singular value decomposition of $\mathbf{G}$ is $\mathbf{PAQ}^\top, \mathbf{P} \in \mathbb{R}^{p \times TK}, \mathbf{A} \in \mathbb{R}^{TK \times TK}, \mathbf{Q} \in \mathbb{R}^{TK \times TK}$, then, $\mathbf{A} \succeq 0$. Define $\hat{\boldsymbol{\theta}}^* = \boldsymbol{\theta}_0 + \mathbf{PA}^{-1}\mathbf{Q}^\top\mathbf{u}/\sqrt{m}$. Then, we have

$$\sqrt{m}\mathbf{G}^\top(\boldsymbol{\theta}^* - \boldsymbol{\theta}_0) = \mathbf{QAP}^\top\mathbf{PA}^{-1}\mathbf{Q}^\top\mathbf{u} = \mathbf{u}.$$

which leads to

$$\sum_{t=1}^{T}\sum_{k=1}^{K}|\ell(\mathbf{y}_{t,k}, \mathbf{y}_t) - \langle\nabla_{\boldsymbol{\theta}_0}\mathbf{f}(\mathbf{x}_t;\boldsymbol{\theta}_0)[k], \hat{\boldsymbol{\theta}}^* - \boldsymbol{\theta}_0\rangle| = 0.$$

Therefore, the result holds:

$$\|\boldsymbol{\theta}^* - \boldsymbol{\theta}_0\|_2^2 = \mathbf{u}^\top\mathbf{QA}^{-2}\mathbf{Q}^\top\mathbf{u}/m = \mathbf{u}^\top(\mathbf{G}^\top\mathbf{G})^{-1}\mathbf{u}/m \leq \mathbf{u}^\top\mathbf{H}^{-1}\mathbf{u}/m \qquad (D.12)$$

Based on Lemma D.4, given $\omega = \frac{R}{m^{1/2}}$ and initialize $\mathbf{f}(\mathbf{x}_t;\boldsymbol{\theta}_0) \to \mathbf{0}$, we have

$$\sum_{t=1}^{T}\mathcal{L}_t(\boldsymbol{\theta}) \leq \sum_{t=1}^{T}\sum_{k=1}^{K}|\mathbf{y}_t[k] - \langle\nabla_{\boldsymbol{\theta}_0}\mathbf{f}(\mathbf{x}_t;\boldsymbol{\theta}_0)[k], \boldsymbol{\theta}^* - \boldsymbol{\theta}_0\rangle| + TK \cdot \mathcal{O}\left(\omega^{1/3}L^3\sqrt{m\log(m)}\right) \cdot \|\boldsymbol{\theta} - \boldsymbol{\theta}_0\|_2$$

$$\leq \sum_{t=1}^{T}\sum_{k=1}^{K}|\mathbf{y}_t[k] - \langle\nabla_{\boldsymbol{\theta}_0}\mathbf{f}(\mathbf{x}_t;\boldsymbol{\theta}_0)[k], \boldsymbol{\theta}^* - \boldsymbol{\theta}_0\rangle| + TK \cdot \mathcal{O}\left(\omega^{4/3}L^3\sqrt{m\log(m)}\right)$$

$$\leq \sum_{t=1}^{T}\sum_{k=1}^{K}|\mathbf{y}_t[k] - \langle\nabla_{\boldsymbol{\theta}_0}\mathbf{f}(\mathbf{x}_t;\boldsymbol{\theta}_0)[k], \boldsymbol{\theta}^* - \boldsymbol{\theta}_0\rangle| + TK \cdot \mathcal{O}\left((R/m^{1/2})^{4/3}L^3\sqrt{m\log(m)}\right)$$

$$\overset{(a}{\leq} \sum_{t=1}^{T}\sum_{k=1}^{K}|\mathbf{y}_t[k] - \langle\nabla_{\boldsymbol{\theta}_0}\mathbf{f}(\mathbf{x}_t;\boldsymbol{\theta}_0)[k], \boldsymbol{\theta}^* - \boldsymbol{\theta}_0\rangle| + \sqrt{TK}R$$

where $(a)$ is by the choice of $m : m \geq \widetilde{\Omega}(T^3 K^3 R^2)$. Therefore, by putting them together, we have

$$\inf_{\boldsymbol{\theta} \in \mathcal{B}(\boldsymbol{\theta}_0; R)} \sum_{t=1}^{T} \mathcal{L}_t(\boldsymbol{\theta}) \leq \sqrt{TK} S'.$$

where $R = S' = \sqrt{\mathbf{u}^\top \mathbf{H}^{-1} \mathbf{u}}$.

$\square$

## D.1    MAIN LEMMAS

**Lemma D.9.** *Suppose $m, \eta_1, \eta_2$ satisfy the conditions in Theorem 5.3. For any $\delta \in (0, 1)$, with probability at least $1 - \delta$ over the initialization, it holds uniformly that*

$$\frac{1}{T} \sum_{t=1}^{T} \mathop{\mathbb{E}}_{\mathbf{y}_t \sim \mathcal{D}_{\mathcal{X} | \mathbf{x}_t}} \left[ \left\| \mathbf{f}_1(\mathbf{x}_t; \widehat{\boldsymbol{\theta}}_t^1) - (\mathbf{u}_t - \mathbf{f}_2(\phi(\mathbf{x}_t); \widehat{\boldsymbol{\theta}}_t^2)) \right\|_2 \wedge 1 | \mathcal{H}_{t-1} \right]$$
$$\leq \mathcal{O}\left( \sqrt{\frac{K}{T}} \cdot S \right) + 2\sqrt{\frac{2\log(3/\delta)}{T}},$$

*where $\mathbf{u}_t = [\ell(\mathbf{y}_{t,1}, \mathbf{y}_t), \dots, \ell(\mathbf{y}_{t,K}, \mathbf{y}_t)]^\top$, $\mathcal{H}_{t-1} = \{\mathbf{x}_\tau, \mathbf{y}_\tau\}_{\tau=1}^{t-1}$ is historical data.*

*Proof.* This lemma is inspired by Lemma 5.1 in [13]. For any round $t \in [T]$, define

$$V_t = \mathop{\mathbb{E}}_{\mathbf{u}_t} \left[ \|\mathbf{f}_2(\phi(\mathbf{x}_t); \widehat{\boldsymbol{\theta}}_t^2) - (\mathbf{u}_t - \mathbf{f}_1(\mathbf{x}_t; \widehat{\boldsymbol{\theta}}_t^1))\|_2 \wedge 1 \right]$$
$$- \|\mathbf{f}_2(\phi(\mathbf{x}_t); \widehat{\boldsymbol{\theta}}_t^2) - (\mathbf{u}_t - \mathbf{f}_1(\mathbf{x}_t; \widehat{\boldsymbol{\theta}}_t^1))\|_2 \wedge 1 \tag{D.13}$$

Then, we have

$$\mathbb{E}[V_t | F_{t-1}] = \mathop{\mathbb{E}}_{\mathbf{u}_t} \left[ \|\mathbf{f}_2(\phi(\mathbf{x}_t); \widehat{\boldsymbol{\theta}}_t^2) - (\mathbf{u}_t - \mathbf{f}_1(\mathbf{x}_t; \widehat{\boldsymbol{\theta}}_t^1))\|_2 \wedge 1 \right]$$
$$- \mathop{\mathbb{E}}_{\mathbf{u}_t} \left[ \|\mathbf{f}_2(\phi(\mathbf{x}_t); \widehat{\boldsymbol{\theta}}_t^2) - (\mathbf{u}_t - \mathbf{f}_1(\mathbf{x}_t; \widehat{\boldsymbol{\theta}}_t^1))\|_2 \wedge 1 | F_{t-1} \right] \tag{D.14}$$
$$= 0$$

where $F_{t-1}$ denotes the $\sigma$-algebra generated by the history $\mathcal{H}_{t-1}^1$. Therefore, $\{V_t\}_{t=1}^{t}$ are the martingale difference sequence. Then, applying the Hoeffding-Azuma inequality and union bound, we have

$$\mathbb{P}\left[ \frac{1}{T} \sum_{t=1}^{T} V_t - \underbrace{\frac{1}{T} \sum_{t=1}^{T} \mathbb{E}[V_t | \mathbf{F}_{t-1}]}_{I_1} > \sqrt{\frac{2\log(1/\delta)}{T}} \right] \leq \delta \tag{D.15}$$

As $I_1$ is equal to 0, with probability at least $1 - \delta$, we have

$$\frac{1}{T} \sum_{t=1}^{T} \mathop{\mathbb{E}}_{\mathbf{u}_t} \left[ \|\mathbf{f}_2(\phi(\mathbf{x}_t); \widehat{\boldsymbol{\theta}}_t^2) - (\mathbf{u}_t - \mathbf{f}_1(\mathbf{x}_t; \widehat{\boldsymbol{\theta}}_t^1))\|_2 \wedge 1 \right]$$
$$\leq \frac{1}{T} \underbrace{\sum_{t=1}^{T} \|\mathbf{f}_2(\phi(\mathbf{x}_t); \widehat{\boldsymbol{\theta}}_t^2) - (\mathbf{u}_t - \mathbf{f}_1(\mathbf{x}_t; \widehat{\boldsymbol{\theta}}_t^1))\|_2}_{I_2} + \sqrt{\frac{2\log(1/\delta)}{T}} \tag{D.16}$$

For $I_2$, applying the Lemma D.6 and Lemma D.8 to $\boldsymbol{\theta}^2$, we have

$$I_2 \leq \mathcal{O}(\sqrt{TK} S'). \tag{D.17}$$

Combining the above inequalities together and applying the union bound, with probability at least $1 - \delta$, we have

$$
\frac{1}{T} \sum_{t=1}^{T} \mathop{\mathbb{E}}_{\mathbf{u}_t} \left[ \| \mathbf{f}_2(\phi(\mathbf{x}_t); \widehat{\boldsymbol{\theta}}_t^2) - (\mathbf{u}_t - \mathbf{f}_1(\mathbf{x}_t; \widehat{\boldsymbol{\theta}}_t^1)) \|_2 \wedge 1 \right]
$$
$$
\leq \mathcal{O} \left( \sqrt{\frac{K}{T}} \cdot S' \right) + \sqrt{\frac{2 \log(2/\delta)}{T}}. \tag{D.18}
$$

Apply the Hoeffding-Azuma inequality again on $S'$, due to $\mathbb{E}[S'] = S$, the result holds:

$$
\frac{1}{T} \sum_{t=1}^{T} \mathop{\mathbb{E}}_{\mathbf{u}_t} \left[ \| \mathbf{f}_2(\phi(\mathbf{x}_t); \widehat{\boldsymbol{\theta}}_t^2) - (\mathbf{u}_t - \mathbf{f}_1(\mathbf{x}_t; \widehat{\boldsymbol{\theta}}_t^1)) \|_2 \wedge 1 \right]
$$
$$
\leq \mathcal{O} \left( \sqrt{\frac{K}{T}} \cdot S \right) + 2 \sqrt{\frac{2 \log(3/\delta)}{T}},
$$

where the union bound is applied. The proof is completed. $\qquad\square$

Lemma D.10 is an variance of Lemma D.9

**Lemma D.10.** *Suppose $m, \eta_1, \eta_2$ satisfy the conditions in Theorem 5.3. For any $\delta \in (0, 1)$, with probability at least $1 - \delta$ over the random initialization, for all $t \in [T]$, it holds uniformly that*

$$
\mathop{\mathbb{E}}_{(\mathbf{x}_t, \mathbf{y}_t) \sim \mathcal{D}} \left[ \| \mathbf{f}_1(\mathbf{x}_t; \boldsymbol{\theta}_t^1) - (\mathbf{u}_t - \mathbf{f}_2(\phi(\mathbf{x}_t); \boldsymbol{\theta}_t^2)) \|_2 \wedge 1 | \mathcal{H}_{t-1}^1 \right]
$$
$$
\leq \mathcal{O} \left( \sqrt{\frac{K}{t}} \cdot S \right) + 2 \sqrt{\frac{2 \log(3T/\delta)}{t}}, \tag{D.19}
$$

*where $\mathbf{u}_t = (\ell(\mathbf{y}_{t,1}, \mathbf{y}_t), \ldots, \ell(\mathbf{y}_{t,K}, \mathbf{y}_t))^\top$, $\mathcal{H}_{t-1} = \{\mathbf{x}_\tau, \mathbf{y}_\tau\}_{\tau=1}^{t-1}$ is historical data, and the expectation is also taken over $(\boldsymbol{\theta}_t^1, \boldsymbol{\theta}_t^2)$.*

*Proof.* For any round $\tau \in [t]$, define

$$
V_\tau = \mathop{\mathbb{E}}_{(\mathbf{x}_\tau, \mathbf{y}_\tau) \sim \mathcal{D}} \left[ \| \mathbf{f}_2(\phi(\mathbf{x}_\tau); \widehat{\boldsymbol{\theta}}_\tau^2) - (\mathbf{u}_\tau - \mathbf{f}_1(\mathbf{x}_\tau; \widehat{\boldsymbol{\theta}}_\tau^1)) \|_2 \wedge 1 \right]
$$
$$
- \| \mathbf{f}_2(\phi(\mathbf{x}_\tau); \widehat{\boldsymbol{\theta}}_\tau^2) - (\mathbf{u}_\tau - \mathbf{f}_1(\mathbf{x}_\tau; \widehat{\boldsymbol{\theta}}_\tau^1)) \|_2 \wedge 1 \tag{D.20}
$$

Then, we have

$$
\mathbb{E}[V_\tau | F_{\tau-1}] = \mathop{\mathbb{E}}_{(\mathbf{x}_\tau, \mathbf{y}_\tau) \sim \mathcal{D}} \left[ \| \mathbf{f}_2(\phi(\mathbf{x}_\tau); \widehat{\boldsymbol{\theta}}_\tau^2) - (\mathbf{u}_\tau - \mathbf{f}_1(\mathbf{x}_\tau; \widehat{\boldsymbol{\theta}}_\tau^1)) \|_2 \wedge 1 \right]
$$
$$
- \mathop{\mathbb{E}}_{(\mathbf{x}_\tau, \mathbf{y}_\tau) \sim \mathcal{D}} \left[ \| \mathbf{f}_2(\phi(\mathbf{x}_\tau); \widehat{\boldsymbol{\theta}}_\tau^2) - (\mathbf{u}_\tau - \mathbf{f}_1(\mathbf{x}_\tau; \widehat{\boldsymbol{\theta}}_\tau^1)) \|_2 \wedge 1 | F_{\tau-1} \right] \tag{D.21}
$$
$$
= 0
$$

where $F_{\tau-1}$ denotes the $\sigma$-algebra generated by the history $\mathcal{H}_{\tau-1}^1$. Therefore, $\{V_\tau\}_{\tau=1}^{t}$ are the martingale difference sequence. Then, applying the Hoeffding-Azuma inequality and union bound, we have

$$
\mathbb{P} \left[ \frac{1}{t} \sum_{\tau=1}^{t} V_\tau - \underbrace{\frac{1}{t} \sum_{\tau=1}^{t} \mathbb{E}[V_\tau | \mathbf{F}_{\tau-1}]}_{I_1} > \sqrt{\frac{2 \log(1/\delta)}{t}} \right] \leq \delta \tag{D.22}
$$

As $I_1$ is equal to 0, with probability at least $1 - \delta$, we have

$$
\frac{1}{t} \sum_{\tau=1}^{t} \mathop{\mathbb{E}}_{(\mathbf{x}_\tau, \mathbf{y}_\tau) \sim \mathcal{D}} \left[ \| \mathbf{f}_2(\phi(\mathbf{x}_\tau); \widehat{\boldsymbol{\theta}}_\tau^2) - (\mathbf{u}_\tau - \mathbf{f}_1(\mathbf{x}_\tau; \widehat{\boldsymbol{\theta}}_\tau^1)) \|_2 \wedge 1 \right]
$$
$$
\leq \frac{1}{t} \sum_{\tau=1}^{t} \| \mathbf{f}_2(\phi(\mathbf{x}_\tau); \widehat{\boldsymbol{\theta}}_\tau^2) - (\mathbf{u}_\tau - \mathbf{f}_1(\mathbf{x}_\tau; \widehat{\boldsymbol{\theta}}_\tau^1)) \|_2 + \sqrt{\frac{2 \log(1/\delta)}{t}} \tag{D.23}
$$

Based on the the definition of $\boldsymbol{\theta}_{t-1}^1, \boldsymbol{\theta}_{t-1}^2$ in Algorithm 1, we have

$$
\begin{aligned}
& \mathop{\mathbb{E}}_{(\mathbf{x}_t, \mathbf{y}_t) \sim \mathcal{D}(\boldsymbol{\theta}^1, \boldsymbol{\theta}^2)} \left[ \| \mathbf{f}_1(\mathbf{x}_t; \boldsymbol{\theta}_t^1) - (\mathbf{u}_t - \mathbf{f}_2(\mathbf{x}_t; \boldsymbol{\theta}_t^2)) \|_2 \wedge 1 \right] \\
& = \frac{1}{t} \sum_{\tau=1}^{t} \mathop{\mathbb{E}}_{(\mathbf{x}_\tau, \mathbf{y}_\tau) \sim \mathcal{D}} \left[ \| \mathbf{f}_1(\mathbf{x}_\tau; \widehat{\boldsymbol{\theta}}_\tau^1) - (\mathbf{u}_\tau - \mathbf{f}_2(\mathbf{x}_\tau; \widehat{\boldsymbol{\theta}}_\tau^2)) \|_2 \wedge 1 \right].
\end{aligned}
\tag{D.24}
$$

Therefore, putting them together, we have

$$
\begin{aligned}
& \mathop{\mathbb{E}}_{(\mathbf{x}_t, \mathbf{y}_t) \sim \mathcal{D}(\boldsymbol{\theta}^1, \boldsymbol{\theta}^2)} \left[ \| \mathbf{f}_1(\mathbf{x}_t; \boldsymbol{\theta}_{t-1}^1) - (\mathbf{u}_t - \mathbf{f}_2(\mathbf{x}_t; \boldsymbol{\theta}_t^2)) \|_2 \wedge 1 \right] \\
& \leq \underbrace{\frac{1}{t} \sum_{\tau=1}^{t} \| \mathbf{f}_2(\mathbf{x}_\tau; \widehat{\boldsymbol{\theta}}_\tau^2) - (\mathbf{u}_\tau - \mathbf{f}_1(\mathbf{x}_\tau; \widehat{\boldsymbol{\theta}}_\tau^1)) \|_2}_{I_2} + \sqrt{\frac{2 \log(1/\delta)}{t}}.
\end{aligned}
\tag{D.25}
$$

For $I_2$, which is an application of Lemma D.6 and Lemma D.8, we have

$$
I_2 \leq \mathcal{O}(\sqrt{tK} S')
\tag{D.26}
$$

where $(a)$ is because of the choice of $m$.

Combining above inequalities together, with probability at least $1 - \delta$, we have

$$
\begin{aligned}
\mathop{\mathbb{E}}_{(\mathbf{x}_t, \mathbf{y}_t) \sim \mathcal{D}} & \left[ \left| \mathbf{f}_1(\mathbf{x}_t; \boldsymbol{\theta}_{t-1}^1) + \mathbf{f}_2(\phi(\mathbf{x}_t); \boldsymbol{\theta}_{t-1}^2) - \mathbf{u}_t \right| \right] \\
& \leq \mathcal{O} \left( \sqrt{\frac{K}{t}} \cdot S' \right) + \sqrt{\frac{2 \log(2T/\delta)}{t}}.
\end{aligned}
\tag{D.27}
$$

where we apply union bound over $\delta$ to make the above events occur concurrently for all $T$ rounds. Apply the Hoeffding-Azuma inequality again on $S'$ completes the proof. $\qquad\square$

**Lemma D.11.** *Suppose $m, \eta_1, \eta_2$ satisfy the conditions in Theorem 5.3. For any $\delta \in (0, 1), \gamma > 1$, with probability at least $1 - \delta$ over the random initialization, for all $t \in [T]$, when $\mathbf{I}_t = 0$, it holds uniformly that*

$$
\mathop{\mathbb{E}}_{\mathbf{x}_t \sim \mathcal{D}_\mathcal{X}} [\mathbf{h}(\mathbf{x}_t)[\widehat{k}]] = \mathop{\mathbb{E}}_{\mathbf{x}_t \sim \mathcal{D}_\mathcal{X}} [\mathbf{h}(\mathbf{x}_t)[k^*]],
$$

*Proof.* As $\mathbf{I}_t = 0$, we have

$$
|\mathbf{f}(\mathbf{x}_t; \boldsymbol{\theta}_t)[\widehat{k}] - \mathbf{f}(\mathbf{x}_t; \boldsymbol{\theta}_t)[k^\circ]| = \mathbf{f}(\mathbf{x}_t; \boldsymbol{\theta}_t)[\widehat{k}] - \mathbf{f}(\mathbf{x}_t; \boldsymbol{\theta}_t)[k^\circ] \geq 2\gamma\beta_t.
$$

In round $t$, based on Lemma D.10, with probability at least $1 - \delta$, the following event happens:

$$
\widehat{\mathcal{E}}_0 = \left\{ \tau \in [t], k \in [K], \mathop{\mathbb{E}}_{\mathbf{x}_\tau \sim \mathcal{D}_\mathcal{X}} [|\mathbf{f}(\mathbf{x}_\tau; \boldsymbol{\theta}_\tau)[k] - \mathbf{h}(\mathbf{x}_\tau)[k]|] \leq \beta_\tau \right\}.
\tag{D.28}
$$

When $\widehat{\mathcal{E}}_0$ happens with probability at least $1 - \delta$, we have

$$
\begin{cases}
\mathop{\mathbb{E}}_{\mathbf{x}_t \sim \mathcal{D}_\mathcal{X}} [\mathbf{f}(\mathbf{x}_t; \boldsymbol{\theta}_t)[\widehat{k}]] - \beta_t \leq \mathop{\mathbb{E}}_{\mathbf{x}_t \sim \mathcal{D}_\mathcal{X}} [\mathbf{h}(\mathbf{x}_t)[\widehat{k}]] \leq \mathop{\mathbb{E}}_{\mathbf{x}_t \sim \mathcal{D}_\mathcal{X}} [\mathbf{f}(\mathbf{x}_t; \boldsymbol{\theta}_t)[\widehat{k}]] + \beta_t \\
\mathop{\mathbb{E}}_{\mathbf{x}_t \sim \mathcal{D}_\mathcal{X}} [\mathbf{f}(\mathbf{x}_t; \boldsymbol{\theta}_t)[k^\circ]] - \beta_t \leq \mathop{\mathbb{E}}_{\mathbf{x}_t \sim \mathcal{D}_\mathcal{X}} [\mathbf{h}(\mathbf{x}_t)[k^\circ]] \leq \mathop{\mathbb{E}}_{\mathbf{x}_t \sim \mathcal{D}_\mathcal{X}} [\mathbf{f}(\mathbf{x}_t; \boldsymbol{\theta}_t)[k^\circ]] + \beta_t
\end{cases}
\tag{D.29}
$$

Then, with probability at least $1 - \delta$, we have

$$
\begin{aligned}
\mathop{\mathbb{E}}_{\mathbf{x}_t \sim \mathcal{D}_\mathcal{X}} [\mathbf{h}(\mathbf{x}_t)[\widehat{k}] - \mathbf{h}(\mathbf{x}_t)[k^\circ]] & \geq \mathop{\mathbb{E}}_{\mathbf{x}_t \sim \mathcal{D}_\mathcal{X}} [\mathbf{f}(\mathbf{x}_t; \boldsymbol{\theta}_t)[\widehat{k}] - \mathbf{f}(\mathbf{x}_t; \boldsymbol{\theta}_t)[k^\circ]] - 2\beta_t \\
& \geq 2\gamma\beta_t - 2\beta_t \\
& > 0
\end{aligned}
\tag{D.30}
$$

where the last inequality is because of $\gamma > 1$. Then, similarly, for any $k' \in ([K] \setminus \{\widehat{k}, k^\circ\})$, we have $\mathop{\mathbb{E}}_{\mathbf{x}_t \sim \mathcal{D}_\mathcal{X}} [\mathbf{h}(\mathbf{x}_t)[\widehat{k}] - \mathbf{h}(\mathbf{x}_t)] \geq 0$. Thus, based on the definition of $\mathbf{h}(\mathbf{x}_t)[k^*]$, we have $\mathop{\mathbb{E}}_{\mathbf{x}_t \sim \mathcal{D}_\mathcal{X}} [\mathbf{h}(\mathbf{x}_t)[\widehat{k}]] = \mathop{\mathbb{E}}_{\mathbf{x}_t \sim \mathcal{D}_\mathcal{X}} [\mathbf{h}(\mathbf{x}_t)[k^*]]$. The proof is completed. $\qquad\square$

**Lemma D.12.** *When* $t \geq \bar{\mathcal{T}} = \widetilde{\mathcal{O}}(\frac{\gamma^2 (KS^2)}{\epsilon^2})$, *it holds that* $2(\gamma + 1)\boldsymbol{\beta}_t \leq \epsilon$.

*Proof.* To achieve $2(\gamma + 1)\boldsymbol{\beta}_t \leq \epsilon$, there exist constants $C_1, C_2$, such that

$$t \geq \frac{4(\gamma + 1)^2 \cdot \left[ KS^2 + \log(3T/\delta) \right]}{\epsilon^2}$$

$$\Rightarrow \sqrt{\frac{KS^2}{t}} + \sqrt{\frac{2\log(3T/\delta)}{t}}) \leq \frac{\epsilon}{2(\gamma + 1)}$$

The proof is completed. $\qquad\square$

**Lemma D.13.** *Suppose* $m, \eta_1, \eta_2$ *satisfy the conditions in Theorem 5.3. Under Assumption 5.1, for any* $\delta \in (0, 1), \gamma > 1$, *with probability at least* $1 - \delta$ *over the random initialization, when* $t \geq \bar{\mathcal{T}}$, *it holds uniformly:*

$$\mathbb{E}_{\mathbf{x}_t \sim \mathcal{D}_{\mathcal{X}}} [\mathbf{I}_t] = 0,$$

$$\mathbb{E}_{\mathbf{x}_t \sim \mathcal{D}_{\mathcal{X}}} [\mathbf{h}(\mathbf{x}_t)[k^*]] = \mathbb{E}_{\mathbf{x}_t \sim \mathcal{D}_{\mathcal{X}}} [\mathbf{h}(\mathbf{x}_t)[\widehat{k}]].$$

*Proof.* Define the events

$$\mathcal{E}_1 = \left\{ t \geq \bar{\mathcal{T}}, \mathbb{E}_{\mathbf{x}_t \sim \mathcal{D}_{\mathcal{X}}} [\mathbf{h}(\mathbf{x}_t)[k^*] - \mathbf{h}(\mathbf{x}_t)[\widehat{k}]] = 0 \right\},$$

$$\mathcal{E}_2 = \left\{ t \geq \bar{\mathcal{T}}, \mathbb{E}_{\mathbf{x}_t \sim \mathcal{D}_{\mathcal{X}}} [\mathbf{f}(\mathbf{x}_t; \boldsymbol{\theta}_t)[k^*] - \mathbf{f}(\mathbf{x}_t; \boldsymbol{\theta}_t)[\widehat{k}]] = 0 \right\}, \qquad \text{(D.31)}$$

$$\widehat{\mathcal{E}}_1 = \left\{ t \geq \bar{\mathcal{T}}, \mathbb{E}_{\mathbf{x}_t \sim \mathcal{D}_{\mathcal{X}}} [\mathbf{f}(\mathbf{x}_t; \boldsymbol{\theta}_t)[k^*] - \mathbf{f}(\mathbf{x}_t; \boldsymbol{\theta}_t)[k^\circ]] < 2\gamma\boldsymbol{\beta}_t \right\}.$$

The proof is to prove that $\widehat{\mathcal{E}}_1$ will not happen. When $\widehat{\mathcal{E}}_0$ Eq. (D.28) happens with probability at least $1 - \delta$, we have

$$\begin{cases} \mathbb{E}_{\mathbf{x}_t \sim \mathcal{D}_{\mathcal{X}}} [\mathbf{f}(\mathbf{x}_t; \boldsymbol{\theta}_t)[k^*]] - \boldsymbol{\beta}_t \leq \mathbb{E}_{\mathbf{x}_t \sim \mathcal{D}_{\mathcal{X}}} [\mathbf{h}(\mathbf{x}_t)[k^*]] \leq \mathbb{E}_{\mathbf{x}_t \sim \mathcal{D}_{\mathcal{X}}} [\mathbf{f}(\mathbf{x}_t; \boldsymbol{\theta}_t)[k^*]] + \boldsymbol{\beta}_t \\ \mathbb{E}_{\mathbf{x}_t \sim \mathcal{D}_{\mathcal{X}}} [\mathbf{f}(\mathbf{x}_t; \boldsymbol{\theta}_t)[k^\circ]] - \boldsymbol{\beta}_t \leq \mathbb{E}_{\mathbf{x}_t \sim \mathcal{D}_{\mathcal{X}}} [\mathbf{h}(\mathbf{x}_t)[k^\circ]] \leq \mathbb{E}_{\mathbf{x}_t \sim \mathcal{D}_{\mathcal{X}}} [\mathbf{f}(\mathbf{x}_t; \boldsymbol{\theta}_t)[k^\circ]] + \boldsymbol{\beta}_t \end{cases} \qquad \text{(D.32)}$$

Therefore, we have

$$\mathbb{E}_{\mathbf{x}_t \sim \mathcal{D}_{\mathcal{X}}} [\mathbf{h}(\mathbf{x}_t)[k^*] - \mathbf{h}(\mathbf{x}_t)[k^\circ]] \leq \mathbb{E}_{\mathbf{x}_t \sim \mathcal{D}_{\mathcal{X}}} [\mathbf{f}(\mathbf{x}_t; \boldsymbol{\theta}_t)[k^*]] + \boldsymbol{\beta}_t - \left( \mathbb{E}_{\mathbf{x}_t \sim \mathcal{D}_{\mathcal{X}}} [\mathbf{f}(\mathbf{x}_t; \boldsymbol{\theta}_t)[k^\circ]] - \boldsymbol{\beta}_t \right)$$

$$\leq \mathbb{E}_{\mathbf{x}_t \sim \mathcal{D}_{\mathcal{X}}} [\mathbf{f}(\mathbf{x}_t; \boldsymbol{\theta}_t)[k^*] - \mathbf{f}(\mathbf{x}_t; \boldsymbol{\theta}_t)[k^\circ]] + 2\boldsymbol{\beta}_t.$$

Suppose $\widehat{\mathcal{E}}_1$ happens, we have

$$\mathbb{E}_{\mathbf{x}_t \sim \mathcal{D}_{\mathcal{X}}} [\mathbf{h}(\mathbf{x}_t)[k^*] - \mathbf{h}(\mathbf{x}_t)[k^\circ]] \leq 2(\gamma + 1)\boldsymbol{\beta}_t.$$

Then, based on Lemma D.12, when $t > \bar{\mathcal{T}}$, $2(\gamma + 1)\boldsymbol{\beta}_t \leq \epsilon$. Therefore, we have

$$\mathbb{E}_{\mathbf{x}_t \sim \mathcal{D}_{\mathcal{X}}} [\mathbf{h}(\mathbf{x}_t)[k^*] - \mathbf{h}(\mathbf{x}_t)[k^\circ]] \leq 2(\gamma + 1)\boldsymbol{\beta}_t \leq \epsilon.$$

This contradicts Assumption 5.1, i.e., $\mathbf{h}(\mathbf{x}_t)[k^*] - \mathbf{h}(\mathbf{x}_t)[k^\circ] \geq \epsilon$. Hence, $\widehat{\mathcal{E}}_1$ will not happen.

Accordingly, with probability at least $1 - \delta$, the following event will happen

$$\widehat{\mathcal{E}}_2 = \left\{ t \geq \bar{\mathcal{T}}, \mathbb{E}_{\mathbf{x}_t \sim \mathcal{D}_{\mathcal{X}}} [\mathbf{f}(\mathbf{x}_t; \boldsymbol{\theta}_t)[k^*] - \mathbf{f}(\mathbf{x}_t; \boldsymbol{\theta}_t)[k^\circ]] \geq 2\gamma\boldsymbol{\beta}_t \right\}. \qquad \text{(D.33)}$$

Therefore, we have $\mathbb{E}[\mathbf{f}(\mathbf{x}_t; \boldsymbol{\theta}_t)[k^*]] > \mathbb{E}[\mathbf{f}(\mathbf{x}_t; \boldsymbol{\theta}_t)[k^\circ]]$.

Recall that $k^* = \arg\max_{k \in [K]} \mathbf{h}(\mathbf{x}_t)[k]$ and $\widehat{k} = \arg\max_{k \in [K]} \mathbf{f}(\mathbf{x}_t; \boldsymbol{\theta}_t)[k]$. As

$$\forall k \in ([K] \setminus \{\widehat{k}\}), \mathbf{f}(\mathbf{x}_t; \boldsymbol{\theta}_t)[k] \leq \mathbf{f}(\mathbf{x}_t; \boldsymbol{\theta}_t)[k^\circ]$$

$$\Rightarrow \forall k \in ([K] \setminus \{\widehat{k}\}), \mathop{\mathbb{E}}_{\mathbf{x}_t \sim \mathcal{D}_{\mathcal{X}}}[\mathbf{f}(\mathbf{x}_t; \boldsymbol{\theta}_t)[k]] \leq \mathop{\mathbb{E}}_{\mathbf{x}_t \sim \mathcal{D}_{\mathcal{X}}}[\mathbf{f}(\mathbf{x}_t; \boldsymbol{\theta}_t)[k^\circ]],$$

we have

$$\forall k \in ([K] \setminus \{\widehat{k}\}), \mathop{\mathbb{E}}_{\mathbf{x}_t \sim \mathcal{D}_{\mathcal{X}}}[\mathbf{f}(\mathbf{x}_t; \boldsymbol{\theta}_t)[k^*]] > \mathop{\mathbb{E}}_{\mathbf{x}_t \sim \mathcal{D}_{\mathcal{X}}}[\mathbf{f}(\mathbf{x}_t; \boldsymbol{\theta}_t)[k]].$$

Based on the definition of $\widehat{k}$, we have

$$\mathop{\mathbb{E}}_{\mathbf{x}_t \sim \mathcal{D}_{\mathcal{X}}}[\mathbf{f}(\mathbf{x}_t; \boldsymbol{\theta}_t)[k^*]] = \mathop{\mathbb{E}}_{\mathbf{x}_t \sim \mathcal{D}_{\mathcal{X}}}[\mathbf{f}(\mathbf{x}_t; \boldsymbol{\theta}_t)[\widehat{k}]] = \mathop{\mathbb{E}}_{\mathbf{x}_t \sim \mathcal{D}_{\mathcal{X}}}[\max_{i \in [k]}\mathbf{f}(\mathbf{x}_t; \boldsymbol{\theta}_t)[k]]. \tag{D.34}$$

This indicates $\mathcal{E}_2$ happens with probability at least $1 - \delta$.

Therefore, based on $\widehat{\mathcal{E}}_2$ and (D.34), the following inferred event $\widehat{\mathcal{E}}_3$ happens with probability at least $1 - \delta$:

$$\widehat{\mathcal{E}}_3 = \left\{ t \geq \bar{\mathcal{T}}, \mathop{\mathbb{E}}_{\mathbf{x}_t \sim \mathcal{D}_{\mathcal{X}}}[\mathbf{f}(\mathbf{x}_t; \boldsymbol{\theta}_t)[\widehat{k}] - \mathbf{f}(\mathbf{x}_t; \boldsymbol{\theta}_t)[k^\circ]] \geq 2\gamma\boldsymbol{\beta}_t \right\}.$$

Then, based on Eq. D.32, we have

$$\begin{aligned}
\mathbb{E}[\mathbf{h}(\mathbf{x}_t)[\widehat{k}] - \mathbf{h}(\mathbf{x}_t)[k^\circ]] &\geq \mathbb{E}[\mathbf{f}(\mathbf{x}_t; \boldsymbol{\theta}_t)[\widehat{k}]] - \boldsymbol{\beta}_t - (\mathbb{E}[\mathbf{f}(\mathbf{x}_t; \boldsymbol{\theta}_t)[k^\circ]] + \boldsymbol{\beta}_t) \\
&= \mathbb{E}[\mathbf{f}(\mathbf{x}_t; \boldsymbol{\theta}_t)[\widehat{k}] - \mathbf{f}(\mathbf{x}_t; \boldsymbol{\theta}_t)[k^\circ]] - 2\boldsymbol{\beta}_t \\
&\overset{E_1}{\geq} 2(\gamma - 1)\boldsymbol{\beta}_t \\
&> 0
\end{aligned} \tag{D.35}$$

where $E_1$ is because $\widehat{\mathcal{E}}_3$ happened with probability at least $1 - \delta$. Therefore, we have

$$\mathop{\mathbb{E}}_{\mathbf{x}_t \sim \mathcal{D}_{\mathcal{X}}}[\mathbf{h}(\mathbf{x}_t)[\widehat{k}]] - \mathop{\mathbb{E}}_{\mathbf{x}_t \sim \mathcal{D}_{\mathcal{X}}}[\mathbf{h}(\mathbf{x}_t)[k^\circ]] > 0.$$

Similarly, we can prove that

$$\Rightarrow \forall k \in ([K] \setminus \{\widehat{k}\}), \mathop{\mathbb{E}}_{\mathbf{x}_t \sim \mathcal{D}_{\mathcal{X}}}[\mathbf{h}(\mathbf{x}_t)[\widehat{k}]] - \mathop{\mathbb{E}}_{\mathbf{x}_t \sim \mathcal{D}_{\mathcal{X}}}[\mathbf{h}(\mathbf{x}_t)[k]] > 0.$$

Then, based on the definition of $k^*$, we have

$$\mathop{\mathbb{E}}_{\mathbf{x}_t \sim \mathcal{D}_{\mathcal{X}}}[\mathbf{h}(\mathbf{x}_t)[\widehat{k}]] = \mathop{\mathbb{E}}_{\mathbf{x}_t \sim \mathcal{D}_{\mathcal{X}}}[\mathbf{h}(\mathbf{x}_t)[k^*]] = \mathop{\mathbb{E}}_{\mathbf{x}_t \sim \mathcal{D}_{\mathcal{X}}}[\max_{k \in [K]} \mathbf{h}(\mathbf{x}_t)[k]].$$

Thus, the event $\mathcal{E}_1$ happens with probability at least $1 - \delta$. $\qquad\square$

## D.2 LABEL COMPLEXITY

**Lemma D.14** (Label Complexity Analysis). *For any $\delta \in (0, 1), \gamma \geq 1$, suppose $m$ satisfies the conditions in Theorem 5.2. Then, with probability at least $1 - \delta$, we have*

$$\mathbf{N}_T \leq \bar{\mathcal{T}}. \tag{D.36}$$

*Proof.* Recall that $\mathbf{x}_{t,\widehat{i}} = \max_{\mathbf{x}_{t,i}, i \in [k]} \mathbf{f}(\mathbf{x}_t; \boldsymbol{\theta}_t)[k]$, and $\mathbf{x}_{t,i^\circ} = \max_{\mathbf{x}_{t,i}, i \in ([k]/\{\mathbf{x}_{t,\widehat{i}}\})} \mathbf{f}(\mathbf{x}_t; \boldsymbol{\theta}_t)[k]$. With probability at least $1 - \delta$, according to Eq. (D.28) the event

$$\widehat{\mathcal{E}}_0 = \left\{ \tau \in [t], k \in [K], \mathop{\mathbb{E}}_{\mathbf{x}_\tau \sim \mathcal{D}_{\mathcal{X}}}[|\mathbf{f}(\mathbf{x}_\tau; \boldsymbol{\theta}_\tau)[k] - \mathbf{h}(\mathbf{x}_\tau)[k]|] \leq \boldsymbol{\beta}_\tau \right\}$$

happens. Therefore, we have

$$\begin{cases}
\mathop{\mathbb{E}}_{\mathbf{x}_t \sim \mathcal{D}_{\mathcal{X}}}[\mathbf{h}(\mathbf{x}_t)[\widehat{k}]] - \boldsymbol{\beta}_t \leq \mathop{\mathbb{E}}_{\mathbf{x}_t \sim \mathcal{D}_{\mathcal{X}}}[\mathbf{f}(\mathbf{x}_t; \boldsymbol{\theta}_t)[\widehat{k}]] \leq \mathop{\mathbb{E}}_{\mathbf{x}_t \sim \mathcal{D}_{\mathcal{X}}}[\mathbf{h}(\mathbf{x}_t)[\widehat{k}]] + \boldsymbol{\beta}_t \\
\mathop{\mathbb{E}}_{\mathbf{x}_t \sim \mathcal{D}_{\mathcal{X}}}[\mathbf{h}(\mathbf{x}_t)[k^\circ]] - \boldsymbol{\beta}_t \leq \mathop{\mathbb{E}}_{\mathbf{x}_t \sim \mathcal{D}_{\mathcal{X}}}[\mathbf{f}(\mathbf{x}_t; \boldsymbol{\theta}_t)[k^\circ]] \leq \mathop{\mathbb{E}}_{\mathbf{x}_t \sim \mathcal{D}_{\mathcal{X}}}[\mathbf{h}(\mathbf{x}_t)[k^\circ]] + \boldsymbol{\beta}_t.
\end{cases}$$

Then, we have

$$
\begin{cases}
\mathop{\mathbb{E}}\limits_{\mathbf{x}_t \sim \mathcal{D}_{\mathcal{X}}} [\mathbf{f}(\mathbf{x}_t; \boldsymbol{\theta}_t)[\widehat{k}] - \mathbf{f}(\mathbf{x}_t; \boldsymbol{\theta}_t)[k^\circ]] \leq \mathop{\mathbb{E}}\limits_{\mathbf{x}_t \sim \mathcal{D}_{\mathcal{X}}} [\mathbf{h}(\mathbf{x}_t)[\widehat{k}]] - \mathop{\mathbb{E}}\limits_{\mathbf{x}_t \sim \mathcal{D}_{\mathcal{X}}} [\mathbf{h}(\mathbf{x}_t)[k^\circ]] + 2\boldsymbol{\beta}_t \\
\mathop{\mathbb{E}}\limits_{\mathbf{x}_t \sim \mathcal{D}_{\mathcal{X}}} [\mathbf{f}(\mathbf{x}_t; \boldsymbol{\theta}_t)[\widehat{k}] - \mathbf{f}(\mathbf{x}_t; \boldsymbol{\theta}_t)[k^\circ]] \geq \mathop{\mathbb{E}}\limits_{\mathbf{x}_t \sim \mathcal{D}_{\mathcal{X}}} [\mathbf{h}(\mathbf{x}_t)[\widehat{k}]] - \mathop{\mathbb{E}}\limits_{\mathbf{x}_t \sim \mathcal{D}_{\mathcal{X}}} [\mathbf{h}(\mathbf{x}_t)[k^\circ]] - 2\boldsymbol{\beta}_t.
\end{cases}
\tag{D.37}
$$

Let $\epsilon_t = | \mathop{\mathbb{E}}\limits_{\mathbf{x}_t \sim \mathcal{D}_{\mathcal{X}}} [\mathbf{h}(\mathbf{x}_t)[\widehat{k}]] - \mathop{\mathbb{E}}\limits_{\mathbf{x}_t \sim \mathcal{D}_{\mathcal{X}}} [\mathbf{h}(\mathbf{x}_t)[k^\circ]]|$. Then, based on Lemma D.12 and Lemma D.13, when $t \geq \bar{\mathcal{T}}$, we have

$$
2(\gamma + 1)\boldsymbol{\beta}_t \leq \epsilon \leq \epsilon_t \leq 1. \tag{D.38}
$$

For any $t \in [T]$ and $t < \bar{\mathcal{T}}$, we have $\mathop{\mathbb{E}}\limits_{\mathbf{x}_t \sim \mathcal{D}_{\mathcal{X}}} [\mathbf{I}_t] \leq 1$.

For the round $t > \bar{\mathcal{T}}$, based on Lemma D.13, it holds uniformly $\mathop{\mathbb{E}}\limits_{\mathbf{x}_t \sim \mathcal{D}_{\mathcal{X}}} [\mathbf{h}(\mathbf{x}_t)[\widehat{k}]] - \mathop{\mathbb{E}}\limits_{\mathbf{x}_t \sim \mathcal{D}_{\mathcal{X}}} [\mathbf{h}(\mathbf{x}_t)[k^\circ]] = \epsilon_t$. Then, based on (D.37), we have

$$
\mathop{\mathbb{E}}\limits_{\mathbf{x}_t \sim \mathcal{D}_{\mathcal{X}}} [\mathbf{f}(\mathbf{x}_t; \boldsymbol{\theta}_t)[\widehat{k}] - \mathbf{f}(\mathbf{x}_t; \boldsymbol{\theta}_t)[k^\circ]] \geq \epsilon_t - 2\boldsymbol{\beta}_t \overset{E_2}{\geq} 2\gamma\boldsymbol{\beta}_t, \tag{D.39}
$$

where $E_2$ is because of Eq. (D.38).

According to Lemma D.13, when $t > \bar{\mathcal{T}}$, $\mathop{\mathbb{E}}\limits_{\mathbf{x}_t \sim \mathcal{D}_{\mathcal{X}}} [\mathbf{f}(\mathbf{x}_t; \boldsymbol{\theta}_t)[k^*]] = \mathop{\mathbb{E}}\limits_{\mathbf{x}_t \sim \mathcal{D}_{\mathcal{X}}} [\mathbf{f}(\mathbf{x}_t; \boldsymbol{\theta}_t)[\widehat{k}]]$. Thus, it holds uniformly

$$
\mathbf{f}(\mathbf{x}_t; \boldsymbol{\theta}_t)[\widehat{k}] - \mathbf{f}(\mathbf{x}_t; \boldsymbol{\theta}_t)[k^\circ] \geq 2\gamma\boldsymbol{\beta}_t, t > \bar{\mathcal{T}}.
$$

Then, for the round $t > \bar{\mathcal{T}}$, we have $\mathop{\mathbb{E}}\limits_{\mathbf{x}_t \sim \mathcal{D}_{\mathcal{X}}} [\mathbf{I}_t] = 0$.

Then, assume $T > \bar{\mathcal{T}}$, we have

$$
\begin{aligned}
\mathbf{N}_T &= \sum_{t=1}^{T} \mathop{\mathbb{E}}\limits_{\mathbf{x}_t \sim \mathcal{D}_{\mathcal{X}}} \left[ \mathbb{1}\{\mathbf{f}(\mathbf{x}_t; \boldsymbol{\theta}_t)[\widehat{k}] - \mathbf{f}(\mathbf{x}_t; \boldsymbol{\theta}_t)[k^\circ] < 2\gamma\boldsymbol{\beta}_t\} \right] \\
&\leq \sum_{t=1}^{\bar{\mathcal{T}}} 1 + \sum_{t=\bar{\mathcal{T}}+1}^{T} \mathop{\mathbb{E}}\limits_{\mathbf{x}_t \sim \mathcal{D}_{\mathcal{X}}} \left[ \mathbb{1}\{\mathbf{f}(\mathbf{x}_t; \boldsymbol{\theta}_t)[\widehat{k}] - \mathbf{f}(\mathbf{x}_t; \boldsymbol{\theta}_t)[k^\circ] < 2\gamma\boldsymbol{\beta}_t\} \right] \\
&= \bar{\mathcal{T}} + 0.
\end{aligned}
\tag{D.40}
$$

Therefore, we have $\mathbf{N}_T \leq \bar{\mathcal{T}}$. $\qquad\qquad\square$

**Theorem 5.2.** *[Stream-based]. Given $T$, for any $\delta \in (0, 1), \lambda_0 > 0$, suppose $\|\mathbf{x}_t\|_2 = 1, t \in [T]$, $\mathbf{H} \succeq \lambda_0 \mathbf{I}$, $m \geq \widetilde{\Omega}(poly(T, K, L, S) \cdot \log(1/\delta)), \eta_1 = \eta_2 = \Theta(\frac{S}{m\sqrt{TK}})$. Then, with probability at least $1 - \delta$ over the initialization of $\boldsymbol{\theta}_1^1, \boldsymbol{\theta}_1^2$, Algorithm 1 achieves the following regret bound:*

$$
\mathbf{R}_{stream}(T) \leq \mathcal{O}(\sqrt{T}) \cdot \left( \sqrt{K}S + \sqrt{2\log(3T/\delta)} \right)
$$

*where $\mathbf{N}(T) \leq \mathcal{O}(T)$.*

*Proof.* Define $R_t = \underset{\mathbf{x}_t \sim \mathcal{D}_{\mathcal{X}}}{\mathbb{E}} \left[ \mathbf{h}(\mathbf{x}_t)[\widehat{k}] - \mathbf{h}(\mathbf{x}_t)[k^*] \right]$.

$$
\begin{aligned}
\mathbf{R}_{stream}(T) &= \sum_{t=1}^{T} R_t(\mathbf{I}_t = 1 \vee \mathbf{I}_t = 0) \\
&\leq \sum_{t=1}^{T} \max\{R_t(\mathbf{I}_t = 1), R_t(\mathbf{I}_t = 0)\} \\
&\overset{(a)}{\leq} \sum_{t=1}^{T} \underset{(\mathbf{x}_t, \mathbf{y}_t) \sim \mathcal{D}}{\mathbb{E}} [\|\mathbf{f}(\mathbf{x}_t; \boldsymbol{\theta}_{t-1}) - \mathbf{u}_t\|_2] \\
&\leq \sum_{t=1}^{T} \mathcal{O}\left(\sqrt{\frac{K}{t}} \cdot S\right) + \mathcal{O}\left(\sqrt{\frac{2\log(3T/\delta)}{t}}\right) \\
&\leq \mathcal{O}(\sqrt{TK}S) + \mathcal{O}\left(\sqrt{2T\log(3T/\delta)}\right),
\end{aligned}
$$

where (a) is based on Lemma D.11: $R_t(\mathbf{I}_t = 0) = 0$. The proof is completed. $\qquad\square$

**Theorem 5.3.** *[Stream-based].* Given $T$, for any $\delta \in (0,1), \gamma > 1, \lambda_0 > 0$, suppose $\|\mathbf{x}_t\|_2 = 1, t \in [T]$, $\mathbf{H} \succeq \lambda_0 \mathbf{I}$, $m \geq \widetilde{\Omega}(poly(T, K, L, S) \cdot \log(1/\delta))$, $\eta_1 = \eta_2 = \Theta(\frac{S}{m\sqrt{TK}})$, and Assumption 5.1 holds. Then, with probability at least $1 - \delta$ over the initialization of $\boldsymbol{\theta}_1^1, \boldsymbol{\theta}_1^2$, Algorithm 1 achieves the following regret bound and label complexity:

$$
\mathbf{R}_{stream}(T) \leq \mathcal{O}((KS^2 + \log(3T/\delta))/\epsilon), \ \mathbf{N}(T) \leq \mathcal{O}((KS^2 + \log(3T/\delta))/\epsilon^2). \tag{5.2}
$$

*Proof.* Given $\bar{\mathcal{T}}$, we divide rounds into the follow two pars. Then, it holds that

$$
\begin{aligned}
\mathbf{R}_{stream}(T) &= \sum_{t=1}^{T} R_t(\mathbf{I}_t = 1 \vee \mathbf{I}_t = 0) \\
&= \sum_{t=1}^{\bar{\mathcal{T}}} R_t(\mathbf{I}_t = 1) + \sum_{t=\bar{\mathcal{T}}+1}^{T} R_t(\mathbf{I}_t = 0) \\
&= \underbrace{\sum_{t=1}^{\bar{\mathcal{T}}} \underset{\mathbf{x}_t \sim \mathcal{D}_{\mathcal{X}}}{\mathbb{E}} [\mathbf{h}(\mathbf{x}_t)[\widehat{k}] - \mathbf{h}(\mathbf{x}_t)[k^*]}_{I_1} + \underbrace{\sum_{t=\bar{\mathcal{T}}+1}^{T} \underset{\mathbf{x}_t \sim \mathcal{D}_{\mathcal{X}}}{\mathbb{E}} [\mathbf{h}(\mathbf{x}_t)[\widehat{k}] - \mathbf{h}(\mathbf{x}_t)[k^*]}_{I_2}
\end{aligned} \tag{D.41}
$$

For $I_1$, it holds that

$$
\begin{aligned}
R_t(\mathbf{I}_t = 1) &= \underset{\mathbf{x}_t \sim \mathcal{D}_{\mathcal{X}}}{\mathbb{E}} \left[ \mathbf{h}(\mathbf{x}_t)[\widehat{k}] - \mathbf{h}(\mathbf{x}_t)[k^*] \right] \\
&= \underset{\mathbf{x}_t \sim \mathcal{D}_{\mathcal{X}}}{\mathbb{E}} \left[ \mathbf{h}(\mathbf{x}_t)[\widehat{k}] - \mathbf{f}(\mathbf{x}_t; \boldsymbol{\theta}_t)[\widehat{k}] + \mathbf{f}(\mathbf{x}_t; \boldsymbol{\theta}_t)[\widehat{k}] - \mathbf{h}(\mathbf{x}_t)[k^*] \right] \\
&\overset{(a)}{\leq} \underset{\mathbf{x}_t \sim \mathcal{D}_{\mathcal{X}}}{\mathbb{E}} \left[ \mathbf{h}(\mathbf{x}_t)[\widehat{k}] - \mathbf{f}(\mathbf{x}_t; \boldsymbol{\theta}_t)[\widehat{k}] + \mathbf{f}(\mathbf{x}_t; \boldsymbol{\theta}_t)[k^*] - \mathbf{h}(\mathbf{x}_t)[k^*] \right] \\
&\leq \underset{\mathbf{x}_t \sim \mathcal{D}_{\mathcal{X}}}{\mathbb{E}} [|\mathbf{h}(\mathbf{x}_t)[\widehat{k}] - \mathbf{f}(\mathbf{x}_t; \boldsymbol{\theta}_t)[\widehat{k}]|] + \underset{\mathbf{x}_t \sim \mathcal{D}_{\mathcal{X}}}{\mathbb{E}} [|\mathbf{f}(\mathbf{x}_t; \boldsymbol{\theta}_t)[k^*] - \mathbf{h}(\mathbf{x}_t)[k^*]|] \\
&\leq 2 \underset{\mathbf{x}_t \sim \mathcal{D}_{\mathcal{X}}}{\mathbb{E}} [\|\mathbf{h}(\mathbf{x}_t) - \mathbf{f}(\mathbf{x}_t; \boldsymbol{\theta}_t)\|_{\infty}] \\
&\leq 2 \underset{(\mathbf{x}_t, \mathbf{y}_t) \sim \mathcal{D}}{\mathbb{E}} [\|\mathbf{f}(\mathbf{x}_t; \boldsymbol{\theta}_{t-1}) - \mathbf{u}_t\|_2] \\
&\overset{(b)}{\leq} \mathcal{O}\left(\sqrt{\frac{K}{T}} \cdot S\right) + \mathcal{O}\left(\sqrt{\frac{2\log(3T/\delta)}{t}}\right),
\end{aligned}
$$

where $(a)$ is duo the selection criterion of NEURONAL and $(b)$ is an application of D.10. Then,

$$I_1 \leq \sum_{t=1}^{\bar{\mathcal{T}}} \mathcal{O}\left(\sqrt{\frac{K}{T}} \cdot S\right) + \mathcal{O}\left(\sqrt{\frac{2\log(3T/\delta)}{t}}\right)$$
$$\leq (2\sqrt{\bar{\mathcal{T}}} - 1)\left[\sqrt{K}S + \sqrt{2\log(3T/\delta)}\right]$$

For $I_2$, we have $R_t|(\mathbf{I}_t = 0) = \mathbb{E}_{\mathbf{x}_t \sim \mathcal{D}_{\mathcal{X}}}[\mathbf{h}(\mathbf{x}_t)[\widehat{k}] - \mathbf{h}(\mathbf{x}_t)[k^*]] = 0$ based on Lemma D.13.

Therefore, it holds that:

$$\mathbf{R}_{stream}(T) \leq (2\sqrt{\bar{\mathcal{T}}} - 1)\left[\sqrt{K}S + \sqrt{2\log(3T/\delta)}\right].$$

The proof is completed. $\qquad\qquad\square$

# E  ANALYSIS IN BINARY CLASSIFICATION

First, we provide the noise condition used in [58].

**Assumption E.1** ( Mammen-Tsybakov low noise [43; 58])**.** There exist absolute constants $c > 0$ and $\alpha \geq 0$, such that for all $0 < \epsilon < 1/2, \mathbf{x} \in \mathcal{X}, k \in \{0, 1\}$,

$$\mathbb{P}(|\mathbf{h}(\mathbf{x})[k] - \frac{1}{2}| \leq \epsilon) \leq c\epsilon^\alpha$$

.

Then, we provide the following theorem.

**Theorem 5.1.** *[Binary Classification]  Given $T$, for any $\delta \in (0, 1)$, $\lambda_0 > 0$, suppose $K = 2$, $\|\mathbf{x}_t\|_2 = 1, t \in [T]$, $\mathbf{H} \succeq \lambda_0 \mathbf{I}$, $m \geq \widetilde{\Omega}(poly(T, L, S) \cdot \log(1/\delta))$, $\eta_1 = \eta_2 = \Theta(\frac{S}{m\sqrt{2T}})$. Then, with probability at least $1 - \delta$ over the initialization of $\boldsymbol{\theta}_1^1, \boldsymbol{\theta}_1^2$, Algorithm 1 achieves the following regret bound:*

$$\mathbf{R}_{stream}(T) \leq \widetilde{\mathcal{O}}((S^2)^{\frac{\alpha+1}{\alpha+2}} T^{\frac{1}{\alpha+2}}),$$
$$\mathbf{N}(T) \leq \widetilde{\mathcal{O}}((S^2)^{\frac{\alpha}{\alpha+2}} T^{\frac{2}{\alpha+2}}).$$

In comparison, with assumption E.1, [58] achieves the following results:

$$\mathbf{R}_{stream}(T) \leq \widetilde{\mathcal{O}}\left(L_{\mathbf{H}}^{\frac{2(\alpha+1)}{\alpha+2}} T^{\frac{1}{\alpha+2}}\right) + \widetilde{\mathcal{O}}\left(L_{\mathbf{H}}^{\frac{\alpha+1}{\alpha+2}}(S^2)^{\frac{\alpha+1}{\alpha+2}} T^{\frac{1}{\alpha+2}}\right),$$
$$\mathbf{N}(T) \leq \widetilde{\mathcal{O}}\left(L_{\mathbf{H}}^{\frac{2\alpha}{\alpha+2}} T^{\frac{2}{\alpha+2}}\right) + \widetilde{\mathcal{O}}\left(L_{\mathbf{H}}^{\frac{\alpha}{\alpha+2}}(S^2)^{\frac{\alpha}{\alpha+2}} T^{\frac{2}{\alpha+2}}\right).$$

For the regret $\mathbf{R}_{stream}(T)$, compared to [58], Theorem 5.1 removes the term $\widetilde{\mathcal{O}}\left(L_{\mathbf{H}}^{\frac{2(\alpha+1)}{\alpha+2}} T^{\frac{1}{\alpha+2}}\right)$ and further improve the regret upper bound by a multiplicative factor $L_{\mathbf{H}}^{\frac{\alpha+1}{\alpha+2}}$. For the label complexity $\mathbf{N}(T)$, compared to [58], Theorem 5.1 removes the term $\widetilde{\mathcal{O}}\left(L_{\mathbf{H}}^{\frac{2\alpha}{\alpha+2}} T^{\frac{2}{\alpha+2}}\right)$ and further improve the regret upper bound by a multiplicative factor $L_{\mathbf{H}}^{\frac{\alpha}{\alpha+2}}$. It is noteworthy that $L_{\mathbf{H}}$ grows linearly with respect to $T$, i.e., $L_{\mathbf{H}} \geq T\log(1 + \lambda_0)$.

*Proof.* First, we define

$$\Delta_t = \mathbf{h}(\mathbf{x}_t)[\widehat{k}] - \mathbf{h}(\mathbf{x}_t)[k^\circ]$$
$$\widehat{\Delta}_t = f(\mathbf{x}_t; \boldsymbol{\theta}_t)[\widehat{k}] - f(\mathbf{x}_t; \boldsymbol{\theta}_t)[k^\circ] + 2\boldsymbol{\beta}_t \qquad\qquad \text{(E.1)}$$
$$T_\epsilon = \sum_{t=1}^{T} \mathbb{E}_{\mathbf{x}_t \sim \mathcal{D}_{\mathcal{X}}}[\mathbb{1}\{\Delta_t^2 \leq \epsilon^2\}].$$

**Lemma E.1.** *Suppose the conditions of Theorem 5.1 are satisfied. With probability at least $1 - \delta$, the following result holds:*

$$\mathbf{N}_T \leq \mathcal{O}\left(T_\epsilon + \frac{1}{\epsilon^2}(S^2 + \log(3T/\delta))\right).$$

*Proof.* First, based on Lemma D.11, with probability at least $1 - \delta$, we have $0 \leq \widehat{\Delta}_t - \Delta_t \leq 4\boldsymbol{\beta}_t$. Because $\widehat{\Delta}_t \geq 0$, then $\widehat{\Delta}_t \leq 4\boldsymbol{\beta}_t$ implies $\Delta_t \leq 4\boldsymbol{\beta}_t$.

Then, we have

$$
\begin{aligned}
\mathbf{I}_t &= \mathbf{I}_t \mathbb{1}\{\widehat{\Delta}_t \leq 4\boldsymbol{\beta}_t\} \\
&\leq \mathbf{I}_t \mathbb{1}\{\widehat{\Delta}_t \leq 4\boldsymbol{\beta}_t, 4\boldsymbol{\beta}_t \geq \epsilon\} + \mathbf{I}_t \mathbb{1}\{\widehat{\Delta}_t \leq 4\boldsymbol{\beta}_t, 4\boldsymbol{\beta}_t < \epsilon\} \\
&\leq \frac{16\mathbf{I}_t\boldsymbol{\beta}_t^2}{\epsilon^2} \wedge 1 + \mathbb{1}\{\Delta_t^2 \leq \epsilon^2\}.
\end{aligned}
$$

Thus, we have

$$
\begin{aligned}
\mathbf{N}_T &= \sum_{t=1}^T \mathbb{E}_{\mathbf{x}_t \sim \mathcal{D}_{\mathcal{X}}}[\mathbf{I}_t \mathbb{1}\{\widehat{\Delta}_t \leq 4\boldsymbol{\beta}_t\}] \\
&\leq \frac{1}{\epsilon^2} \sum_{t=1}^T (16\mathbf{I}_t\boldsymbol{\beta}_t^2 \wedge \epsilon^2) + T_\epsilon \\
&\leq \frac{1}{\epsilon^2} \sum_{t=1}^T (16\mathbf{I}_t\boldsymbol{\beta}_t^2 \wedge \frac{1}{4}) + T_\epsilon \\
&\leq \frac{16}{\epsilon^2}(2S^2 + 2\log(3T/\delta)) + T_\epsilon \\
&= \mathcal{O}(\frac{1}{\epsilon^2}(2S^2 + 2\log(3T/\delta))) + T_\epsilon.
\end{aligned}
$$

The proof is completed. $\qquad\square$

**Lemma E.2.** *Suppose the conditions of Theorem 5.1 are satisfied. With probability at least $1 - \delta$, the following result holds:*

$$\mathbf{R}_T \leq \mathcal{O}\left(\epsilon T_\epsilon + \frac{1}{\epsilon}(S^2 + \log(3T/\delta))\right)$$

*Proof.*

$$
\begin{aligned}
\mathbf{R}_T &= \sum_{t=1}^T \mathbb{E}_{\mathbf{x}_t \sim \mathcal{D}_{\mathcal{X}}}\left[\mathbf{h}(\mathbf{x}_t)[\widehat{k}] - \mathbf{h}(\mathbf{x}_t)[k^*]\right] \\
&\leq \sum_{t=1}^T \mathbb{E}_{\mathbf{x}_t \sim \mathcal{D}_{\mathcal{X}}}[|\Delta_t|] \\
&= \sum_{t=1}^T \mathbb{E}_{\mathbf{x}_t \sim \mathcal{D}_{\mathcal{X}}}[|\Delta_t|\mathbb{1}\{|\Delta_t| > \epsilon\}] + \sum_{t=1}^T \mathbb{E}_{\mathbf{x}_t \sim \mathcal{D}_{\mathcal{X}}}[|\Delta_t|\mathbb{1}\{|\Delta_t| \leq \epsilon\}]
\end{aligned}
$$

where the second term is upper bounded by $\epsilon T_\epsilon$. For the first term, we have

$$
\begin{aligned}
&\sum_{t=1}^{T} \mathop{\mathbb{E}}_{\mathbf{x}_t \sim \mathcal{D}_\mathcal{X}} [|\Delta_t| \mathbb{1}\{|\Delta_t| > \epsilon\}]\\
\leq& \frac{1}{\epsilon} \sum_{t=1}^{T} \mathop{\mathbb{E}}_{\mathbf{x}_t \sim \mathcal{D}_\mathcal{X}} [|\Delta_t|^2] \wedge \epsilon\\
\overset{(a)}{\leq}& \frac{1}{\epsilon} \sum_{t=1}^{T} \mathop{\mathbb{E}}_{\mathbf{x}_t \sim \mathcal{D}_\mathcal{X}} [|\Delta_t|^2 \mathbb{1}\{\Delta_t \leq 2\boldsymbol{\beta}_t\}] \wedge \epsilon + \frac{1}{\epsilon} \sum_{t=1}^{T} \mathop{\mathbb{E}}_{\mathbf{x}_t \sim \mathcal{D}_\mathcal{X}} [|\Delta_t|^2 \mathbb{1}\{\Delta_t > 2\boldsymbol{\beta}_t\}] \wedge \epsilon\\
\leq& \frac{1}{\epsilon} \sum_{t=1}^{T} 4\boldsymbol{\beta}_t^2 \wedge \frac{1}{2}\\
\leq& \mathcal{O}\left(\frac{1}{\epsilon}(2S^2 + 2\log(3T/\delta))\right)
\end{aligned}
$$

where the second term in $(a)$ is zero based on the Lemma D.13. The proof is completed. $\qquad\square$

Then, because $\mathbf{x}_1, ..., \mathbf{x}_T$ are generated i.i.d. with Assumption E.1. Then, applying Lemma 23 in [58], with probability at least $1 - \delta$, the result holds:

$$
T_\epsilon \leq 3T\epsilon^\alpha + \mathcal{O}(\log \frac{\log T}{\delta}).
$$

Using the above bound of $T_\epsilon$ back into both Lemma E.1 and Lemma E.2, and then optimizing over $\epsilon$ in the two bounds separately complete the proof:

$$
\begin{aligned}
\mathbf{R}_T &\leq \mathcal{O}((S^2 + \log(3T/\delta))^{\frac{\alpha+1}{\alpha+2}} T^{\frac{1}{\alpha+2}}),\\
\mathbf{N}_T &\leq \mathcal{O}((S^2 + \log(3T/\delta))^{\frac{\alpha}{\alpha+2}} T^{\frac{2}{\alpha+2}}).
\end{aligned}
$$

$\qquad\square$

# F  ANALYSIS FOR POOL-BASED SETTING

Define $\mathcal{L}_t(\boldsymbol{\theta}_t) = \|\mathbf{f}(\mathbf{x}_t; \boldsymbol{\theta}_t) - \mathbf{u}_t\|_2^2/2$, $\mathcal{L}_{t,k}(\boldsymbol{\theta}_t) = (\mathbf{f}(\mathbf{x}_t; \boldsymbol{\theta}_t)[k] - \mathbf{u}_t[k])^2/2$, and $\mathbf{u}_t = [\ell(\mathbf{y}_{t,1}, \mathbf{y}_t), \dots, l(\mathbf{y}_{t,K}, \mathbf{y}_t)]$.

**Lemma F.1** (Almost Convexity). *With probability at least $1 - \mathcal{O}(TL^2K) \exp[-\Omega(m\omega^{2/3}L)]$ over random initialization, for all $t \in [T]$, and $\boldsymbol{\theta}, \boldsymbol{\theta}'$ satisfying $\|\boldsymbol{\theta} - \boldsymbol{\theta}_1\|_2 \leq \omega$ and $\|\boldsymbol{\theta}' - \boldsymbol{\theta}_1\|_2 \leq \omega$ with $\omega \leq \mathcal{O}(L^{-6}[\log m]^{-3/2})$, it holds uniformly that*

$$
\mathcal{L}_t(\boldsymbol{\theta}') \geq \mathcal{L}_t(\boldsymbol{\theta})/2 + \sum_{k=1}^{K} \langle \nabla_{\boldsymbol{\theta}} \mathcal{L}_{t,k}(\boldsymbol{\theta}), \boldsymbol{\theta}' - \boldsymbol{\theta} \rangle - \epsilon.
$$

*where $\omega \leq \mathcal{O}\left((Km\log m)^{-3/8} L^{-9/4} \epsilon^{3/4}\right)$.*

*Proof.* Let $\mathcal{L}_{t,k}(\boldsymbol{\theta})$ be the loss function with respect to $\mathbf{f}(\mathbf{x}_t; \boldsymbol{\theta}')[k]$. By convexity of $\mathcal{L}_t$, it holds uniformly that

$$\|\mathbf{f}(\mathbf{x}_t; \boldsymbol{\theta}') - \mathbf{u}_t\|_2^2/2 - \|\mathbf{f}(\mathbf{x}_t; \boldsymbol{\theta}) - \mathbf{u}_t\|_2^2/2$$

$$\geq \sum_{k=1}^{K} \mathcal{L}'_{t,k}(\boldsymbol{\theta}) \left( \mathbf{f}(\mathbf{x}_t; \boldsymbol{\theta}')[k] - \mathbf{f}(\mathbf{x}_t; \boldsymbol{\theta})[k] \right)$$

$$\overset{(b)}{\geq} \sum_{k=1}^{K} \mathcal{L}'_{t,k}(\boldsymbol{\theta}) \langle \nabla \mathbf{f}(\mathbf{x}_t; \boldsymbol{\theta})[k], \boldsymbol{\theta}' - \boldsymbol{\theta} \rangle$$

$$- \sum_{k=1}^{K} \left| \mathcal{L}'_{t,k}(\boldsymbol{\theta}) \cdot [\mathbf{f}(\mathbf{x}_t; \boldsymbol{\theta}')[k] - \mathbf{f}(\mathbf{x}_t; \boldsymbol{\theta})[k] - \langle \nabla \mathbf{f}(\mathbf{x}_t; \boldsymbol{\theta})[k], \boldsymbol{\theta}' - \boldsymbol{\theta} \rangle] \right|$$

$$\overset{(c)}{\geq} \sum_{k=1}^{K} \langle \nabla_{\boldsymbol{\theta}} \mathcal{L}_{t,k}(\boldsymbol{\theta}), \boldsymbol{\theta}' - \boldsymbol{\theta} \rangle$$

$$- \sum_{k=1}^{K} \left( |\mathbf{f}(\mathbf{x}_t; \boldsymbol{\theta})[k] - \mathbf{u}_t[k]| \cdot |\mathbf{f}(\mathbf{x}_t; \boldsymbol{\theta}')[k] - \mathbf{f}(\mathbf{x}_t; \boldsymbol{\theta})[k] - \langle \nabla \mathbf{f}(\mathbf{x}_t; \boldsymbol{\theta})[k], \boldsymbol{\theta}' - \boldsymbol{\theta} \rangle | \right)$$

$$\overset{(d)}{\geq} \sum_{k=1}^{K} \langle \nabla_{\boldsymbol{\theta}} \mathcal{L}_{t,k}(\boldsymbol{\theta}), \boldsymbol{\theta}' - \boldsymbol{\theta} \rangle - \mathcal{O} \left( \frac{\omega^{4/3} L^3 \sqrt{m \log m}}{\sqrt{K}} \right) \cdot \sum_{k=1}^{K} |\mathbf{f}(\mathbf{x}_t; \boldsymbol{\theta})[k] - \mathbf{u}_t[k]|$$

$$\overset{(e)}{\geq} \sum_{k=1}^{K} \langle \nabla_{\boldsymbol{\theta}} \mathcal{L}_{t,k}(\boldsymbol{\theta}), \boldsymbol{\theta}' - \boldsymbol{\theta} \rangle - \mathcal{O} \left( \frac{\omega^{4/3} L^3 \sqrt{m \log m}}{\sqrt{K}} \right) \cdot \sum_{k=1}^{K} \left( |\mathbf{f}(\mathbf{x}_t; \boldsymbol{\theta})[k] - \mathbf{u}_t[k]|^2 + \frac{1}{4} \right)$$

where $(a)$ is due to the convexity of $\mathcal{L}_t$, $(b)$ is an application of triangle inequality, $(c)$ is because of the Cauchy–Schwarz inequality, and $(d)$ is the application of Lemma D.4 and Lemma D.3, $(e)$ is by the fact $x \leq x^2 + \frac{1}{4}$. Therefore, the results hold

$$\|\mathbf{f}(\mathbf{x}_t; \boldsymbol{\theta}') - \mathbf{u}_t\|_2^2/2 - \|\mathbf{f}(\mathbf{x}_t; \boldsymbol{\theta}) - \mathbf{u}_t\|_2^2/2$$

$$\leq \sum_{k=1}^{K} \langle \nabla_{\boldsymbol{\theta}} \mathcal{L}_{t,k}(\boldsymbol{\theta}), \boldsymbol{\theta}' - \boldsymbol{\theta} \rangle - \mathcal{O} \left( \frac{\omega^{4/3} L^3 \sqrt{m \log m}}{\sqrt{K}} \right) \cdot \|\mathbf{f}(\mathbf{x}_t; \boldsymbol{\theta}) - \mathbf{u}_t\|_2^2$$

$$- \left( \frac{\omega^{4/3} L^3 \sqrt{m \log m}}{\sqrt{K}} \right) \cdot \frac{K}{4}$$

$$\overset{(a)}{\leq} \sum_{k=1}^{K} \langle \nabla_{\boldsymbol{\theta}} \mathcal{L}_{t,k}(\boldsymbol{\theta}), \boldsymbol{\theta}' - \boldsymbol{\theta} \rangle - \frac{1}{4} \cdot \|\mathbf{f}(\mathbf{x}_t; \boldsymbol{\theta}) - \mathbf{u}_t\|_2^2$$

$$- \left( \frac{\omega^{4/3} L^3 \sqrt{m \log m}}{\sqrt{K}} \right) \cdot \frac{K}{4}$$

where $(a)$ holds when $\omega \leq \mathcal{O} \left( 4^{-3/4} K^{3/8} (m \log m)^{-3/8} L^{-9/4} \right)$.

In the end, when $\omega \leq \mathcal{O} \left( 4^{-3/4} (Km \log m)^{-3/8} L^{-9/4} \epsilon^{3/4} \right)$, the result hold:

$$\|\mathbf{f}(\mathbf{x}_t; \boldsymbol{\theta}') - \mathbf{u}_t\|_2^2/2 - \|\mathbf{f}(\mathbf{x}_t; \boldsymbol{\theta}) - \mathbf{u}_t\|_2^2/4$$

$$\leq \sum_{k=1}^{K} \langle \nabla_{\boldsymbol{\theta}} \mathcal{L}_{t,k}(\boldsymbol{\theta}), \boldsymbol{\theta}' - \boldsymbol{\theta} \rangle - \epsilon.$$

The proof is completed. $\qquad\square$

**Lemma F.2.** *With the probability at least $1 - \mathcal{O}(TL^2 K) \exp[-\Omega(m\omega^{2/3} L)]$ over random initialization, for all $t \in [T]$, $\|\boldsymbol{\theta}' - \boldsymbol{\theta}_1\| \leq \omega$, and $\omega \leq \mathcal{O}(L^{-9/2}[\log m]^{-3})$, the results hold:*

$$\mathbf{f}(\mathbf{x}_t; \boldsymbol{\theta}') \leq \log m$$

*Proof.* Using Lemma 7.1 in [3], we have $\mathbf{g}_{t,L-1} \leq \mathcal{O}(1)$, Then, Based on the randomness of $\mathbf{W}_L$, the result hold: $\mathbf{f}(\mathbf{x}_t; \boldsymbol{\theta}') \leq \log m$ (analogical analysis as Lemma 7.1). Using Lemma 8.2 in [3], we have $\mathbf{g}'_{t,L-1} \leq \mathcal{O}(1)$, and thus $\mathbf{f}(\mathbf{x}_t; \boldsymbol{\theta}') \leq \log m$ □

**Lemma F.3** (Loss Bound). *With probability at least $1 - \mathcal{O}(TL^2K)\exp[-\Omega(m\omega^{2/3}L)]$ over random initialization, it holds that*

$$\sum_{t=1}^{T} \|\mathbf{f}(\mathbf{x}_t; \boldsymbol{\theta}_t) - \mathbf{u}_t\|_2^2 / 2 \leq \inf_{\boldsymbol{\theta}' \in \mathcal{B}(\boldsymbol{\theta}_{0;})} \sum_{t=1}^{T} \|\mathbf{f}(\mathbf{x}_t; \boldsymbol{\theta}') - \mathbf{u}_t\|_2^2 + 4LKR^2 \tag{F.1}$$

*where $\boldsymbol{\theta}^* = \arg\inf_{\boldsymbol{\theta} \in \mathcal{B}(\boldsymbol{\theta}_1, R)} \sum_{t=1}^{T} \mathcal{L}_t(\boldsymbol{\theta})$.*

*Proof.* Let $\omega \leq \mathcal{O}\left((Km\log m)^{-3/8}L^{-9/4}\epsilon^{3/4}\right)$, such that the conditions of Lemma D.5 are satisfied.

The proof follows a simple induction. Obviously, $\boldsymbol{\theta}_1$ is in $\mathcal{B}(\boldsymbol{\theta}_1, R)$. Suppose that $\boldsymbol{\theta}_1, \boldsymbol{\theta}_2, \ldots, \boldsymbol{\theta}_T \in \mathcal{B}(\boldsymbol{\theta}_1, R)$. We have, for any $t \in [T]$,

$$\|\boldsymbol{\theta}_T - \boldsymbol{\theta}_1\|_2 \leq \sum_{t=1}^{T} \|\boldsymbol{\theta}_{t+1} - \boldsymbol{\theta}_t\|_2 \leq \sum_{t=1}^{T} \eta \|\nabla \mathcal{L}_t(\boldsymbol{\theta}_t)\|_2$$

$$\leq \eta \cdot \sum_{t=1}^{T} \|\mathbf{f}(\mathbf{x}_t; \boldsymbol{\theta}_t) - \mathbf{u}_t\|_2 \cdot \|\nabla_{\boldsymbol{\theta}_t} \mathbf{f}(\mathbf{x}_t; \boldsymbol{\theta}_t)\|_2$$

$$\leq \eta \cdot \sqrt{Lm} \sum_{t=1}^{T} \|\mathbf{f}(\mathbf{x}_t; \boldsymbol{\theta}_t) - \mathbf{u}_t\|_2$$

$$\overset{(a)}{\leq} \eta \cdot \sqrt{Lm}T \cdot \log m$$

$$\overset{(b)}{\leq} \omega$$

where $(a)$ is applying Lemma F.2 and $(b)$ holds when $m \geq \widetilde{\Omega}(T^8K^{-1}L^{14}\epsilon^{-6})$.

Moreover, when $m > \widetilde{\Omega}(R^8T^8K^3L^{18}\epsilon^{-6})$, it leads to $\frac{R}{\sqrt{m}} \leq \omega$. In round $t$, based on Lemma D.5, for any $\|\boldsymbol{\theta}_t - \boldsymbol{\theta}'\|_2 / 2 \leq \omega$, it holds uniformly

$$\|\mathbf{f}(\mathbf{x}_t; \boldsymbol{\theta}_t) - \mathbf{u}_t\|_2^2 / 4 - \|\mathbf{f}(\mathbf{x}_t; \boldsymbol{\theta}') - \mathbf{u}_t\|_2^2 / 2 \leq \sum_{k=1}^{K} \langle \nabla_{\boldsymbol{\theta}} \mathcal{L}_{t,k}(\boldsymbol{\theta}_t), \boldsymbol{\theta}_t - \boldsymbol{\theta}' \rangle + \epsilon.$$

Then, it holds uniformly

$$\|\mathbf{f}(\mathbf{x}_t; \boldsymbol{\theta}_t) - \mathbf{u}_t\|_2^2 / 4 - \|\mathbf{f}(\mathbf{x}_t; \boldsymbol{\theta}') - \mathbf{u}_t\|_2^2 / 2$$

$$\overset{(a)}{\leq} \frac{\langle \boldsymbol{\theta}_t - \boldsymbol{\theta}_{t+1}, \boldsymbol{\theta}_t - \boldsymbol{\theta}' \rangle}{\eta} + \epsilon$$

$$\overset{(b)}{=} \frac{\|\boldsymbol{\theta}_t - \boldsymbol{\theta}'\|_2^2 + \|\boldsymbol{\theta}_t - \boldsymbol{\theta}_{t+1}\|_2^2 - \|\boldsymbol{\theta}_{t+1} - \boldsymbol{\theta}'\|_2^2}{2\eta} + \epsilon$$

$$\overset{(c)}{\leq} \frac{\|\boldsymbol{\theta}_t - \boldsymbol{\theta}'\|_2^2 - \|\boldsymbol{\theta}_{t+1} - \boldsymbol{\theta}'\|_2^2}{2\eta} + \eta \|\sum_{k=1}^{K} \nabla_{\boldsymbol{\theta}_t} \mathcal{L}_{t,k}(\boldsymbol{\theta}_t)\|_F^2 + \epsilon$$

$$\leq \frac{\|\boldsymbol{\theta}_t - \boldsymbol{\theta}'\|_2^2 - \|\boldsymbol{\theta}_{t+1} - \boldsymbol{\theta}'\|_2^2}{2\eta} + \eta \|\sum_{k=1}^{K} (\mathbf{f}(\mathbf{x}_t; \boldsymbol{\theta}_t)[k] - \mathbf{u}_t[k]) \cdot \nabla_{\boldsymbol{\theta}_t} \mathbf{f}(\mathbf{x}_t; \boldsymbol{\theta}_t)[k]\|_2^2 + \epsilon$$

$$\leq \frac{\|\boldsymbol{\theta}_t - \boldsymbol{\theta}'\|_2^2 - \|\boldsymbol{\theta}_{t+1} - \boldsymbol{\theta}'\|_2^2}{2\eta} + \eta mL \sum_{k=1}^{K} |(\mathbf{f}(\mathbf{x}_t; \boldsymbol{\theta}_t)[k] - \mathbf{u}_t[k])|^2 + \epsilon$$

$$\leq \frac{\|\boldsymbol{\theta}_t - \boldsymbol{\theta}'\|_2^2 - \|\boldsymbol{\theta}_{t+1} - \boldsymbol{\theta}'\|_2^2}{2\eta} + \eta mLK \|\mathbf{f}(\mathbf{x}_t; \boldsymbol{\theta}_t) - \mathbf{u}_t\|_2^2 + \epsilon$$

where $(a)$ is because of the definition of gradient descent, $(b)$ is due to the fact $2\langle A, B\rangle = \|A\|_F^2 + \|B\|_F^2 - \|A - B\|_F^2$, $(c)$ is by $\|\boldsymbol{\theta}_t - \boldsymbol{\theta}_{t+1}\|_2^2 = \|\eta\nabla_{\boldsymbol{\theta}}\mathcal{L}_t(\boldsymbol{\theta}_t)\|_2^2 \leq \mathcal{O}(\eta^2 KLm)$.

Then, for $T$ rounds, we have

$$\sum_{t=1}^{T}\|\mathbf{f}(\mathbf{x}_t;\boldsymbol{\theta}_t) - \mathbf{u}_t\|_2^2/4 - \sum_{t=1}^{T}\|\mathbf{f}(\mathbf{x}_t;\boldsymbol{\theta}') - \mathbf{u}_t\|_2^2/2$$

$$\overset{(a)}{\leq}\frac{\|\boldsymbol{\theta}_1 - \boldsymbol{\theta}'\|_2^2}{2\eta} + \sum_{t=2}^{T}\|\boldsymbol{\theta}_t - \boldsymbol{\theta}'\|_2^2(\frac{1}{2\eta} - \frac{1}{2\eta}) + \sum_{t=1}^{T}\eta mLK\|\mathbf{f}(\mathbf{x}_t;\boldsymbol{\theta}_t) - \mathbf{u}_t\|_2^2 + T\epsilon$$

$$\leq\frac{\|\boldsymbol{\theta}_1 - \boldsymbol{\theta}'\|_2^2}{2\eta} + \sum_{t=1}^{T}\eta mLK\|\mathbf{f}(\mathbf{x}_t;\boldsymbol{\theta}_t) - \mathbf{u}_t\|_2^2 + T\epsilon$$

$$\leq\frac{R^2}{2m\eta} + \eta mLK\sum_{t=1}^{T}\|\mathbf{f}(\mathbf{x}_t;\boldsymbol{\theta}_t) - \mathbf{u}_t\|_2^2 + T\epsilon$$

$$\leq 5R^2 LK + \sum_{t=1}^{T}\|\mathbf{f}(\mathbf{x}_t;\boldsymbol{\theta}_t) - \mathbf{u}_t\|_2^2/8$$

where $(a)$ is by simply discarding the last term and $(b)$ is setting by $\eta = \frac{1}{mLK}$ and $\epsilon = \frac{KR^2L}{T}$. Based on the above inequality, taking the infimum over $\boldsymbol{\theta}' \in \mathcal{B}(\boldsymbol{\theta}_\infty, \mathcal{R})$, the results hold :

$$\sum_{t=1}^{T}\|\mathbf{f}(\mathbf{x}_t;\boldsymbol{\theta}_t) - \mathbf{u}_t\|_2^2/8 \leq \inf_{\boldsymbol{\theta}' \in \mathcal{B}(\boldsymbol{\theta}_\infty, \mathcal{R})}\sum_{t=1}^{T}\|\mathbf{f}(\mathbf{x}_t;\boldsymbol{\theta}') - \mathbf{u}_t\|_2^2/2 + 5LKR^2.$$

The proof is completed. $\square$

**Lemma F.4.** *Let $R = S' = \sqrt{\mathbf{u}^\top\mathbf{h}^{-1}\mathbf{u}}$. With probability at least $1 - \mathcal{O}(TL^2K)\exp[-\Omega(m\omega^{2/3}L)]$, the result holds that*

$$\inf_{\boldsymbol{\theta}\in\mathcal{B}(\boldsymbol{\theta}_0;R)}\sum_{t=1}^{T}\mathcal{L}_t(\boldsymbol{\theta}) \leq \mathcal{O}(LK)$$

*Proof.* Let $\mathbf{G} = [\nabla_{\boldsymbol{\theta}_0}\mathbf{f}(\mathbf{x}_1;\boldsymbol{\theta}_0)[1], \nabla_{\boldsymbol{\theta}_0}\mathbf{f}(\mathbf{x}_1;\boldsymbol{\theta}_0)[2], \ldots, \nabla_{\boldsymbol{\theta}_0}\mathbf{f}(\mathbf{x}_T;\boldsymbol{\theta}_0)[K]]/\sqrt{m} \in \mathbb{R}^{p\times TK}$. Suppose the singular value decomposition of $\mathbf{G}$ is $\mathbf{PAQ}^\top$, $\mathbf{P} \in \mathbb{R}^{p\times TK}, \mathbf{A} \in \mathbb{R}^{TK\times TK}, \mathbf{Q} \in \mathbb{R}^{TK\times TK}$, then, $\mathbf{A} \succeq 0$.

Define $\boldsymbol{\theta}^* = \boldsymbol{\theta}_0 + \mathbf{PA}^{-1}\mathbf{Q}^\top\mathbf{u}/\sqrt{m}$. Then, we have

$$\sqrt{m}\mathbf{G}^\top(\boldsymbol{\theta}^* - \boldsymbol{\theta}_0) = \mathbf{QAP}^\top\mathbf{PA}^{-1}\mathbf{Q}^\top\mathbf{u} = \mathbf{u}.$$

This leads to

$$\sum_{t=1}^{T}\sum_{k=1}^{K}|\mathbf{u}_t[k] - \langle\nabla_{\boldsymbol{\theta}_0}\mathbf{f}(\mathbf{x}_t;\boldsymbol{\theta}_0)[k], \boldsymbol{\theta}^* - \boldsymbol{\theta}_0\rangle| = 0.$$

Therefore, the result holds:

$$\|\boldsymbol{\theta}^* - \boldsymbol{\theta}_0\|_2^2 = \mathbf{u}^\top\mathbf{QA}^{-2}\mathbf{Q}^\top\mathbf{u}/m = \mathbf{u}^\top(\mathbf{G}^\top\mathbf{G})^{-1}\mathbf{u}/m \leq 2\mathbf{u}^\top\mathbf{H}^{-1}\mathbf{u}/m \qquad \text{(F.2)}$$

Based on Lemma D.4 and initialize $(\mathbf{x_t}; \boldsymbol{\theta_0}) = \mathbf{0}, \mathbf{t} \in [\mathbf{T}]$, we have

$$
\begin{aligned}
\sum_{t=1}^{T} \mathcal{L}_t(\boldsymbol{\theta}^*) &\leq \sum_{t=1}^{T} \sum_{k=1}^{K} \left( \mathbf{u}_t[k] - \langle \nabla_{\boldsymbol{\theta}_0} \mathbf{f}(\mathbf{x}_t; \boldsymbol{\theta}_0)[k], \boldsymbol{\theta}^* - \boldsymbol{\theta}_0 \rangle - \mathcal{O}(\omega^{4/3} L^3 \sqrt{m \log(m)}) \right)^2 \\
&\leq \sum_{t=1}^{T} \sum_{k=1}^{K} (\mathbf{u}_t[k] - \langle \nabla_{\boldsymbol{\theta}_0} \mathbf{f}(\mathbf{x}_t; \boldsymbol{\theta}_0)[k], \boldsymbol{\theta}^* - \boldsymbol{\theta}_0 \rangle)^2 + TK \cdot \mathcal{O}(\omega^{8/3} L^6 m \log(m)) \\
&\leq \sum_{t=1}^{T} \sum_{k=1}^{K} (\mathbf{u}_t[k] - \langle \nabla_{\boldsymbol{\theta}_0} \mathbf{f}(\mathbf{x}_t; \boldsymbol{\theta}_0)[k], \boldsymbol{\theta}^* - \boldsymbol{\theta}_0 \rangle)^2 + TK \cdot \mathcal{O}((S'/m^{1/2})^{8/3} L^6 m \log(m)) \\
&\leq \sum_{t=1}^{T} \sum_{k=1}^{K} (\mathbf{u}_t[k] - \langle \nabla_{\boldsymbol{\theta}_0} \mathbf{f}(\mathbf{x}_t; \boldsymbol{\theta}_0)[k], \boldsymbol{\theta}^* - \boldsymbol{\theta}_0 \rangle)^2 + \mathcal{O}(LK) \\
&\leq \mathcal{O}(LK)
\end{aligned}
$$

Because $\boldsymbol{\theta}^* \in \mathcal{B}(\boldsymbol{\theta}_0, S')$, the proof is completed. $\qquad \square$

**Lemma F.5.** *Suppose $m, \eta_1, \eta_2$ satisfy the conditions in Theorem 5.4. With probability at least $1 - \delta$, for all $t \in [Q]$, it holds uniformly that*

$$
\frac{1}{t} \sum_{\tau=1}^{t} \left[ \| \mathbf{f}_2(\phi(\mathbf{x}_\tau); \boldsymbol{\theta}_\tau^2) - (\mathbf{h}(\mathbf{x}_\tau) - \mathbf{f}_1(\mathbf{x}_\tau; \boldsymbol{\theta}_\tau^1)) \|_2 \wedge 1 \right] \leq \sqrt{\frac{\mathcal{O}(LKS^2)}{t}} + 2\sqrt{\frac{2 \log(1/\delta)}{t}} \quad \text{(F.3)}
$$

$\mathbf{u}_t = (\ell(\mathbf{y}_{t,1}, \mathbf{y}_t), \dots, \ell(\mathbf{y}_{t,K}, \mathbf{y}_t))^\top$, $\mathcal{H}_{t-1} = \{\mathbf{x}_\tau, \mathbf{y}_\tau\}_{\tau=1}^{t-1}$ *is historical data, and the expectation is also taken over $(\boldsymbol{\theta}_t^1, \boldsymbol{\theta}_t^2)$.*

*Proof.* Then, applying the Hoeffding-Azuma inequality as same as in Lemma D.6, we have

$$
\begin{aligned}
&\frac{1}{t} \sum_{\tau=1}^{t} \left[ \| \mathbf{f}_2(\phi(\mathbf{x}_\tau); \boldsymbol{\theta}_\tau^2) - (\mathbf{h}(\mathbf{x}_\tau) - \mathbf{f}_1(\mathbf{x}_\tau; \boldsymbol{\theta}_\tau^1)) \|_2 \wedge 1 \right] \\
&\leq \frac{1}{t} \sum_{\tau=1}^{t} \| \mathbf{f}_2(\phi(\mathbf{x}_\tau); \boldsymbol{\theta}_\tau^2) - (\mathbf{u}_\tau - \mathbf{f}_1(\mathbf{x}_\tau; \boldsymbol{\theta}_\tau^1)) \|_2 + \sqrt{\frac{2 \log(1/\delta)}{t}} \\
&\leq \frac{1}{t} \sqrt{ t \sum_{\tau=1}^{t} \| \mathbf{f}_2(\phi(\mathbf{x}_\tau); \boldsymbol{\theta}_\tau^2) - (\mathbf{u}_\tau - \mathbf{f}_1(\mathbf{x}_\tau; \boldsymbol{\theta}_\tau^1)) \|_2^2 } + \sqrt{\frac{2 \log(1/\delta)}{t}} \quad \text{(F.4)} \\
&\overset{(a)}{\leq} \sqrt{\frac{\mathcal{O}(LKS'^2)}{t}} + \sqrt{\frac{2 \log(1/\delta)}{t}} \\
&\overset{(b)}{\leq} \sqrt{\frac{\mathcal{O}(LKS^2)}{t}} + 2\sqrt{\frac{2 \log(1/\delta)}{t}}
\end{aligned}
$$

where $(a)$ is an application of Lemma F.3 and Lemma F.4 and $(b)$ is applying the Hoeffding-Azuma inequality again on $S'$. The proof is complete. $\qquad \square$

**Lemma F.6.** *Suppose $m, \eta_1, \eta_2$ satisfy the conditions in Theorem 5.4. With probability at least $1 - \delta$, the result holds:*

$$
\begin{aligned}
\sum_{t=1}^{Q} \sum_{\mathbf{x}_i \in \mathbf{P}_t} p_i (\mathbf{h}(\mathbf{x}_i)[\widehat{k}] - \mathbf{h}(\mathbf{x}_i)[k^*]) &\leq \frac{\gamma}{4} \sum_{t=1}^{Q} \sum_{\mathbf{x}_i \in \mathbf{P}_t} p_i (\mathbf{f}(\mathbf{x}_i; \boldsymbol{\theta}_t)[k^*] - \mathbf{h}(\mathbf{x}_i)[k^*])^2 + \frac{Q}{\gamma} \\
&\quad + \sqrt{QKS^2} + O(\log(1/\delta))
\end{aligned}
$$

*Proof.* For some $\eta > 0$, we have

$$\sum_{t=1}^{Q} \sum_{\mathbf{x}_i \in \mathbf{P}_t} p_i(\mathbf{h}(\mathbf{x}_i)[\widehat{k}] - \mathbf{h}(\mathbf{x}_i)[k^*]) - \eta \sum_{t=1}^{Q} \sum_{\mathbf{x}_i \in \mathbf{P}_t} p_i(\mathbf{f}(\mathbf{x}_i; \boldsymbol{\theta}_t)[k^*] - \mathbf{h}(\mathbf{x}_i)[k^*])^2$$

$$= \sum_{t=1}^{Q} \sum_{\mathbf{x}_i \in \mathbf{P}_t} p_i[(\mathbf{h}(\mathbf{x}_i)[\widehat{k}] - \mathbf{h}(\mathbf{x}_i)[k^*]) - \eta(\mathbf{f}(\mathbf{x}_i; \boldsymbol{\theta}_t)[k^*] - \mathbf{h}(\mathbf{x}_i)[k^*])^2]$$

$$= \sum_{t=1}^{Q} \sum_{\mathbf{x}_i \in \mathbf{P}_t} p_i[(\mathbf{h}(\mathbf{x}_i)[\widehat{k}] - \mathbf{f}(\mathbf{x}_i; \boldsymbol{\theta}_t)[\widehat{k}]) + (\mathbf{f}(\mathbf{x}_i; \boldsymbol{\theta}_t)[\widehat{k}] - \mathbf{h}(\mathbf{x}_i)[k^*]) - \eta(\mathbf{f}(\mathbf{x}_i; \boldsymbol{\theta}_t)[k^*] - \mathbf{h}(\mathbf{x}_i)[k^*])^2]$$

$$\leq \sum_{t=1}^{Q} \sum_{\mathbf{x}_i \in \mathbf{P}_t} p_i[(\mathbf{h}(\mathbf{x}_i)[\widehat{k}] - \mathbf{f}(\mathbf{x}_i; \boldsymbol{\theta}_t)[\widehat{k}]) + (\mathbf{f}(\mathbf{x}_i; \boldsymbol{\theta}_t)[k^*] - \mathbf{h}(\mathbf{x}_i)[k^*]) - \eta(\mathbf{f}(\mathbf{x}_i; \boldsymbol{\theta}_t)[k^*] - \mathbf{h}(\mathbf{x}_i)[k^*])^2]$$

$$\overset{(a)}{\leq} \sum_{t=1}^{Q} \sum_{\mathbf{x}_i \in \mathbf{P}_t} p_i(\mathbf{h}(\mathbf{x}_i)[\widehat{k}] - \mathbf{f}(\mathbf{x}_i; \boldsymbol{\theta}_t)[\widehat{k}]) + \frac{Q}{4\eta}$$

$$\overset{(b)}{\leq} \sum_{t=1}^{Q} \|\mathbf{u}_t - \mathbf{f}(\mathbf{x}_t; \boldsymbol{\theta}_t)\|_2 + O(\log(1/\delta)) + \frac{Q}{4\eta}$$

$$\overset{(c)}{\leq} \sqrt{QKS^2} + O(\log(1/\delta)) + \frac{Q}{4\eta}$$

where (a) is an application of AM-GM: $x - \eta x^2 \leq \frac{1}{\eta}$ and (b) is application of Lemma F.7. (c) is based on Lemma F.5. Then, replacing $\eta$ by $\frac{\gamma}{4}$ completes the proof. $\qquad\square$

**Lemma F.7.** *(Lemma 2, [25]) With probability $1 - \delta$, the result holds:*

$$\sum_{t=1}^{Q} \sum_{\mathbf{x}_i \in \mathbf{P}_t} p_i \|\mathbf{f}(\mathbf{x}_i; \boldsymbol{\theta}_t) - \mathbf{h}(\mathbf{x}_i)\|_2^2 \leq 2 \sum_{t=1}^{Q} \|\mathbf{f}(\mathbf{x}_t; \boldsymbol{\theta}_t) - \mathbf{u}_t\|_2^2 + O(\log(1/\delta)).$$

Finally, we show the proof of Theorem 5.4.

*Proof.* Assume the event in F.7 holds. We have

$$\mathbf{R}_{pool}(Q) = \sum_{t=1}^{Q} \left[ \underset{(\mathbf{x}_t, \mathbf{y}_t) \sim \mathcal{D}}{\mathbb{E}} [\ell(\mathbf{y}_{t, \widehat{k}}, \mathbf{y}_t)] - \underset{(\mathbf{x}_t, \mathbf{y}_t) \sim D}{\mathbb{E}} [\ell(\mathbf{y}_{t, k^*}, \mathbf{y}_t)] \right]$$

$$= \sum_{t=1}^{Q} \left[ \underset{\mathbf{x}_t \sim \mathcal{D}_{\mathcal{X}}}{\mathbb{E}} [\mathbf{h}(\mathbf{x}_t)[\widehat{k}]] - \underset{\mathbf{x}_t \sim \mathcal{D}_{\mathcal{X}}}{\mathbb{E}} [\mathbf{h}(\mathbf{x}_t)[k^*]] \right]$$

$$\overset{(a)}{\leq} \sum_{t=1}^{Q} \mathbb{E}[\mathbf{h}(\mathbf{x}_t)[\widehat{k}] - \mathbf{h}(\mathbf{x}_t)[k^*]] + \sqrt{2Q \log(2/\delta)}$$

$$\leq \sum_{t=1}^{Q} \sum_{\mathbf{x}_i \in \mathbf{P}_t} p_i(\mathbf{h}(\mathbf{x}_i)[\widehat{k}] - \mathbf{h}(\mathbf{x}_i)[k^*]) + \sqrt{2Q \log(2/\delta)}$$

where $(a)$ is an application of Azuma-Hoeffding inequality.

Next, applying Lemma F.6 and Letting $\xi = \sqrt{QKS^2} + O(\log(1/\delta))$, we have

$$
\begin{aligned}
\mathbf{R}_{pool}(Q) &\overset{(a)}{\leq} \frac{\gamma}{4} \sum_{t=1}^{Q} \sum_{\mathbf{x}_i \in \mathbf{P}_t} p_i(\mathbf{f}(\mathbf{x}_i; \boldsymbol{\theta}_t)[k^*] - \mathbf{h}(\mathbf{x}_i)[k^*])^2 + \frac{Q}{\gamma} + \sqrt{2Q\log(2/\delta)} + \xi \\
&\leq \frac{\gamma}{4} \sum_{t=1}^{Q} \sum_{\mathbf{x}_i \in \mathbf{P}_t} p_i\|\mathbf{f}(\mathbf{x}_i; \boldsymbol{\theta}_t) - \mathbf{h}(\mathbf{x}_i)\|_2^2 + \frac{Q}{\gamma} + \sqrt{2Q\log(2/\delta)} + \xi \\
&\overset{(b)}{\leq} \frac{\gamma}{2} \sum_{t=1}^{Q} \|\mathbf{f}(\mathbf{x}_i; \boldsymbol{\theta}_t) - \mathbf{u}_t)\|^2 + \gamma\log(\delta^{-1})/4 + \frac{Q}{\gamma} + \sqrt{2Q\log(2/\delta)} + \xi \\
&\overset{(c)}{\leq} \frac{\gamma}{2} K S^2 + \gamma\log(2\delta^{-1})/4 + \frac{Q}{\gamma} + \sqrt{2Q\log(2/\delta)} + \sqrt{QKS^2} + O(\log(1/\delta)) \\
&\leq \sqrt{\mathcal{O}(QKS^2)} + \sqrt{\frac{Q}{KS^2}} \cdot \mathcal{O}(\log(\delta^{-1})) + \mathcal{O}(\sqrt{2Q\log(3/\delta)})
\end{aligned}
$$

where (a) follows from F.6, (b) follows from F.7, (c) is based on Lemma F.5, and we apply the union bound to the last inequality and choose $\gamma = \sqrt{\frac{Q}{KS^2}}$. Therefore:

$$
\mathbf{R}_{pool}(Q) \leq \mathcal{O}(\sqrt{QK}S) + \mathcal{O}\left(\sqrt{\frac{Q}{KS^2}}\right) \cdot \log(\delta^{-1}) + \mathcal{O}(\sqrt{2Q\log(3/\delta)})
$$

The proof is complete.

$\square$

