# OpenReview forum: "Neural Active Learning Beyond Bandits"
_ICLR.cc/2024/Conference — ICLR 2024 poster_

### Official Review · Reviewer_4Q6h · 2023-10-15

**Soundness:** 3 good
**Presentation:** 3 good
**Contribution:** 2 fair
**Rating:** 6
**Confidence:** 3

**Summary:**

The paper investigates active learning strategies using neural networks for both stream-based and pool-based scenarios. It addresses the challenge of mitigating the adverse impacts of the number of classes ($K$) on active learning performance. The authors propose two algorithms that incorporate neural networks for exploration and exploitation in active learning, providing theoretical performance guarantees. Experiments demonstrate that these algorithms consistently outperform existing baselines.

**Strengths:**

The paper is well written and easy to follow.

The paper studies an important and interesting problem of how to do active learning in two settings. The proposed algorithms based on exploitation-exploration NN have some novelty.

I checked parts of the proof and feel they should be correct. While I am not familiar with this line of theoretical work, I would refer to other reviewers' opinions for evaluation.

**Weaknesses:**

I have no major concern on the current manuscript.

On minor issue: I feel it is better if the authors could explain how your work and the previous literature (e.g. bandit-based algorithms) decide whether to observe the label for each sample in detail. How your work is different from the existing literature on this problem?

**Questions:**

Please refer to the above Weaknesses section.

---

> ### Author Response · Authors · 2023-11-17
> **Response to Reviewer 4Q6h**
>
> Thank you very much for your constructive comments and suggestions. We are glad to provide the response to your questions.
>
> **Q**: Difference from existing literature regarding the label-query decision-maker.
>
> **A**: In the stream-based setting, the intuition behind the label decision-maker is similar to the bandit-based approach: We only query labels when we have low confidence to the neural network estimation. However, the crucial difference lies in the construction of confidence bound for the neural network estimation. Bandit-based approaches still adopt the technique in bandit optimization to construct the confidence bound, i.e., they only take the reward of the selected arm (class) for back-propagation and use the ridge regression in the RHKS spanned by NTK to build the ellipsoid confidence ball, causing a looser confidence bound. In contrast, our approach directly takes the label vector for the back-propagation of neural networks and we construct the generalization bound of online gradient descent as the confidence bound. Because our confidence bound is tighter, our label decision-maker is able to make more accurate decisions.
>
> In the pool-based setting, our approach is the first label decision-maker based on the inverse gap weighting strategy to measure the uncertainty from both exploitation and exploration perspectives. In the related literature, their label decision-makers are based on gradient weight [6], gradient change [47], or core-set [42].
>
>
>
> Thank you very much for your time and insightful comments again.

---

> > ### Comment · Reviewer_4Q6h · 2023-11-22
> > **Thank you for your response**
> >
> > I appreciate authors' detailed rebuttal to my concern. I have no more questions.

---

> > > ### Author Response · Authors · 2023-11-22
> > >
> > > Thanks so much for your time and insightful comments and questions!

---

### Official Review · Reviewer_Jpt8 · 2023-10-30

**Soundness:** 3 good
**Presentation:** 3 good
**Contribution:** 2 fair
**Rating:** 6
**Confidence:** 3

**Summary:**

This paper studies active learning using newly designed exploration and exploitation neural networks. Instead of transforming the instance into $K$ arms and then calculating scores for them, they directly input the original instance into the network, hence avoiding the cost of $K$ time forward-propagation and dimension multiplication due to embedding. As a result, they present theoretical guarantees with a slower error-growth rate, concerning $K$. Besides, extensive simulation results are presented.

**Strengths:**

# Origionality
- The idea of exploration-exploitation networks is not so novel but related works are covered in detail. It originates in [10] and is then adopted by [11], which is one of the works this paper is compared to.
- The idea of reducing the dimension to $d$ is novel considering the literature starting from [10].
# Quality
- The theoretical proofs seem to be concrete.
- The experiments are extensive and detailed. They validate the performance and computation efficiency claimed.
# Clarity
- The paper is in general well-written and smooth to follow, with the exception of some proofs in the supplementary material.
# Significance
- The proposed algorithms are of interest as the input is more coherent and the performance is great.

**Weaknesses:**

- The main originality seems to be theoretical proofs rather than network structures. However, the proofs heavily depend on NTK techniques.
- The experiments were conducted only five times, which could potentially impact the reliability of the results. I would appreciate seeing outcomes derived from a greater number of runs.
- The theoretical results are only for the setting where the Tsybakov noise $\alpha = 0$, making it less comparable to existing literature, except when compared to [48] indirectly as $\alpha \rightarrow \infty$.

**Questions:**

- May I ask the reason behind limiting the experiments to five iterations? Was it influenced by computational constraints or other considerations?
- What's the main difficulty of analyzing settings other than $\alpha = 0$? What do the authors expect different levels of $\alpha$ to influence the theoretical results intuitively?

---

> ### Author Response · Authors · 2023-11-17
> **Response to Reviewer Jpt8**
>
> Thank you very much for your constructive comments and inspiring questions.
> We've updated the manuscript to address your concerns and we are glad to provide the response to your questions.
>
> **Q1**: Originality.
>
> **A1**: It might be a little lengthy and please allow us to summarize the novel aspects here for reviewers' convenience to capture the essence of this paper.
>
> 1. As pointed out by the reviewer, one novel contribution comes from the newly designed input and output of neural networks to reduce the input dimension to $d$. In addition to the empirical improvement, this design also brings theoretical improvement. This is because our approach can directly take the label vector for the back-propagation of neural networks, such that we can derive a tighter confidence bound for the estimation based on the generalization bound of online gradient descent. In contrast, [48] only takes the reward of the selected arm (class) for the back-propagation and adopts the technique in bandit optimization to build the confidence bound, which leads to the dependence of $L_{H}$ in their confidence bound.
> 2. Another novel contribution stems from the label-query decision-maker to minimize the label complexity in active learning. For example, in the proposed pool-based algorithm, we are the first to use the Inverse Gap Weighting strategy to select data points in active learning, to measure the uncertainty from both exploitation and exploration perspectives.
> 3. As pointed out by the reviewer, the third novel contribution comes from the theoretical analysis. Our proof technique is very different from previous work, and importantly, we are able to remove the dependence of the intractable complexity term $L_\mathbf{H}$, detailed improvement as follows in Q3.
>
>
>
> **Q2**: More experimental runs.
>
> **A2**: Thanks for the reviewer's suggestion. We've extended our experiments to 10 runs (Table 1 and Table 2). The new results are consistent with the previous results and conclusion.
>
>
> **Q3**: Adapt the analysis to the varying $\alpha$.
>
> **A3**: Thanks for the reviewer's suggestion. We've added the new theoretical results (Section F in the latest manuscript) adapting to the varying $\alpha$. Since the Mammen-Tsybakov noise $\alpha$ assumption and related work [48] only study the binary classification, we present the following results in the binary classification.
>
> The results for our proposed algorithm:
>
> \begin{aligned}
> \mathbf{R}\_{stream}(T) \leq \widetilde{\mathcal{O}}( (S^2 )^{\frac{\alpha+1}{\alpha+2}} T^{\frac{1}{\alpha + 2}}),  \\\\
> \mathbf{N}(T) \leq \widetilde{\mathcal{O}}( (S^2 )^{\frac{\alpha}{\alpha+2}} T^{\frac{2}{\alpha + 2}}).
> \end{aligned}
>
> For comparison, the results in [48]:
>
> \begin{aligned}
> \mathbf{R}\_{stream}(T) & \leq \widetilde{\mathcal{O}} \left ( {L_{\mathbf{H}}}^{\frac{2(\alpha +1)}{\alpha+2}}  T^{\frac{1}{\alpha +2}} \right) +   \widetilde{\mathcal{O}} \left( {L_\mathbf{H}}^{\frac{\alpha +1}{\alpha+2}} (S^2)^{\frac{\alpha+1}{\alpha+2}}   T^{\frac{1}{\alpha +2}} \right),  \\\\
> \mathbf{N}(T) &\leq \widetilde{\mathcal{O}} \left ( {L_\mathbf{H}}^{\frac{2\alpha}{\alpha+2}}  T^{\frac{2}{\alpha +2}} \right) +   \widetilde{\mathcal{O}} \left( {L_\mathbf{H}}^{\frac{\alpha}{\alpha+2}} (S^2)^{\frac{\alpha}{\alpha+2}}   T^{\frac{2}{\alpha +2}} \right).
> \end{aligned}
>
>
>
> For the regret $\mathbf{R}\_{stream}(T)$, compared to [48], our result removes the term $\widetilde{\mathcal{O}} \left ( {L_{\mathbf{H}}}^{\frac{2(\alpha +1)}{\alpha+2}}  T^{\frac{1}{\alpha +2}} \right)$ and further improves the regret upper bound by a multiplicative factor of $ {L_\mathbf{H}}^{\frac{\alpha +1}{\alpha+2}}$. For the label complexity $\mathbf{N}(T)$, compared to [48], our result removes the term
> $\widetilde{\mathcal{O}} \left ( {L_\mathbf{H}}^{\frac{2\alpha}{\alpha+2}}  T^{\frac{2}{\alpha +2}} \right)$ and further improves the regret upper bound by a multiplicative factor of $ {L_\mathbf{H}}^{\frac{\alpha}{\alpha+2}}$. It is noteworthy that  $L_\mathbf{H}$ grows linearly with respect to $T$, i.e., $L_\mathbf{H} \geq T \log (1 + \lambda_0)$.
> Note that the main technical challenges lie in building the confidence bound for the neural network estimation rather than adapting the result to varying $\alpha$.
>
>
>
>
>
> Thank you very much for your time and insightful comments again.

---

> > ### Comment · Reviewer_Jpt8 · 2023-11-22
> >
> > I appreciate the authors' detailed responses, new simulation runs, and theories in the revision. They all look good to me and I increased my rating to 6.

---

> > > ### Author Response · Authors · 2023-11-22
> > >
> > > Thanks very much for your feedback and valuable suggestions, which definitely helped improve our paper!

---

### Official Review · Reviewer_5FNs · 2023-11-02

**Soundness:** 4 excellent
**Presentation:** 3 good
**Contribution:** 4 excellent
**Rating:** 8
**Confidence:** 3

**Summary:**

This paper builds on prior work by Wang et al., 2021 on neural active-learning with performance guarantees. As in that work, the authors construct two neural networks -- an exploitation network and an exploration network which is trained to fit the residuals (noise) of the exploitation network -- and sum their outputs to estimate the expected loss of each class given the context.

On the applied side, what makes this work different from Wang et al.'s is that the neural proposed in this paper jointly predicts the expected losses for all actions (classes) simultaneously. Whereas Wang et al. apply their neural networks to a context+action vector for each class individually. This makes inference K times faster for the authors' new method, where K is the number of classes. Additionally, the inputs to the neural network are K times shorter than in Wang et al. because they don't need to use a kronecker product construction to encode the chosen action (class) on the input side. And the smaller dimensionality also makes everything more efficient.

The authors then propose and analyze two algorithms for active learning which both utilize the signal & noise trained neural networks. The first algorithm is for the stream-based setting in which one example arrives at a time and the learner can either can either request a label or not, and there is a budget imposed on how many labels can be requested. The second algorithm applies to the pool-based setting where a set of examples is presented at each instance and the learner can request a label for one example in the set. For both settings the authors aim to minimize the population cumulative regret, which is the regret of the true / out-of-sample model performance of the learned model accumulated over the course of learning, minus the same for the Bayes optimal classifier.

On the theory side, the authors utilize the neural tangent kernel framework to derive lower and upper bounds on the cumulative population regret for two regimes of Mammen-Tsybakov noise: the hard margin regime and the unrestricted regime. The derived bounds improve those of Wang et al., 2021 by a factor of as much as O(md) where "m" is the network width and "d" is the input dimensionality. The authors also provide regret and label complexity bounds for the case where a unique Bayes optimal classifier exists. All bounds depend on a quantity related to the underlying dimension of the neural tangent kernel and another quantity related to the data.

Finally, the authors present experiments comparing their active learning algorithms (NeurOnAL-S and NeurOnAL-P) for the stream and pool-based settings on several UCI datasets and FashionMNist against other neural active learning methods. The methods they compare against are: I-NeurAL (Wang et al., 2021), ALPS, BADGE, ALBL, CoreSet, DynamicAL. NeurOnAL-S/P generally outperform all others in terms of both test accuracy and running time, making for compelling empirical results.

**Strengths:**

The theoretical bounds for the proposed method are quite an improvement over those for the method presented in Wang et al. 2021. The empirical wins, both on increased test set accuracy and decreased running time, for the stream and pool-based settings also demonstrate siginificant improvement over multiple prior approaches.

**Weaknesses:**

One weakness of the proposed method is that it is demonstrated with a particular neural network structure (an MLP) and the authors do not state whether it can be used with arbitrary network structures, and if it can, then how the bounds might change (I realize this would be very difficult analysis). But some discussion around how general the approach is would make this paper much less niche/narrow.

A related weakness is that the bounds are in terms of quantities (S and L_H) that are difficult for practitioners to quantify or trade off in order to find the network architecture for their problem.

In the experiments, I did not see stated which subsets of data were used to tune hyperparameters. One might therefore worry that hyperparameters were tuned to give good test performance.

**Questions:**

For the pool-based setting, I am interested to know why the authors chose cumulative population regret as their subject of analysis rather than the simple regret? With pool based learning, one typically doesn't care about the performance of the intermediate models constructed during training, only the last model.

I'm also curious about the use of the exploration and exploitation networks and why they are better able to model the signal and the noise. In particular, using the gradients of the exploitation network as inputs to the exploration network -- why does this work? Is there any theoretical explanation (not just intuition) that justifies how this helps fit the noise?

Finally, I am trying to understand the statement in section 5 that

          h(x_t)[k] = <Grad f(x_t, theta*)[k], theta* - theta1>.

Is this an assumption or does such a theta* always exist? And if it always exists, is it the same theta* for all "t" and "k"?

---

> ### Author Response · Authors · 2023-11-17
> **Response to Reviewer 5FNs**
>
> Thank you very much for your detailed and constructive comments and we are glad to provide the response to your questions.
>
>
> **Q1**: Extend to other network structures.
>
> **A1**: Yes, our analysis can be readily converted into many more advanced neural structures, because we only use the almost-convexity property of MLP in over-parameterized networks. Using the results of [3], our results can be directly extended to CNN and ResNet.
>
>
> **Q2**: Value of S.
>
> **A2**: We set $S =1$  in all the experiments (see implementation details on Page 14). Since we have the exploration hyper-parameter $\gamma$ which plays a similar role in tuning the exploration strength, we conducted the grid search for $\gamma$ in the experiments instead.
>
> **Q3**: Cumulative population regret metric.
>
> **A3**: We agree with the reviewer in that the population regret of the last round is also a promising regret metric. The cumulative population regret metric reflects the overall performance of each label query, which implies the average sub-linear decreasing error rate with respect to the number of queries $O(1/\sqrt{Q})$.
>
>
> **Q4**: Advantages of using the exploration network.
>
> **A4**: The main advantage of using the exploration network is to achieve adaptive exploration for "upward" exploration and "downward" exploration. The upward exploration is to describe the case when our model $f$ underestimates the expected reward (label value) $h$ and the downward exploration is to describe the case when $f$ overestimates $h$ (i.e., $f(x) > h(x)$). A motivating example is that, when $f(x)$ overestimates $h(x)$, the UCB-based method will further add a positive value to $f(x)$, making the deviation larger. Instead, the exploration network is able to add a negative value to $f(x)$ by learning and predicting the potential gain $h(x) - f(x)$, making the gap smaller.
>
> The theoretical motivation for this exploration network and using the gradient as input is the statistical confidence bound. The confidence bound can be represented as follows:
> $|f(x) - h(x)| \leq \Phi(g(x))$, where $g(x)$ is the gradient of $f(x)$ and $\Phi$ represents the statistical form with respect to $g(x)$. Thus, instead of deriving a fixed form of $\Phi$, the exploration network uses a network to learn a form of $\Phi$, i.e., the mapping from $g(x)$ to $f(x) - h(x)$.
> In this way, the exploration network is able to learn a flexible form of $\Phi$ that adapts to different datasets. In contrast, the statistical form of $\Phi$ is fixed on any dataset. The reviewer may want to refer to [10] for more details.
>
>
> **Q5**: Existence of $\theta^\ast$.
>
> **A5**: $\theta^\ast$ always exists as long as the parameter space of the neural network is large enough (controlled by $m$). This comes from the universal approximation property of neural networks, i.e., given any $T$ data points, there exists at least a solution for the neural network to fit these $T$ data points. Therefore, $\theta^\ast$ always exists but will depend on the dataset, i.e.,  $\theta^\ast$ varies on different $T$ or different datasets.
>
>
>
> Thank you very much for your time and insightful comments again.

---

> > ### Comment · Reviewer_5FNs · 2023-11-23
> > **Reply to authors' response**
> >
> > I have read the authors' responses and thank them for answering my questions. I will keep my review scores as they were.

---

### Official Review · Reviewer_73vR · 2023-11-02

**Soundness:** 3 good
**Presentation:** 3 good
**Contribution:** 2 fair
**Rating:** 6
**Confidence:** 2

**Summary:**

The paper presents two algorithms for active learning using neural network approximations, specifically designed for stream-based and pool-based scenarios. The goal of these algorithms is to address the challenges posed by the number of classes (K) in bandit-based approaches, such as performance degradation and increased computational costs.

**Strengths:**

The paper is both intuitive and theoretically sound. It introduces a new exploitation network and exploration network that take the original instance as input and simultaneously output the predicted probabilities for K classes. This approach eliminates the need to transform the instance into DK long vectors. The paper provides theoretical performance guarantees, showing a slower error-growth rate as K increases. Furthermore, it demonstrates that the proposed algorithms achieve the optimal active learning rate under different noise conditions.

**Weaknesses:**

From a theoretical perspective, the core difference between existing methods that transform instances into DK long vectors is not clear.

**Questions:**

See Weaknesses

---

> ### Author Response · Authors · 2023-11-17
> **Response to Reviewer 73vR**
>
> Thank you very much for your constructive comments and we are glad to provide the response to your questions.
>
> **Q**: Theoretical difference from existing methods.
>
> **A**: Because the bandit-based approaches (e.g., [48]) transform each class to an arm, they only use the reward of the selected arm (one class) for the back-propagation of neural networks and use the bandit technique to build the confidence bound for the network estimation. To be specific, they build the ellipsoid confidence ball based on the ridge regression in the RKHS spanned by NTK, adopted from [56], which causes the dependence on $L_{\mathbf{H}}$. In contrast, our approach directly takes the label vector for the back-propagation of neural networks and we construct the generalization bound of online gradient descent as the confidence bound, which is distinct from the existing bandit-based approaches, removing the dependence on $L_{\mathbf{H}}$. We sincerely invite the reviewer to read our global response, where we provide detailed information regarding the performance improvement with varying $\alpha$.
>
>
> Thank you very much for your time and insightful comments again.

---

### Author Response · Authors · 2023-11-17
**Global Response: Updates of Manuscript**

We really appreciate all reviewers' constructive feedback and time in reviewing our paper. Here, we would like to provide the summary regarding the updates of the manuscript:

**(1) Experiments**:  Per Reviewer Jpt8's suggestion, we've extended our experiments from 5 runs to 10 runs (Table 1 and Table 2). The new results are consistent with the previous results and conclusion.

**(2) Theoretical analysis**: We've added a new theorem in Section F to show the complexity of regret and label queries with varying $\alpha$ in binary classification. The results are listed as follows.

Our results:

\begin{aligned}
\mathbf{R}\_{stream}(T) \leq \widetilde{\mathcal{O}}( (S^2 )^{\frac{\alpha+1}{\alpha+2}} T^{\frac{1}{\alpha + 2}}),  \\\\
\mathbf{N}(T) \leq \widetilde{\mathcal{O}}( (S^2 )^{\frac{\alpha}{\alpha+2}} T^{\frac{2}{\alpha + 2}}).
\end{aligned}


For comparison, the results in [48]:


\begin{aligned}
\mathbf{R}\_{stream}(T) & \leq \widetilde{\mathcal{O}} \left ( {L_{\mathbf{H}}}^{\frac{2(\alpha +1)}{\alpha+2}}  T^{\frac{1}{\alpha +2}} \right) +   \widetilde{\mathcal{O}} \left( {L_\mathbf{H}}^{\frac{\alpha +1}{\alpha+2}} (S^2)^{\frac{\alpha+1}{\alpha+2}}   T^{\frac{1}{\alpha +2}} \right),  \\\\
\mathbf{N}(T) &\leq \widetilde{\mathcal{O}} \left ( {L_\mathbf{H}}^{\frac{2\alpha}{\alpha+2}}  T^{\frac{2}{\alpha +2}} \right) +   \widetilde{\mathcal{O}} \left( {L_\mathbf{H}}^{\frac{\alpha}{\alpha+2}} (S^2)^{\frac{\alpha}{\alpha+2}}   T^{\frac{2}{\alpha +2}} \right).
\end{aligned}



For the regret $\mathbf{R}\_{stream}(T)$, compared to [48], our result removes the term $\widetilde{\mathcal{O}} \left ( {L_{\mathbf{H}}}^{\frac{2(\alpha +1)}{\alpha+2}}  T^{\frac{1}{\alpha +2}} \right)$ and further improves the regret upper bound by a multiplicative factor $ {L_\mathbf{H}}^{\frac{\alpha +1}{\alpha+2}}$. For the label complexity $\mathbf{N}(T)$, compared to [48], our result removes the term
$\widetilde{\mathcal{O}} \left ( {L_\mathbf{H}}^{\frac{2\alpha}{\alpha+2}}  T^{\frac{2}{\alpha +2}} \right)$ and further improves the regret upper bound by a multiplicative factor of $ {L_\mathbf{H}}^{\frac{\alpha}{\alpha+2}}$. It is noteworthy that  $L_\mathbf{H}$ grows linearly with respect to $T$, i.e., $L_\mathbf{H} \geq T \log (1 + \lambda_0)$.

---

### Meta-Review · Area_Chair_gjaZ · 2023-12-08

**Metareview:**

The paper presents two novel algorithms for active learning using neural network approximations, specifically tailored for stream-based and pool-based scenarios. The key innovation lies in the utilization of neural networks for exploration and exploitation, which effectively addresses the challenges posed by the number of classes (K) in bandit-based approaches. The algorithms show a significant reduction in computational costs and a slower error-growth rate as K increases. The paper also provides theoretical performance guarantees and demonstrates the algorithms' effectiveness through extensive experiments.

Strengths:
+ The paper offers robust theoretical guarantees and introduces novel input-output structures for neural networks, improving the dimensionality and computational efficiency.
+ Extensive experiments validate the algorithms' effectiveness, demonstrating superiority over existing methods in terms of accuracy and efficiency.
+ New exploitation and exploration networks and unique approaches for label-query decision-making are introduced.

Weaknesses:
- The paper could provide a clearer distinction between its approach and existing methods, particularly in theoretical aspects.
- The paper mainly focuses on MLP, raising questions about its applicability to other neural network architectures.
- Some terms and bounds used in the paper might be challenging for practitioners to quantify or apply in real-world scenarios.

**Justification For Why Not Higher Score:**

While the technical contributions are sufficient, the lack of clear theoretical differentiation from existing methods and potential issues in generalizability and practical applicability remains a concern. Additionally, the focus on specific network structures without extensive discussion on broader applicability limits its scope.

**Justification For Why Not Lower Score:**

The paper introduced an innovative approach in active learning, supported by comprehensive theoretical analysis and substantial empirical evidence. The improvements in computational efficiency and performance in active learning contexts are noteworthy contributions to the field.

---

### Decision · Program_Chairs · 2024-01-16

Accept (poster)